# Beyond Euclidean Summaries: Online Change Point Detection for Distribution-Valued Data

**Yingyan Zeng** [1]   **Yujing (Zipan) Huang** [2]   **Xiaoyu Chen** [2]

## Abstract

Existing online change-point detection (CPD) methods rely on fixed-dimensional Euclidean summaries, implicitly assuming that distributional changes are well captured by moment-based or feature-based representations. They can obscure important changes in distributional shape or geometry. We propose an intrinsic distribution-valued CPD framework that treats streaming batch data as a stochastic process on the 2-Wasserstein space. Our method detects changes in the law of this process by mapping each empirical distribution to a tangent space relative to a pre-change Fréchet barycenter, yielding a reference-centered local linearization of 2-Wasserstein space. This representation enables sequential detectors by adapting classical multivariate monitoring statistics to tangent fields. We provide theoretical guarantees and demonstrate, via synthetic and real-world experiments, that our approach detects complex distributional shifts with reduced detection delay at matched $\mathrm{ARL}_0$ compared with moments-based and model-free baselines. The code is available at `https://github.com/yyzeng43/IDD-icml`.

## 1. Introduction

Online change point detection (CPD) aims to detect an abrupt change in a sequential data stream as quickly as possible while controlling false alarms. Formally, one constructs a stopping time $\tau$ and seeks small detection delay under post-change regimes subject to a lower bound on the no-change average run length $\mathrm{ARL}_0 = \mathbb{E}_\infty[\tau]$ (Page, 1954; Truong et al., 2020). Online CPD is a fundamental

tool for monitoring nonstationarity and distribution shift in deployed learning systems, with applications spanning large-scale Machine Learning (ML) services, healthcare and bioinformatics, finance, and social platforms (Aminikhanghahi & Cook, 2017; Aminikhanghahi et al., 2018; Truong et al., 2019).

Most online CPD methodologies are formulated for point-valued streams $\{x_t\}$ in Euclidean space. In model-free settings, a common construction is to compare "past" and "recent" segments via a sequential two-sample statistic (often kernel-based) and to stop when this statistic exceeds a threshold (Harchaoui et al., 2008; Li et al., 2015; Chang et al., 2019). More recent work targets high-dimensional streams through computationally efficient, constant-memory recursions (*e.g.*, random-feature approximations and multi-timescale exponential averages) (Keriven et al., 2020). At the same time, CPD has been extended beyond Euclidean vectors to random objects located on manifolds or more general metric spaces (Wang et al., 2013; Bouchard et al., 2020; Dubey & Müller, 2020; Wang et al., 2024).

However, many modern sensing and observational systems produce batches of observations at each time step (*e.g.*, thousands of cells in flow cytometry, or a burst of user-generated content). The natural representation of the observation at time $t$ is the empirical measure $\mu_t = \frac{1}{N_t} \sum_{j=1}^{N_t} \delta_{X_{t,j}}$, where $X_{t,j} \in \mathbb{R}^d$ denotes the $N_t$ observed realizations in the batch at time $t$, and CPD becomes testing for changes in the law of a distribution-valued process $\{\mu_t\}_{t \geq 1}$ (Horváth et al., 2021). In this regime, one may either (i) compare batches via sequential two-sample discrepancies, or (ii) monitor a vector-valued functional of $\mu_t$ obtained from moments or learned feature maps. These strategies effectively test changes in selected functionals of $\mu_t$. Consequently, they can have limited power when the departure is predominantly geometric, *i.e.*, when probability mass is redistributed through deformations of the support (translations, shape changes, multimodal reweighting) rather than through changes in low-order moments. This motivates treating each batch as a distribution-valued observation and modeling $\{\mu_t\}$ intrinsically on $(\mathcal{P}_2(\mathcal{X}), W_2)$, where the Wasserstein metric directly quantifies such mass-transport (geometric) changes.

We therefore treat $\{\mu_t\}$ as a stochastic process taking val-

---

[1]Department of Mechanical, Materials, and Industrial Engineering, University of Cincinnati, Cincinnati, USA. [2]Department of Industrial and Systems Engineering, University at Buffalo, Buffalo, USA. Correspondence to: Xiaoyu Chen \<xchen325@buffalo.edu\>.

*Proceedings of the 43$^{rd}$ International Conference on Machine Learning*, Seoul, South Korea. PMLR 306, 2026. Copyright 2026 by the author(s).

ues in the metric space $(\mathcal{P}_2(\mathcal{X}), W_2)$. Under quadratic cost, the 2-Wasserstein distance metrizes discrepancies in mass displacement and is sensitive to changes in support and shape that are not well characterized by low-order moments (Panaretos & Zemel, 2019). From a computational perspective, modern optimal transport (OT) solvers (*e.g.*, entropic regularization and Sinkhorn iterations) make these geometric comparisons tractable at scale (Cuturi, 2013). For sequential monitoring, however, one must further construct a representation that (i) respects the underlying Wasserstein geometry, (ii) lives in a common coordinate system across time, and (iii) admits principled calibration of false-alarm control.

We address this by anchoring at a pre-change Fréchet barycenter $\bar{\mu}$ and mapping each $\mu_t$ to a tangent space through OT. Under mild regularity, the Brenier map yields the radial identity $W_2^2(\bar{\mu}, \mu_t) = \|T_{\bar{\mu}}^{\mu_t} - \mathrm{Id}\|_{L^2(\bar{\mu})}^2$, which furnishes a local linearization of $(\mathcal{P}_2\mathcal{X}, \mathcal{W}_2)$ at $\bar{\mu}$ in the Hilbert space $L^2(\bar{\mu}; \mathbb{R}^d)$. We then build an online detector by applying multivariate functional PCA with two classical quadratic monitoring statistics (Hotelling's $T^2$ and SPE), using a pre-change calibration step to set thresholds targeting a desired $\mathrm{ARL}_0$.

**Contributions.** We propose an intrinsic distribution-valued CPD (IDD) scheme with the following contributions:

1. We formulate online CPD for a distribution-valued stochastic process $\{\mu_t\}_{t \geq 1}$ and detect changes in its law by transporting each $\mu_t$ to a pre-change Fréchet barycenter $\bar{\mu}$, thereby working directly with Wasserstein geometry rather than moment-based or *ad hoc* Euclidean summaries.

2. We develop a two-chart online monitoring procedure leveraging Multivariate Functional PCA (MFPCA) to derive Hotelling's $T^2$ and Squared Prediction Error (SPE) statistics in the Wasserstein tangent space, providing a scalable detector with data-driven threshold calibration.

3. We provide theoretical guarantees and demonstrate performance on extensive simulations and two real-world case studies (FlowCAP abnormal-cell detection and public opinion monitoring on Reddit), where distributional geometry shifts are central. Across these settings, the proposed approach consistently achieves strong detection power for distributional and dependence-geometry perturbations while outperforming moment-based or model-free online CPD approaches.

## 2. Related Work

**Statistical Process Control.** The foundation of statistical process control (SPC) was established by Shewhart (1931), who introduced control charts for monitoring process stability using summary statistics such as sample mean and range. While simple and interpretable, this framework compresses data into low-dimensional statistics, thereby missing shifts in the overall distribution shape. Multivariate extensions have been developed to handle vector-valued processes. Crosier (1988) developed multivariate Cumulative Sum (CUSUM) procedures, while Nelson (1984) introduced supplementary pattern tests for Shewhart charts. However, these rules do not generalize naturally to distribution-valued data. Lowry & Montgomery (1995) provided a review of multivariate SPC methods, noting their limitations in handling non-Euclidean or high-dimensional data. Crucially, Woodall & Montgomery (1999) identified the challenges caused by the shift of distributions in pre-change phase. Our work addresses this gap by directly monitoring distributions in the Wasserstein space, capturing geometric shifts that summary-based charts would miss.

**Statistics for Distribution-valued Samples.** Optimal transport provides the geometric foundation for our work. Panaretos & Zemel (2019) provided a comprehensive review of Wasserstein statistics. Bhattacharya & Patrangenaru (2005) established the asymptotic theory for Fréchet means on manifolds. Petersen & Müller (2016) proposed transforming densities to a Hilbert space for regression analysis, and Bigot et al. (2017) developed geodesic principal component analysis (PCA) in Wasserstein space, though the latter is computationally demanding. While Chernozhukov et al. (2017) introduced OT-based ranks and Sommerfeld & Munk (2018) derived inference for empirical Wasserstein distances, these works focus largely on static analysis or batch testing. Similarly, energy statistics (Székely & Rizzo, 2013) and kernel two-sample tests (Gretton et al., 2012) are powerful for hypothesis testing but lack the sequential control limits required for CPD. Our contribution is to operationalize these geometric insights into a CPD scheme.

**Change-Point Detection.** Our work builds upon the expanding literature on change-point detection (CPD) for non-Euclidean data. Xie et al. (2021) provides a comprehensive review of classical sequential detection methods. For metric-valued data that belongs to a metric space, Dubey & Müller (2020) established a framework for Fréchet change-point detection, while Zhang et al. (2025) and Jiang et al. (2024) developed self-normalization based inference for object-valued time series, respectively. Specific to probability distributions, Horváth et al. (2021) proposed a sequential monitoring procedure based on $L^2$-Wasserstein distances. However, these approaches often rely on scalar distances or operate in univariate settings, ignoring the direction of distributional shift.

# 3. Intrinsic Distribution-valued CPD

## 3.1. Preliminaries and Assumptions

At each time index $t \in \{1, 2, \dots\}$ we observe on-line streaming batch data, where a batch of $N_t$ samples $\{x_{t,1}, \dots, x_{t,N_t}\} \subset \mathcal{X} \subset \mathbb{R}^d$ ($X_{t,j}$ denotes the random variable and $x_{t,j}$ its realization). We represent the batch by its empirical measure

$$\mu_t := \frac{1}{N_t} \sum_{j=1}^{N_t} \delta_{x_{t,j}} \in \mathcal{P}_2(\mathcal{X}),$$

and view $\{\mu_t\}_{t \geq 1}$ as a distribution-valued time series sampled from a stochastic process taking values in the metric space $(\mathcal{P}_2(\mathcal{X}), W_2)$. Let $\mathcal{P}(\mathcal{X})$ denote the set of Borel probability measures on $\mathcal{X}$ and define

$$\mathcal{P}_2(\mathcal{X}) := \left\{ \mu \in \mathcal{P}(\mathcal{X}) : \int_{\mathcal{X}} \|x\|^2 \, \mu(dx) < \infty \right\}.$$

For $\mu, \nu \in \mathcal{P}_2(\mathcal{X})$, the squared 2-Wasserstein distance in *Monge* form is

$$W_2^2(\mu, \nu) := \inf_T \left\{ \int_{\mathcal{X}} \|x - T(x)\|^2 \, \mu(dx) : \right.$$
$$\left. T : \mathcal{X} \to \mathcal{X} \text{ Borel}, \ T_\# \mu = \nu \right\}, \quad (1)$$

where $T_\# \mu$ denotes the pushforward measure, *i.e.*, $T_\# \mu(B) = \mu(T^{-1}(B))$ for all Borel sets $B \subset \mathcal{X}$. When an optimal map exists, Eq. (1) coincides with the *Kantorovich* formulation:

$$W_2^2(\mu, \nu) = \inf_{\pi \in \Pi(\mu, \nu)} \int_{\mathcal{X} \times \mathcal{X}} \|x - y\|^2 \, \pi(dx, dy), \quad (2)$$

where $\Pi(\mu, \nu)$ denotes the set of couplings with marginals $\mu$ and $\nu$.

**Assumption 3.1** (Within-batch sampling). For each $t$, conditional on an underlying distribution $\nu_t \in \mathcal{P}_2(\mathcal{X})$, the observations within the batch are i.i.d.: $x_{t,1}, \dots, x_{t,N_t} \overset{iid}{\sim} \nu_t$.

We are interested in detecting the potential change in the distribution-valued sequence $\{\nu_t\}_{t=1}^n$ online, formulated as:

$$H_0 : \nu_t \overset{iid}{\sim} \mathbb{P}_1, \qquad t = 1, \dots, n, \quad (3)$$

where $\mathbb{P}_1$ is the common distribution of the samples (*i.e.*, the parent law) on the space of measures $\mathcal{P}_2(\mathcal{X})$. Note that $\mu_t$ is the empirical measure approximating the true underlying distribution $\nu_t$. Following the standard convention in sequential analysis (Page, 1954; Truong et al., 2020), we write $\mathbb{P}_\infty := \mathbb{P}_1$ and $\mathbb{E}_\infty := \mathbb{E}_{\mathbb{P}_1}$ to denote probability and expectation under the pre-change regime. Under the alternative, there exists an unknown change point $\kappa \in \{1, \dots, n\}$ such that

$$t < \kappa, \ \nu_t \sim \mathbb{P}_1, \quad t \geq \kappa, \ \nu_t \sim \mathbb{P}_2, \quad \text{where } \mathbb{P}_1 \neq \mathbb{P}_2. \quad (4)$$

For clarity of exposition, Eq. (4) is stated for a single change point. However, the proposed detection method can naturally detect multiple change points online.

We assume a pre-change period of the length $M \geq 1$ of the process, *i.e.*, $\nu_1, \dots, \nu_M \overset{iid}{\sim} \mathbb{P}_1$. Let $\bar{\mu}$ denote the Fréchet barycenter of the measures in the pre-change calibration phase:

$$\bar{\mu} \in \arg \min_{\nu \in \mathcal{P}_2(\mathcal{X})} \frac{1}{M} \sum_{t=1}^M W_2^2(\nu, \mu_t). \quad (5)$$

**Assumption 3.2** (Regularity of the reference measure). $\bar{\mu}$ is absolutely continuous with respect to Lebesgue measure.

*Remark* 3.3. Under quadratic cost $c(x, y) = \|x - y\|^2$, Assumption 3.2 implies that for any $\mu \in \mathcal{P}_2(\mathbb{R}^d)$ there exists an optimal transport map $T_{\bar{\mu} \to \mu}$ pushing $\bar{\mu}$ to $\mu$, which is $\bar{\mu}$-a.e. unique (Brenier's theorem). Moreover, the McCann interpolation

$$\mu_s := \left( (1 - s)\mathrm{Id} + s T_{\bar{\mu} \to \mu} \right)_\# \bar{\mu}, \qquad s \in [0, 1],$$

defines the unique $W_2$-geodesic from $\bar{\mu}$ to $\mu$ (Brenier, 1991; Villani et al., 2008). In our implementation, $\mu_t$ is empirical and we compute a Kantorovich plan and its barycentric projection (see Remark 3.5).

## 3.2. Local Geometric Linearization via Optimal Transport

To enable sequential inference on distribution-valued data, we map the empirical measure $\mu_t$ into a common Hilbert space by representing it through the optimal-transport displacement from the pre-change Fréchet barycenter $\bar{\mu}$.

**Proposition 3.4** (Radial Isometry, Theorem 8.6 (Villani, 2009)). *Assume* $\bar{\mu} \in \mathcal{P}_2(\mathbb{R}^d)$ *is absolutely continuous with respect to Lebesgue measure, and consider the quadratic cost* $c(x, y) = \|x - y\|^2$. *Then for any* $\mu \in \mathcal{P}_2(\mathbb{R}^d)$ *there exists a* $\bar{\mu}$-a.e. *unique optimal transport map* $T_{\bar{\mu}}^\mu$ *such that* $(T_{\bar{\mu}}^\mu)_\# \bar{\mu} = \mu$. *Defining the logarithm map in the tangent space*

$$v(\mu) := T_{\bar{\mu}}^{\mu_t} - \mathrm{Id} \ \in L^2(\bar{\mu}; \mathbb{R}^d),$$

*we have the identity*

$$W_2^2(\bar{\mu}, \mu) = \int_{\mathcal{X}} \|v(\mu)(x)\|^2 \, d\bar{\mu}(x) = \|v(\mu)\|_{L^2(\bar{\mu})}^2. \quad (6)$$

Proposition 3.4 establishes that the Wasserstein distance from the barycenter $\bar{\mu}$ is exactly preserved by the $L^2(\bar{\mu})$ norm of the OT displacement field. Motivated by this radial isometry, we represent each distribution-valued sample $\mu_t$ by its linearized embedding, *i.e.*, OT displacement field:

$$v_t := T_{\bar{\mu}}^{\mu_t} - \mathrm{Id} \ \in L^2(\bar{\mu}; \mathbb{R}^d). \quad (7)$$

Geometrically, $v_t$ corresponds to the Wasserstein logarithm map at reference measure $\bar{\mu}$, projecting the measure $\mu_t$ onto the tangent space at $\bar{\mu}$. This transformation maps the manifold $\mathcal{P}_2(\mathcal{X})$ into the tangent space $H := L^2(\bar{\mu}; \mathbb{R}^d)$, allowing us to perform linear statistical analysis in the Hilbert space. Consequently, the problem of monitoring distributional shifts is transformed to monitoring a sequence of multivariate functional data, enabling standard covariance operators, functional analysis, and Hilbert-space statistics to be constructed directly on $\{v_t\}$, avoiding distributional information loss with Euclidean summaries.

*Remark* 3.5. In practice, observed batches are discrete, where a deterministic Monge map may not exist. We therefore solve the Kantorovich relaxation to obtain an optimal coupling $\pi_t \in \Pi(\bar{\mu}, \mu_t)$ and recover a map via the barycentric projection:

$$T_t^\pi(x) := \mathbb{E}_{\pi_t}[Y \mid X = x] = \int y\,\pi_t(dy \mid x), \quad (8)$$

We then define the tangent field as $v_t(x) := T_t^\pi(x) - x$. This projection provides a canonical vector field in $L^2(\bar{\mu}; \mathbb{R}^d)$ that coincides with the true Monge map whenever the latter exists. For simplicity, we henceforth denote this projected field simply as $v_t$.

To quantify the approximation gap when using a plan-based projection instead of a Monge map, we provide the following variance decomposition, which is a direct application of the $L^2$-projection property of conditional expectation (Doob, 1953; Villani, 2021).

**Proposition 3.6** (Barycentric projection variance decomposition)**.** *Let $\mu, \nu \in \mathcal{P}_2(\mathcal{X})$ and let $\pi \in \Pi(\mu, \nu)$ be any coupling with $(X, Y) \sim \pi$. Define the barycentric projection $T^\pi(x) := \mathbb{E}_\pi[Y \mid X = x]$ and the displacement field $v^\pi(x) := T^\pi(x) - x \in L^2(\mu; \mathbb{R}^d)$. Then the following decomposition holds:*

$$\int_\mathcal{X} \|x - T^\pi(x)\|^2\,\mu(dx) = \int_{\mathcal{X} \times \mathcal{X}} \|x - y\|^2\,\pi(dx, dy)$$
$$- \int_\mathcal{X} \mathbb{E}_\pi\big[\|Y - T^\pi(X)\|^2 \mid X = x\big]\,\mu(dx) \quad (9)$$

*and in particular*

$$\|v^\pi\|_{L^2(\mu)}^2 = \int_\mathcal{X} \|x - T^\pi(x)\|^2\,\mu(dx) \quad (10)$$
$$\leq \int_{\mathcal{X} \times \mathcal{X}} \|x - y\|^2\,\pi(dx, dy). \quad (11)$$

*If $\pi^\star$ is an optimal coupling for $W_2^2(\mu, \nu)$, then $\|v^{\pi^\star}\|_{L^2(\mu)}^2 \leq W_2^2(\mu, \nu)$, with equality if and only if $\pi^\star$ is induced by a deterministic map (equivalently, $\mathrm{Var}_{\pi^\star}(Y \mid X) = 0$ $\mu$-a.e.).*

### 3.3. The Optimal Transport-based Detector

Having mapped the distribution-valued stream data to the reference tangent space, we formulate the detection task as monitoring a sequence of multivariate functional observations. In online CPD for high-dimensional and functional streams, the standard paradigm is subspace-based detection. This strategy projects the functional observation onto a low-dimensional principal subspace (*e.g.,* via Multivariate Functional PCA (MFPCA)) (Kuncheva, 2011; Skubalska-Rafajłowicz, 2013; Bakdi & Kouadri, 2017). Motivated by this paradigm, we apply MFPCA to the displacement fields to obtain a finite-dimensional representation and orthogonal residual component, which produces an OT-based detector built from score-space statistic (*i.e.,* functional Hotelling $T^2$) and a complementary residual-energy statistic (*i.e.,* Squared Prediction Error (SPE)) on the tangent space.

Let $\{v_t\}_{t=1}^{n_0} \subset H := L^2(\bar{\mu}; \mathbb{R}^d)$ denote a reference sample of $n_0$ tangent vector fields collected from the pre-change process. Define the pre-change sample mean $\bar{v} = \frac{1}{n_0} \sum_{t=1}^{n_0} v_t$ and centered fields $\tilde{v}_t = v_t - \bar{v}$. The empirical covariance operator $\widehat{C} : H \to H$ is

$$(\widehat{C}f)(\cdot) = \frac{1}{n_0 - 1} \sum_{t=1}^{n_0} \langle f, \tilde{v}_t \rangle_H\,\tilde{v}_t(\cdot)$$
$$= \frac{1}{n_0 - 1} \sum_{t=1}^{n_0} (\tilde{v}_t \otimes \tilde{v}_t)f, \quad (12)$$

where $(a \otimes b)f := \langle f, b \rangle_H\,a$.

Due to the finite rank of the empirical covariance operator $\widehat{C}$, we employ a spectral decomposition strategy to regularize the inversion, justified by the Karhunen-Loève theorem. Let $\{(\widehat{\lambda}_m, \widehat{\phi}_m)\}_{m \geq 1}$ denote the eigenpairs derived from the pre-change reference sample, ordered such that $\widehat{\lambda}_1 \geq \widehat{\lambda}_2 \geq \cdots \geq 0$, with orthonormal eigenfunctions $\{\widehat{\phi}_m\}$ spanning the principal directions of variation in $H$. For any newly observed tangent vector field $v_t$ from the online stream, we compute its centered displacement $\Delta_t := v_t - \bar{v}$ relative to the pre-change mean. By projecting $\Delta_t$ onto the eigenbasis $\{\widehat{\phi}_m\}$, we obtain the scores $\widehat{\xi}_{tm} := \langle \Delta_t, \widehat{\phi}_m \rangle_H$. By truncating the Karhunen-Loève expansion to the leading $K$ dimensions, we construct the functional Hotelling's $T^2$ statistic for the principal subspace:

$$T_{t,K}^2 = \sum_{m=1}^K \frac{\widehat{\xi}_{tm}^2}{\widehat{\lambda}_m}. \quad (13)$$

*Remark* 3.7. In the above MFPCA decomposition, the empirical covariance operator $\widehat{C}$ maps any linear combination of the centered displacement $\tilde{v}_t$ back to the closure of the gradient subspace in $H$. Consequently, its eigenfunctions $\{\phi_m\}_{m \geq 1}$ also lie in this same subspace. Each $\phi_m$ is again

a gradient field, so the principal component basis does not leave the tangent space $T_{\bar{\mu}}\mathcal{P}_2(\mathcal{X})$.

Simultaneously, we monitor residuals orthogonal to this subspace using the SPE, also known as the $Q$–statistic (Jackson & Mudholkar, 1979):

$$\mathrm{SPE}_t = \left|(I - \widehat{P}_K)\Delta_t\right|_H^2 = \sum_{m>K} \widehat{\xi}_{tm}^2, \qquad (14)$$

where $\widehat{P}_K$ denotes the orthogonal projection onto $\mathrm{span}\{\widehat{\phi}_1, \ldots, \widehat{\phi}_K\}$.

Formally, the functional Hotelling's $T^2$ statistic can be written as $T_t^2 = \langle \widehat{C}^\dagger \Delta_t, \Delta_t \rangle_H$, where $\widehat{C}^\dagger$ is the Moore–Penrose pseudoinverse of $\widehat{C}$. The computable expression Eq. (13) follows from the spectral decomposition of $\widehat{C}$ (see Appendix E).

### 3.4. Theoretical Properties and Threshold Calibration

To derive the asymptotic behavior of the monitoring statistics, we introduce a regularity condition on the distributional generation process. Motivated by Central Limit Theorems for Fréchet means on manifolds (Bhattacharya & Bhattacharya, 2008; Mattingly et al., 2023), we model the pre-change tangent fields as a Gaussian random field.

**Assumption 3.8** (Pre-change Gaussian Structure). The pre-change tangent vector fields $\{v_t\}_{t \leq n_0}$ are independent and identically distributed (i.i.d.) realizations of a mean-zero Gaussian Random Element $V$ in the Hilbert space $H = L^2(\bar{\mu}; \mathbb{R}^d)$. The process is fully characterized by a compact, positive trace-class covariance operator $\Gamma : H \to H$, with eigenvalues $\lambda_1 > \lambda_2 > \cdots > 0$.

Under Assumption 3.8, the functional Hotelling $T^2$ statistic defined in Eq. (13) follows a known limiting distribution.

**Proposition 3.9** (Asymptotic Null Distribution). *Let the pre-change sample size $n_0 \to \infty$. Under standard consistency conditions for empirical functional eigenpairs (Hall & Hosseini-Nasab, 2006), for any fixed truncation level $K$, the online Hotelling statistic $T_{t,K}^2$ for a new pre-change observation converges in distribution to a chi-squared random variable:*

$$T_{t,K}^2 \xrightarrow{d} \chi_K^2. \qquad (15)$$

*Simultaneously, the residual statistic $\mathrm{SPE}_t$ converges to a weighted sum of independent chi-squared variables, $\sum_{m>K} \lambda_m Z_m^2$, where $Z_m \sim \mathcal{N}(0,1)$.*

*Proof.* See Appendix C. $\square$

**Calibration Strategy.** While Proposition 3.9 suggests a parametric threshold $h_{T^2} = \chi_{K,1-\alpha}^2$, real-world distribution-valued streams may exhibit non-Gaussian tail behavior. To ensure robustness, we adopt a non-parametric calibration approach. We set the detection thresholds $h_{T^2}$ and $h_{\mathrm{SPE}}$ as the empirical $(1 - \alpha)$ quantiles of the pre-change statistics $\{T_{t,K}^2\}_{t=1}^{n_0}$ and $\{\mathrm{SPE}_t\}_{t=1}^{n_0}$. This data-driven strategy automatically accounts for finite-sample effects and the intractable infinite sum of eigenvalues in the SPE limit.

For distribution-valued streams, the null law of $(T_{t,K}^2, \mathrm{SPE}_t)$ is not available in closed form because it depends on the unknown pre-change distribution of tangent fields induced by the OT construction. Once the stream has been mapped into the monitoring statistics and a fixed threshold pair $(h_{T^2}, h_{\mathrm{SPE}})$ has been chosen, sequential false-alarm control reduces to controlling the one-step exceedance probability under $\mathbb{P}_\infty$. The following theorem makes this reduction explicit (under i.i.d. monitoring statistics), and the corollary links our empirical-quantile calibration to an explicit finite-sample $\mathrm{ARL}_0$ lower bound.

**Theorem 3.10** (Sequential false-alarm control under fixed thresholds). *Let $(h_{T^2}, h_{\mathrm{SPE}})$ be fixed thresholds. Under the assumption that the statistics are i.i.d. during the monitoring phase, the run-length $\tau$ follows a geometric distribution with success probability $p_\infty = \mathbb{P}_\infty(T_{1,K}^2 > h_{T^2} \cup \mathrm{SPE}_1 > h_{\mathrm{SPE}})$. Consequently, the global Average Run Length is given by:*

$$\mathrm{ARL}_0 = n_0 + 1 + \frac{1}{p_\infty} \geq n_0 + 1 + \frac{1}{\alpha_{T^2} + \alpha_{\mathrm{SPE}}}. \qquad (16)$$

**Corollary 3.11** (Empirical-quantile calibration implies ARL control). *If thresholds are set as the $(1-\alpha)$ empirical quantiles from a calibration sample of size $n_0$, the marginal exceedance probabilities for an independent monitoring-phase observation satisfy $\mathbb{P}_\infty(\cdot > h) \leq \alpha + \frac{1}{n_0+1}$. Substituting this into Theorem 3.10 yields the finite-sample guarantee:*

$$\mathrm{ARL}_0 \geq n_0 + 1 + \frac{1}{\alpha_{T^2} + \alpha_{\mathrm{SPE}} + 2/(n_0 + 1)}.$$

**Proposition 3.12** (Detection delay under sustained post-change). *Suppose that for $t \geq \kappa$ the tangent fields $\{v_t\}$ are i.i.d. Gaussian in $H$ with mean $m \neq 0$ and the same covariance operator $\Gamma$ as in Assumption 3.8. Let $m_k = \langle m, \phi_k \rangle$ be the projections of the mean shift onto the pre-change eigenbasis, and set $\delta_T^2 = \sum_{k=1}^K m_k^2/\lambda_k$. Then $T_{t,K}^2 \sim \chi_K^2(\delta_T^2)$ and $\mathrm{SPE}_t$ follows a generalized non-central chi-square. Since $T^2$ and $\mathrm{SPE}$ are functions of orthogonal projections of a Gaussian field, they are independent, and $\mathrm{ARL}_1 = 1/[1 - (1 - p_T)(1 - p_S)]$, where $p_T = \mathbb{P}(T_{t,K}^2 > h_{T^2})$ and $p_S = \mathbb{P}(\mathrm{SPE}_t > h_{\mathrm{SPE}})$. See Appendix J for the derivation and a distribution-free bound.*

**Algorithm 1** IDD— Pre-Change Calibration

**Input** : Pre-change reference stream $\{\mu_t\}_{t=1}^{n_0}$, truncation level $K$, false-alarm rates $\alpha_{T^2}, \alpha_{\mathrm{SPE}}$.

**Phase 1: Reference Estimation** Compute Fréchet barycenter $\bar{\mu}$ of $\{\mu_t\}_{t=1}^{n_0}$ via Eq. (5)

**Phase 2: Tangent Space Construction for** $t = 1, \ldots, n_0$ **do**
  Solve OT from $\bar{\mu}$ to $\mu_t$ to get optimal plan $\pi_t$ Compute displacement field: $v_t(x) = \int y\, \pi_t(dy \mid x) - x$
**end**

**Phase 3: Functional PCA** Compute reference mean $\bar{v} = \frac{1}{n_0} \sum_{t=1}^{n_0} v_t$ Center fields: $\tilde{v}_t = v_t - \bar{v}$, for $t = 1, \ldots, n_0$ Perform MFPCA on $\{\tilde{v}_t\}_{t=1}^{n_0}$ to obtain eigenpairs $\{(\widehat{\phi}_m, \widehat{\lambda}_m)\}_{m=1}^{n_0-1}$

**Phase 4: Threshold Calibration for** $t = 1, \ldots, n_0$ **do**
  $\Delta_t \leftarrow v_t - \bar{v}$  $\widehat{\xi}_{tm} \leftarrow \langle \Delta_t, \widehat{\phi}_m \rangle_H, \quad m = 1, \ldots, K$
  $T_{t,K}^2 \leftarrow \sum_{m=1}^{K} \widehat{\xi}_{tm}^2 / \widehat{\lambda}_m$  [Eq. (13)]  $\mathrm{SPE}_t \leftarrow \|\Delta_t\|_H^2 - \sum_{m=1}^{K} \widehat{\xi}_{tm}^2$  [Eq. (14)]
**end**
$h_{T^2} \leftarrow (1 - \alpha_{T^2})$-quantile of $\{T_{t,K}^2\}_{t=1}^{n_0}$  $h_{\mathrm{SPE}} \leftarrow (1 - \alpha_{\mathrm{SPE}})$-quantile of $\{\mathrm{SPE}_t\}_{t=1}^{n_0}$
**Output :** $\bar{\mu}, \bar{v}, \{(\widehat{\phi}_m, \widehat{\lambda}_m)\}_{m=1}^{K}, h_{T^2}, h_{\mathrm{SPE}}$.

---

**Algorithm 2** IDD— Online Monitoring

**Input :** Calibration output $(\bar{\mu}, \bar{v}, \{(\widehat{\phi}_m, \widehat{\lambda}_m)\}_{m=1}^{K}, h_{T^2}, h_{\mathrm{SPE}})$.

**for** $t = n_0 + 1, n_0 + 2, \ldots$ **do**
  Receive new sample $\mu_t$. Solve OT from $\bar{\mu}$ to $\mu_t \rightarrow$ displacement field $v_t$. Compute centered deviation: $\Delta_t \leftarrow v_t - \bar{v}$. Compute scores: $\widehat{\xi}_{tm} \leftarrow \langle \Delta_t, \widehat{\phi}_m \rangle_H$ for $m = 1, \ldots, K$. $T_{t,K}^2 \leftarrow \sum_{m=1}^{K} \widehat{\xi}_{tm}^2 / \widehat{\lambda}_m$  $\mathrm{SPE}_t \leftarrow \|\Delta_t\|_H^2 - \sum_{m=1}^{K} \widehat{\xi}_{tm}^2$ **if** $T_{t,K}^2 > h_{T^2}$ **or** $\mathrm{SPE}_t > h_{\mathrm{SPE}}$
  **then**
    Flag $t$ as a change point; **stop** (or reset and continue)
  **end**
**end**

---

**Monitoring Procedure.** The complete procedure is organized into two stages. Algorithm 1 (pre-change calibration) computes the reference barycenter, tangent-space displacement fields, funcational PCA eigenbasis $\{(\widehat{\phi}_m, \widehat{\lambda}_m)\}_{m=1}^{K}$, and empirical-quantile thresholds from the pre-change samples. Algorithm 2 (online monitoring) computes $T_{t,K}^2$ and $\mathrm{SPE}_t$ for each incoming sample and raises an alarm when either exceeds its calibrated threshold.

### 3.5. The $\varepsilon$-Isometry Guarantee.

The proposed online detector is efficient only if the sequence of tangent vectors can be accurately approximated. Therefore, we provide the following $\varepsilon$-isometry as a theoretical guarantee for this approximation, showing that the regularity of the optimal transport maps ensures a rapid spectral decay of the covariance operator. We begin by establishing the regularity of the pre-change process.

**Assumption 3.13** (Regularity of transport maps)**.** The pre-change process generates i.i.d. distributions $\{\mu_i\}$ on a convex, bounded domain $\mathcal{X} \subset \mathbb{R}^d$. Each $\mu_i$ admits a density $\rho_i$ that is bounded and $\alpha$-Hölder continuous (for some $\alpha > 0$). Let $T_t : \bar{\mu} \rightarrow \mu_t$ denote the unique optimal transport map from the reference barycenter $\bar{\mu}$ to $\mu_t$, and define the tangent vector field $v_t(x) := T_t(x) - x$. By standard regularity results for Monge-Ampère equations (Brenier, 1991; Caffarelli, 1992; 2000), there exist deterministic constants

$L, B < \infty$ such that almost surely:

$$\mathrm{Lip}(v_t) \leq L \quad \text{and} \quad \|v_t\|_{L^\infty(\mathcal{X})} \leq B, \quad \text{for all } t. \quad (17)$$

This geometric regularity of the individual realizations implies the smoothness of the population covariance structure.

**Proposition 3.14** (Lipschitz Continuity of the Covariance Kernel)**.** *Under Assumption 3.13, let $V$ be the random tangent field representing the pre-change process. The matrix-valued covariance kernel $\mathcal{K}(\boldsymbol{x}, \boldsymbol{y}) := \mathbb{E}[V(\boldsymbol{x})V(\boldsymbol{y})^\top]$ is Lipschitz continuous on $\mathcal{X} \times \mathcal{X}$. Specifically,*

$$\|\mathcal{K}(\boldsymbol{x}, \boldsymbol{y}) - \mathcal{K}(\boldsymbol{x}', \boldsymbol{y}')\|_F \leq C_{\mathcal{K}}(\|\boldsymbol{x} - \boldsymbol{x}'\| + \|\boldsymbol{y} - \boldsymbol{y}'\|),$$

*where $C_{\mathcal{K}}$ depends on the Lipschitz constant $L$ and the second moment of the process.*

*Proof.* See Appendix B. □

Leveraging the spectral theory of integral operators, the Lipschitz continuity of the kernel guarantees a specific decay rate for the eigenvalues of the covariance operator, allowing us to bound the truncated detector's approximation error.

**Theorem 3.15** ($\varepsilon$-Isometry)**.** *Let the domain $\mathcal{X} \subset \mathbb{R}^d$ and the covariance kernel $\mathcal{K}$ satisfy the conditions of Proposition 3.14. Let $\{\lambda_m\}_{m \geq 1}$ be the eigenvalues of the covariance operator $\Gamma$ in non-increasing order. Then, there exists a constant $A_{\mathcal{X}}$ depending only on the domain dimension $d$ such that the tail sum of eigenvalues satisfies:*

$$\sum_{m > K} \lambda_m \leq A_{\mathcal{X}} C_{\mathcal{K}} K^{-1/d}. \quad (18)$$

*Consequently, to achieve an $\varepsilon$-isometry in mean square (i.e., relative reconstruction error $\leq \varepsilon^2$), the required number of principal components $K$ scales as:*

$$K \geq \left( \frac{A_{\mathcal{X}} C_{\mathcal{K}}}{\varepsilon^2 \,\mathrm{tr}(\Gamma)} \right)^d \quad (19)$$

*Proof.* The proof extends the trace-bound arguments of Reade (1983) to $d$-dimensional Lipschitz domains via a dyadic partition argument. See Appendix B for the full derivation. ☐

*Remark* 3.16. Theorem 3.15 provides the theoretical justification for our dimension reduction. It ensures that the Wasserstein tangent space can be accurately approximated by a low-dimensional Euclidean subspace with a truncation level $K$ that grows polynomially with the desired precision.

# 4. Numerical Experiments

We evaluate the proposed IDD against baselines on synthetic batch streams (continuous and discrete) and two real-world case studies: (i) Acute Myeloid Leukemia (AML) detection (Aghaeepour et al., 2013), and (ii) Vaccine sentiment monitoring (Brambilla & Kharmale, 2022a). All implementation detail are provided in Appendix H, H, P Q.

**Baselines.** We compare IDD against five representative baselines categorized by their underlying geometric representation. First, Euclidean summary baselines include Shewhart-type charts (Xie et al., 2021) (*e.g.*, Hotelling's $T^2$ or attribute charts), which reduce distributions to simple moments like means or variances. Second, distance-based metric baselines utilize kernel or sketch-based discrepancies, specifically Scan-B (Li et al., 2019) (using Maximum Mean Discrepancy) and NEWMA (Keriven et al., 2020) (using Random Fourier Features). Third, manifold-value baselines explicitly model the space of probability measures but use alternative geometries (Ramsay & Silverman, 2005): Log-KDE monitors functional $L^2$ distances between log-density estimates, while F-CPD (Dubey & Müller, 2020) employs Fréchet variance on graph-based metrics.

**Metrics.** We evaluate performance via the standard false-alarm-delay trade-off. For a stopping time $\tau$ and change-point $\kappa$, we report the in-control average run length $\mathrm{ARL}_0 = \mathbb{E}_\infty[\tau]$ (expected time to false alarm) and the detection delay $\mathrm{ARL}_1 = \mathbb{E}_\kappa[\tau - \kappa \mid \tau > \kappa]$, where $\mathbb{E}_\infty[\cdot]$ denotes the expectation under $H_0$. All methods are calibrated to a fixed target $\mathrm{ARL}_0$ to compare $\mathrm{ARL}_1$ (smaller is better). The pre-change calibration length is $n_0 = 300$ for all synthetic and FlowCAP-II experiments and $n_0 = 50$ for the Reddit study (constrained by data availability). A sensitivity discussion is in Appendix K.

## 4.1. Synthetic Experiments

We generate distribution-valued streams $\{\mu_t\}$ by pushing forward a reference measure $\bar{\mu}$ via convex potentials (continuous), where it is verified to yield optimal transport maps in Appendix G.3, and by simulating count and categorical processes (discrete).

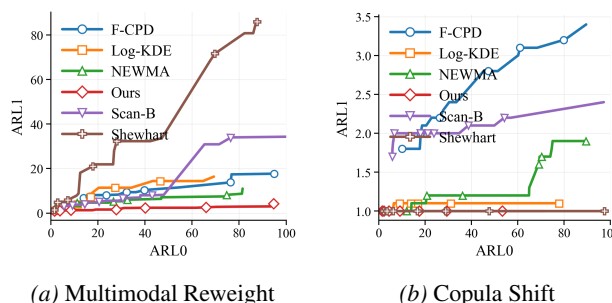

*(a)* Multimodal Reweight  *(b)* Copula Shift

*Figure 1.* $\mathrm{ARL}_1$ vs. $\mathrm{ARL}_0$ comparison on synthetic continuous streams. IDD (red) shows superior detection speed.

**Continuous Streams.** We simulate $d$-dimensional streams ($d \in \{1, 5, 10, 50\}$) with batch sizes $N \in \{50, 100, 300\}$ under three shift scenarios: (1) Barycenter change: shifting the central mass; (2) Multimodal reweight: altering mixture weights of a multimodal distribution; and (3) Copula shift: altering variable dependencies while fixing marginals (see Appendix G for generation details). To ensure scalability in high-dimensional settings ($d \geq 10$), we accelerate computations using Sinkhorn iterations (Cuturi, 2013) and estimate barycenters via Conditional Normalizing Flows (Visentin & Cheridito, 2026) (see Appendix I).

**Results:** Results for Multimodal Reweight ($d = 10$) and Copula Shift ($d = 50$) are shown in Fig. 1a and Fig. 1b. IDD (red curve) consistently achieves the lowest $\mathrm{ARL}_1$ at matched $\mathrm{ARL}_0$, demonstrating superior detection power. Notably, Log-KDE (orange square) remains robust and competitive across both dimensions. In contrast, F-CPD (blue circle) shows reduced sensitivity in the high-dimensional Copula shift ($d = 50$), which might be due to the inefficiency of its graph-based metric in sparse spaces. The Euclidean Shewhart baseline (brown) exhibits inconsistent performance: it fails catastrophically in the Multimodal setting (Fig. 1a) where means are preserved, but detects the covariance change in the Copula shift (Fig. 1b) effectively. Overall, IDD provides resilient performance across diverse scenarios, with full results in Appendix O.

**Discrete Streams.** We evaluate the framework on streams where observations lie on discrete supports. Details are provided in Appendix H. (1) Poisson Counts. For post-change, we introduce a Spike Injection shift, where a small fraction of counts are replaced by a fixed high-value outlier ($k^*$), simulating rare burst events. As shown in Fig. 2a, IDD significantly outperforms Log-KDE. This confirms that the kernel smoothing required by Log-KDE blurs distinct spikes on discrete grids, whereas IDD preserves signal fidelity by leveraging the intrinsic geometry of the full discrete distribution. (2) Ordered Categorical Drift. We simulate sequential survey-type data on an ordinal scale ($d = 6$ classes). The shift involves a Gradual Drift, where probability mass

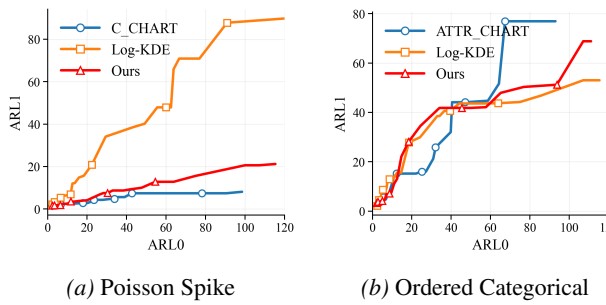

*(a)* Poisson Spike      *(b)* Ordered Categorical

*Figure 2.* Trade-off curves for discrete streams ($N = 100$). IDD handles discrete geometry robustly.

*Table 1.* Gaussian translation: $\mathrm{ARL}_1$ (mean $\pm$ SE) at matched $\mathrm{ARL}_0$; smaller is better. Best log-KDE bandwidth is reported.

| $d$ | $\sigma$ | $\delta_1$ | Hotelling $T^2$ | IDD (Ours) | Best log-KDE |
|---|---|---|---|---|---|
| 1 | 0.5 | 0.1 | **1.8 $\pm$ 0.2 (↑28.0%)** | **1.8 $\pm$ 0.2 (↑28.0%)** | 2.5 $\pm$ 0.6 |
| 1 | 0.5 | 0.5 | **1.0 $\pm$ 0.0 (↑95.1%)** | **1.0 $\pm$ 0.0 (↑95.1%)** | **1.0 $\pm$ 0.0 (↑95.1%)** |
| 1 | 1.0 | 0.1 | 4.2 $\pm$ 1.1 (↑20.8%) | 5.3 $\pm$ 1.6 | 10.8 $\pm$ 2.5 |
| 1 | 1.0 | 0.5 | **1.0 $\pm$ 0.0 (↑9.1%)** | **1.0 $\pm$ 0.0 (↑9.1%)** | **1.0 $\pm$ 0.0 (↑9.1%)** |
| 1 | 2.0 | 0.1 | 16.7 $\pm$ 3.5 | 17.0 $\pm$ 3.4 | **16.4 $\pm$ 2.6** |
| 1 | 2.0 | 0.5 | 1.6 $\pm$ 0.3 | **1.5 $\pm$ 0.2 (↑6.3%)** | 3.8 $\pm$ 1.3 |
| 2 | 0.5 | 0.1 | 1.5 $\pm$ 0.2 | **1.3 $\pm$ 0.2 (↑13.3%)** | 2.3 $\pm$ 0.7 |
| 2 | 0.5 | 0.5 | **1.0 $\pm$ 0.0 (↑96.4%)** | **1.0 $\pm$ 0.0 (↑96.4%)** | **1.0 $\pm$ 0.0 (↑96.4%)** |
| 2 | 1.0 | 0.1 | 2.7 $\pm$ 0.7 | **2.4 $\pm$ 0.6 (↑11.1%)** | 5.7 $\pm$ 1.6 |
| 2 | 1.0 | 0.5 | **1.0 $\pm$ 0.0 (↑9.1%)** | **1.0 $\pm$ 0.0 (↑9.1%)** | **1.0 $\pm$ 0.0 (↑9.1%)** |
| 2 | 2.0 | 0.1 | 11.3 $\pm$ 5.4 | **10.2 $\pm$ 5.3 (↑9.7%)** | 15.4 $\pm$ 5.2 |
| 2 | 2.0 | 0.5 | 1.2 $\pm$ 0.1 | **1.0 $\pm$ 0.0 (↑16.7%)** | 2.6 $\pm$ 0.6 |

slowly migrates to adjacent higher classes. It can be shown that IDD outperforms Shewhart attribute charts (Fig. 2b). Unlike attribute charts, which treat classes as nominal, our method exploits the underlying ordinal metric. This allows it to detect the coherent flow of mass between adjacent ranks more efficiently than bin-wise monitoring.

**Gaussian Translation Analysis.** To rigorously quantify the geometric advantage, we compared IDD against Log-KDE on a pure Gaussian mean-shift $\mathcal{N}(m, \Sigma) \to \mathcal{N}(m + \delta/\sqrt{n}, \Sigma)$. We formalize this in **Theorem F.1** (Appendix F), which proves that KDE smoothing attenuates the signal in log-density space ($\|g_\Delta\|^2 \propto \delta^T(\Sigma + H)^{-1}\delta$), whereas the optimal transport signal remains invariant to smoothing. Empirical validation (Table 1, and Table 2 in Appendix) confirms this theory: IDD achieves up to a 95% reduction in detection delay compared to the best-tuned Log-KDE in high-variance settings.

**Comprehensive comparison across scenarios.** IDD is not uniformly superior. For location shifts, Hotelling $T^2$ is more efficient. Similarly, for count data with known parametric structure, specialized charts outperform IDD. IDD's strength lies in complex geometric shifts (*i.e.*, reweighting, dependency changes, shape deformations), where moment-based detectors lose power. A full per-scenario comparison is in Table 3 and Appendix L.

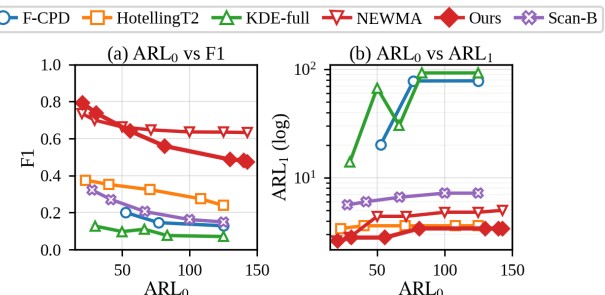

*Figure 3.* AML detection on FlowCAP-II. **Left:** F1-score versus $\mathrm{ARL}_0$. **Right:** detection delay $\mathrm{ARL}_1$ versus $\mathrm{ARL}_0$. IDD (red diamond) attains the highest F1 ($\approx 0.75$) and the lowest $\mathrm{ARL}_1$ across the full $\mathrm{ARL}_0$ range.

### 4.2. Real-World Case Studies

**1. Abnormal Cell Detection** We apply our framework to the detection of Acute Myeloid Leukemia (AML) using the FlowCAP-II dataset (Aghaeepour et al., 2013). Flow cytometry data is naturally distribution-valued: each patient sample consists of thousands of single cells, where each cell is a vector of multidimensional fluorescence markers. We model these patient samples as a distribution-valued stream in $\mathcal{P}_2(\mathbb{R}^7)$. The objective is to detect AML-positive subjects. We calibrate the detector on a reference set of healthy samples and monitor a test stream containing injected AML-positive patients (see Appendix P).

**Results:** Figure 3 reports F1 (left) and detection delay $\mathrm{ARL}_1$ (right) versus $\mathrm{ARL}_0$. IDD attains the highest F1 ($\approx 0.75$) with near-immediate detection ($\mathrm{ARL}_1 \approx 1$) and near-perfect precision ($\approx 0.99$), indicating that our geometric-aware statistic captures the intrinsic shape of the leukemic shift. NEWMA's kernel statistic produces broader alarms (precision $\approx 0.83$, higher recall) and matches IDD on F1 at lenient $\mathrm{ARL}_0$ targets ($30-75$), but its first alarm arrives later, so IDD remains the fastest detector across the full $\mathrm{ARL}_0$ range (see Appendix P). In contrast, Hotelling's $T^2$ yields low precision ($< 0.4$) because reducing distributions to simple moments discards critical sub-population structures. Meanwhile, Log-KDE suffers severe detection latency, confirming its struggle with high-dimensional density estimation.

**2. Public Opinion Monitoring** We analyze the semantic evolution of public discourse using the Reddit Vaccine Sentiment dataset (Brambilla & Kharmale, 2022a;b). The stream consists of daily batches of user comments from December 2020 to May 2021, embedded into a 20-dimensional space via SBERT and PCA (see Appendix Q for data processing details). We focus on the critical early rollout phase, dividing the timeline into two distinct periods. Phase I serves as the pre-change calibration set (December 2, 2020 to January 30, 2021, 50 days), preceding any major vaccine-policy

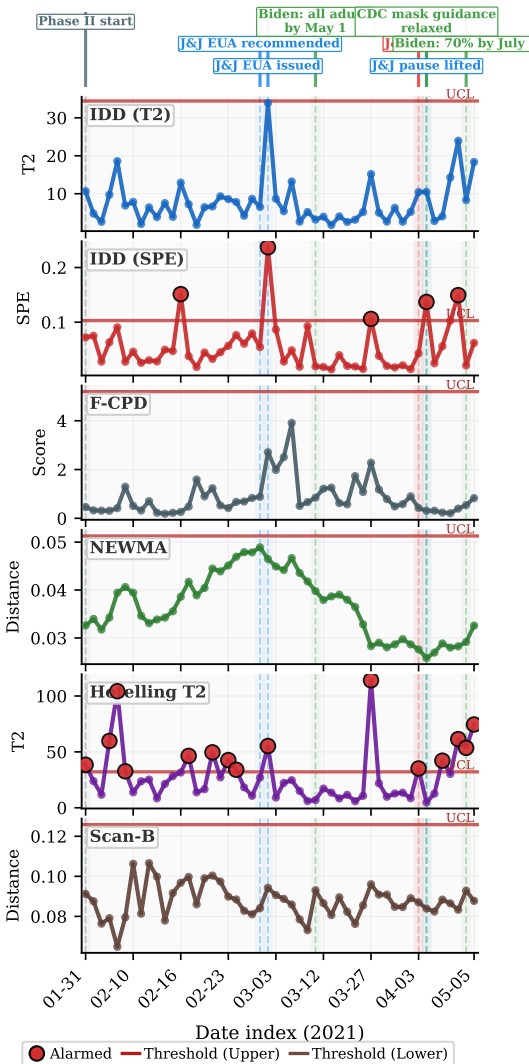

*Figure 4.* Phase II monitoring of Reddit vaccine comments (Jan 31 – May 5, 2021), calibrated on Phase I ($n_0 = 50$ days). Red lines indicate thresholds at $\alpha = 0.05$; filled circles mark alarmed days. Vertical markers denote documented vaccine-policy events. IDD (SPE) raises 5 targeted alarms aligned with the J&J EUA and pause-lifting; Hotelling $T^2$ raises 13 mostly event-unaligned alarms; F-CPD, NEWMA, and Scan-B raise zero alarms.

disruption. Phase II is the monitoring period (January 31 to May 5, 2021, 50 days), spanning the Johnson & Johnson Emergency Use Authorization and subsequent rollout events. This window contains high-impact shock events, most notably the J&J vaccine pause due to safety concerns and subsequent policy shifts.

**Results:** Lacking ground truth labels for daily sentiment shifts, we evaluate performance qualitatively by correlating detection alarms with curated vaccine-policy events. As shown in Fig. 4, IDD (SPE statistic) raises five alarms over the 50-day Phase II window, two of which align with curated

events (J&J EUA issuance on Mar 2; post-J&J-pause discourse reorganization on Apr 30), and one (May 3) precedes Biden's July-4th vaccination goal announced May 4. Despite the shifted calibration window, IDD's SPE threshold changes minimally, whereas the baselines are fragile: F-CPD's and NEWMA's thresholds shift sharply and produce zero alarms; Hotelling $T^2$ alarms on $13/50$ days, mostly event-unaligned. This demonstrates the calibration robustness of the proposed geometric statistic on high-dimensional semantic streams. Further analysis (including a Google Trends proxy correlation) is in Appendix Q.

## 5. Discussion and Conclusion

We propose a geometry-aware online CPD framework that treats streaming batch data as a stochastic process on 2-Wasserstein space. By employing a reference-centered local linearization, our approach maps empirical measures into a tangent space. We provide theoretical results establishing an $\varepsilon$-isometry guarantee for this approximation, which enables the rigorous monitoring of distributional change that Euclidean summaries obscure. Empirically, our experiments on synthetic and real-world case studies validate that IDD achieves superior detection performance compared to moment-based and model-free baselines, particularly when monitoring complex distributional shifts.

We also identify the main limitations of our work: Despite acceleration via entropic regularization and Conditional Normalizing Flows, the computational cost of OT maps remains higher than simple Euclidean statistics. This poses challenges for extremely high-frequency streams. Future work will explore incremental barycenter updates, and investigate scalable OT approximations to mitigate computational latency and estimation error in high dimensions.

## Acknowledgements

The authors would like to thank Dr. Ryan Brinkman for generously sharing the FlowCAP II dataset directly with us. We also extend our gratitude to Dr. Wenmeng Tian for her insightful discussions and valuable feedback on this work. This work was supported by the National Science Foundation [Awards CMMI-2543434, CMMI-2430998] and the American Heart Association Collaborative Science [Award 23CSA1052735].

## Impact statement

This paper presents work whose goal is to advance the field of machine learning. There are many potential societal consequences of our work, none of which we feel must be specifically highlighted here.

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

## A. Assumptions

**Lemma A.1** (Brenier map and unique $W_2$-geodesic). *Assume $\mu_0 \in \mathcal{P}_2(\mathbb{R}^d)$ is absolutely continuous and the transport cost is $c(x, y) = \|x - y\|^2$. Then for any $\mu_1 \in \mathcal{P}_2(\mathbb{R}^d)$ there exists an optimal transport map $T : \mathbb{R}^d \to \mathbb{R}^d$ pushing $\mu_0$ to $\mu_1$ (i.e., $T_{\#}\mu_0 = \mu_1$), which is $\mu_0$-a.e. unique and of the form $T = \nabla\varphi$ for a convex potential $\varphi$. Moreover, the curve*

$$\mu_s := \big((1 - s)\mathrm{Id} + sT\big)_{\#}\mu_0, \qquad s \in [0, 1],$$

*is the (unique) constant-speed $W_2$-geodesic connecting $\mu_0$ and $\mu_1$ (Brenier, 1991; Villani et al., 2008).*

## B. Derivation of Radial Isometry and $\varepsilon$-isometry

### B.1. Proof of Proposition 3.4

*Proof.* Under the quadratic cost, the squared 2-Wasserstein distance admits the Monge formulation

$$W_2^2(\bar{\mu}, \mu) \;=\; \inf_{T : T_{\#}\bar{\mu} = \mu} \int_{\mathbb{R}^d} \|x - T(x)\|^2 \, d\bar{\mu}(x).$$

Since $\bar{\mu}$ is absolutely continuous with respect to Lebesgue measure and $\mu \in \mathcal{P}_2(\mathbb{R}^d)$, Brenier's theorem guarantees the existence of an optimal transport map $T_{\bar{\mu}}^{\mu}$ that is $\bar{\mu}$-a.e. unique and satisfies $(T_{\bar{\mu}}^{\mu})_{\#}\bar{\mu} = \mu$. Evaluating the Monge objective at this optimizer, we have

$$W_2^2(\bar{\mu}, \mu) \;=\; \int_{\mathbb{R}^d} \|x - T_{\bar{\mu}}^{\mu}(x)\|^2 \, d\bar{\mu}(x) \;=\; \int_{\mathbb{R}^d} \|T_{\bar{\mu}}^{\mu}(x) - x\|^2 \, d\bar{\mu}(x).$$

Defining $v(\mu) := T_{\bar{\mu}}^{\mu} - \mathrm{Id}$ gives $W_2^2(\bar{\mu}, \mu) = \int \|v(\mu)(x)\|^2 \, d\bar{\mu}(x) = \|v(\mu)\|_{L^2(\bar{\mu})}^2$, completing the proof. $\square$

### B.2. Proof of Proposition 3.6

*Proof.* Let $(X, Y) \sim \pi$ and define $T^{\pi}(X) := \mathbb{E}[Y \mid X]$. By the conditional-variance identity,

$$\mathbb{E}\big[\|Y - T^{\pi}(X)\|^2\big] \;=\; \mathbb{E}\big[\|Y\|^2\big] - \mathbb{E}\big[\|T^{\pi}(X)\|^2\big].$$

Expanding the squared cost and using $\mathbb{E}[Y^{\top}T^{\pi}(X)] = \mathbb{E}[\mathbb{E}[Y \mid X]^{\top}T^{\pi}(X)] = \mathbb{E}[\|T^{\pi}(X)\|^2]$, we obtain

$$\begin{aligned}
\mathbb{E}\big[\|X - Y\|^2\big] &= \mathbb{E}[\|X\|^2] - 2\mathbb{E}[X^{\top}Y] + \mathbb{E}[\|Y\|^2] \\
&= \mathbb{E}[\|X\|^2] - 2\mathbb{E}[X^{\top}T^{\pi}(X)] + \mathbb{E}[\|T^{\pi}(X)\|^2] \;+\; \mathbb{E}\big[\|Y - T^{\pi}(X)\|^2\big] \\
&= \mathbb{E}\big[\|X - T^{\pi}(X)\|^2\big] \;+\; \mathbb{E}\big[\|Y - T^{\pi}(X)\|^2\big],
\end{aligned}$$

which is exactly Eq. (9) after rewriting expectations as integrals with respect to $\mu$ and $\pi$. The inequality Eq. (10) follows since the second term is nonnegative. Finally, for an optimal coupling $\pi^{\star}$, $\mathbb{E}[\|X - Y\|^2] = W_2^2(\mu, \nu)$ and equality in Eq. (10) holds if and only if $\mathbb{E}[\|Y - T^{\pi^{\star}}(X)\|^2] = 0$, i.e., $Y = T^{\pi^{\star}}(X)$ $\pi^{\star}$-a.s., equivalently $\mathrm{Var}_{\pi^{\star}}(Y \mid X) = 0$ $\mu$-a.e. $\square$

### B.3. Proof of Proposition 3.14

*Proof.* We bound the Frobenius norm of the kernel difference. By the triangle inequality:

$$\begin{aligned}
|\mathcal{K}(\boldsymbol{x}, \boldsymbol{y}) - \mathcal{K}(\boldsymbol{x}', \boldsymbol{y}')|_F &= |\mathbb{E}[V(\boldsymbol{x})V(\boldsymbol{y})^{\top} - V(\boldsymbol{x}')V(\boldsymbol{y}')^{\top}]|_F \\
&\leq \mathbb{E}|V(\boldsymbol{x})V(\boldsymbol{y})^{\top} - V(\boldsymbol{x}')V(\boldsymbol{y})^{\top}|_F + \mathbb{E}|V(\boldsymbol{x}')V(\boldsymbol{y})^{\top} - V(\boldsymbol{x}')V(\boldsymbol{y}')^{\top}|_F. \quad (20)
\end{aligned}$$

Applying the Lipschitz property $\|V(\boldsymbol{x}) - V(\boldsymbol{x}')\| \leq L\|\boldsymbol{x} - \boldsymbol{x}'\|$ (Assumption 3.13) and the Cauchy-Schwarz inequality:

$$\mathbb{E}\|(V(\boldsymbol{x}) - V(\boldsymbol{x}'))V(\boldsymbol{y})^{\top}\|_F \leq \sqrt{\mathbb{E}\|V(\boldsymbol{x}) - V(\boldsymbol{x}')\|^2}\sqrt{\mathbb{E}\|V(\boldsymbol{y})\|^2} \leq L\|\boldsymbol{x} - \boldsymbol{x}'\|B_2,$$

where $B_2 := \sup_{\boldsymbol{z} \in \mathcal{X}}(\mathbb{E}\|V(\boldsymbol{z})\|^2)^{1/2}$ is the maximum root-mean-square norm. Applying the same bound to the second term yields:

$$\|\mathcal{K}(\boldsymbol{x}, \boldsymbol{y}) - \mathcal{K}(\boldsymbol{x}', \boldsymbol{y}')\|_F \leq LB_2\|\boldsymbol{x} - \boldsymbol{x}'\| + LB_2\|\boldsymbol{y} - \boldsymbol{y}'\|.$$

Thus, $\mathcal{K}$ is Lipschitz with constant $C_{\mathcal{K}} = LB_2$. $\square$

## B.4. Proof of Theorem 3.15

*Proof.* The proof proceeds in three steps, generalizing the 1D results of Reade (1983). First, fix $M \in \mathbb{N}$ and set $N = M^d$. Partition the domain $\mathcal{X}$ into $N$ measurable sets $\{\mathcal{X}_j\}_{j=1}^N$ with diameter proportional to $M^{-1} \asymp N^{-1/d}$. Let $R_N$ be the kernel of the orthogonal projector onto piecewise constant functions on this partition. Second, we derive the trace bound. Let $S_N$ be the rank-$N$ approximation of the kernel operator $\Gamma$ induced by this projection. Since $\Gamma$ is positive definite, the tail sum of eigenvalues is bounded by the trace of the residual operator:

$$\sum_{m>N} \lambda_m \leq \mathrm{tr}(\Gamma - S_N) = \int_{\mathcal{X}} (\mathcal{K}(\boldsymbol{x}, \boldsymbol{x}) - S_N(\boldsymbol{x}, \boldsymbol{x})) d\boldsymbol{x}.$$

Third, we utilkize the Lipschitz condition for the estimation. Using the Lipschitz continuity of $\mathcal{K}$ (Proposition 3.14), the pointwise approximation error of the piecewise constant projector is bounded by the Lipschitz constant $C_{\mathcal{K}}$ times the diameter of the partition cells:

$$|\mathcal{K}(\boldsymbol{x}, \boldsymbol{x}) - S_N(\boldsymbol{x}, \boldsymbol{x})| \lesssim C_{\mathcal{K}} \cdot \mathrm{diam}(\mathcal{X}_j) \asymp C_{\mathcal{K}} N^{-1/d}.$$

Integrating this error over the bounded domain $\mathcal{X}$ yields the tail bound:

$$\sum_{m>N} \lambda_m \leq A_{\mathcal{X}} C_{\mathcal{K}} N^{-1/d}.$$

Setting the truncation level $K = N$ completes the proof. □

# C. Derivation of Asymptotic Distributions

*Proof of Proposition 3.9.* Let $V$ denote the generic random field generating the pre-change process, with population covariance $\Gamma$ and eigenpairs $(\lambda_m, \phi_m)$. By the Karhunen-Loève theorem, the random field $V$ admits the expansion $V = \sum_{m=1}^{\infty} \xi_m \phi_m$, where the scores $\xi_m = \langle V, \phi_m \rangle_H$ are independent Gaussian variables $\xi_m \sim \mathcal{N}(0, \lambda_m)$. Consider the empirical covariance $\widehat{\mathcal{C}}$ estimated from $n_0$ samples, with eigenpairs $(\widehat{\lambda}_m, \widehat{\phi}_m)$. For a new independent observation $V^\star$, the computed score on the $m$-th component is $\widehat{\xi}_m = \langle V^\star, \widehat{\phi}_m \rangle_H$. Under the conditions of Hall & Hosseini-Nasab (2006), as $n_0 \to \infty$, we have consistent estimation:

$$\|\widehat{\phi}_m - \phi_m\|_H = O_p(n_0^{-1/2}) \quad \text{and} \quad |\widehat{\lambda}_m - \lambda_m| = O_p(n_0^{-1/2}).$$

Consequently, $\widehat{\xi}_m^2/\widehat{\lambda}_m \xrightarrow{p} \xi_m^2/\lambda_m$. Since $\xi_m/\sqrt{\lambda_m} \sim \mathcal{N}(0,1)$, it follows that $\xi_m^2/\lambda_m \sim \chi_1^2$. Summing over $m = 1, \ldots, K$ yields:

$$T_{t,K}^2 = \sum_{m=1}^{K} \frac{\widehat{\xi}_m^2}{\widehat{\lambda}_m} \xrightarrow{d} \sum_{m=1}^{K} Z_m^2 \sim \chi_K^2,$$

which concludes the proof. □

# D. Details on Threshold Calibration and ARL Control

In this appendix, we provide the formal justification for the empirical quantile calibration strategy used in the main text. We show that controlling the marginal false alarm rates of the monitoring statistics via order statistics is sufficient to lower-bound the global $\mathrm{ARL}_0$.

**Theorem D.1** (Sequential false-alarm control under fixed thresholds). *Fix truncation $K$ and thresholds $h_{T^2}, h_{\mathrm{SPE}}$. Under the no-change law $\mathbb{P}_\infty$, assume the monitoring-phase pairs $\{(T_{t,K}^2, \mathrm{SPE}_t)\}_{t \geq n_0+1}$ are i.i.d. and independent of the calibration phase segment used to construct $(h_{T^2}, h_{\mathrm{SPE}})$. Let the stopping time*

$$\tau := \inf\left\{t \geq n_0 + 1 : T_{t,K}^2 > h_{T^2} \text{ or } \mathrm{SPE}_t > h_{\mathrm{SPE}}\right\}.$$

*Then $\tau - (n_0 + 1)$ has a geometric distribution with success probability (e.g., the classical run-length calculation for Shewhart-type charts; Shewhart, 1931; Montgomery, 2020)*

$$p_\infty := \mathbb{P}_\infty\left(T_{1,K}^2 > h_{T^2} \text{ or } \mathrm{SPE}_1 > h_{\mathrm{SPE}}\right),$$

*and hence* $\mathrm{ARL}_0 = \mathbb{E}_\infty[\tau] = n_0 + 1 + 1/p_\infty$. *Moreover, if* $\mathbb{P}_\infty(T_{1,K}^2 > h_{T^2}) \le \alpha_{T^2}$ *and* $\mathbb{P}_\infty(\mathrm{SPE}_1 > h_{\mathrm{SPE}}) \le \alpha_{\mathrm{SPE}}$, *then by the union bound* $p_\infty \le \alpha_{T^2} + \alpha_{\mathrm{SPE}}$ *and therefore*

$$\mathrm{ARL}_0 \ \ge \ n_0 + 1 + \frac{1}{\alpha_{T^2} + \alpha_{\mathrm{SPE}}}.$$

**Corollary D.2** (Empirical-quantile calibration implies ARL control)**.** *Assume the setting of Theorem 3.10. Suppose that under* $\mathbb{P}_\infty$ *the marginal distributions of* $T_{t,K}^2$ *and* $\mathrm{SPE}_t$ *are continuous, and let* $\{T_{t,K}^2\}_{t=1}^{n_0}$ *and* $\{\mathrm{SPE}_t\}_{t=1}^{n_0}$ *be the samples in the calibration phase. Define thresholds as order statistics*

$$h_{T^2} := T_{(k_{T^2})}^2, \qquad h_{\mathrm{SPE}} := \mathrm{SPE}_{(k_{\mathrm{SPE}})},$$

*where* $T_{(1)}^2 \le \cdots \le T_{(n_0)}^2$ *and* $\mathrm{SPE}_{(1)} \le \cdots \le \mathrm{SPE}_{(n_0)}$, *with* $k_{T^2} = \lceil (1 - \alpha_{T^2})n_0 \rceil$ *and* $k_{\mathrm{SPE}} = \lceil (1 - \alpha_{\mathrm{SPE}})n_0 \rceil$. *Then, for an independent monitoring phase observation,*

$$\mathbb{P}_\infty(T_{1,K}^2 > h_{T^2}) = \frac{n_0 + 1 - k_{T^2}}{n_0 + 1} \le \alpha_{T^2} + \frac{1}{n_0 + 1},$$

$$\mathbb{P}_\infty(\mathrm{SPE}_1 > h_{\mathrm{SPE}}) = \frac{n_0 + 1 - k_{\mathrm{SPE}}}{n_0 + 1} \le \alpha_{\mathrm{SPE}} + \frac{1}{n_0 + 1}.$$

*Consequently,*

$$p_\infty \le \alpha_{T^2} + \alpha_{\mathrm{SPE}} + \frac{2}{n_0 + 1},$$

$$\mathrm{ARL}_0 \ \ge \ n_0 + 1 + \frac{1}{\alpha_{T^2} + \alpha_{\mathrm{SPE}} + 2/(n_0 + 1)}.$$

# E. Construction of Tangent Space Detectors

**Pre-change reference segment and covariance operator.**    In online CPD, we assume access to a *pre-change* reference segment of length $n_0$ (used only for calibration), which provides tangent fields $\{v_t\}_{t=1}^{n_0}$ in the common Hilbert space $H := L^2(\bar\mu; \mathbb{R}^d)$ with inner product $\langle f, g \rangle_H := \int_{\mathcal{X}} f(x)^\top g(x) \, d\bar\mu(x)$. Define the sample mean and centered fields

$$\bar v := \frac{1}{n_0} \sum_{t=1}^{n_0} v_t, \qquad \tilde v_t := v_t - \bar v.$$

The empirical covariance operator $\widehat{C} : H \to H$ is

$$(\widehat{C}f)(\cdot) \ = \ \frac{1}{n_0 - 1} \sum_{t=1}^{n_0} \langle f, \tilde v_t \rangle_H \, \tilde v_t(\cdot) \ = \ \frac{1}{n_0 - 1} \sum_{t=1}^{n_0} (\tilde v_t \otimes \tilde v_t) f, \tag{21}$$

where $(a \otimes b)f := \langle f, b \rangle_H \, a$. Then $\widehat{C}$ is self-adjoint and positive semidefinite, and $\mathrm{rank}(\widehat{C}) \le n_0 - 1$.

**Hotelling-$T^2$ statistic via the Moore-Penrose pseudoinverse.**    Because $\widehat{C}$ is finite-rank on the infinite-dimensional space $H$, it is not invertible. We therefore use the Moore–Penrose pseudoinverse $\widehat{C}^\dagger : H \to H$. Let $\mathrm{ran}(\widehat{C})$ and $\ker(\widehat{C})$ denote the range and kernel of $\widehat{C}$. Since $\widehat{C}$ is self-adjoint, $H$ decomposes as

$$H \ = \ \mathrm{ran}(\widehat{C}) \ \oplus \ \ker(\widehat{C}), \qquad \ker(\widehat{C}) = \mathrm{ran}(\widehat{C})^\perp.$$

The pseudoinverse $\widehat{C}^\dagger$ is the unique self-adjoint operator satisfying

$$\widehat{C} \, \widehat{C}^\dagger \, \widehat{C} = \widehat{C}, \qquad \widehat{C}^\dagger \, \widehat{C} \, \widehat{C}^\dagger = \widehat{C}^\dagger, \qquad (\widehat{C} \, \widehat{C}^\dagger)^* = \widehat{C} \, \widehat{C}^\dagger, \qquad (\widehat{C}^\dagger \, \widehat{C})^* = \widehat{C}^\dagger \, \widehat{C}.$$

For an incoming (possibly post-change) tangent field $v_t \in H$, define the centered field

$$\Delta_t := v_t - \bar v \in H.$$

We define the functional Hotelling's $T^2$ statistic-based detector

$$T_t^2 \ := \ \left\langle \widehat{C}^\dagger \Delta_t, \, \Delta_t \right\rangle_H \ = \ \left\| (\widehat{C}^\dagger)^{1/2} \Delta_t \right\|_H^2, \tag{22}$$

which measures deviation along the dominant pre-change variability directions encoded by $\widehat{C}$.

**Spectral form of the pseudoinverse and the computable $T^2$ statistic.** Let $\{(\widehat{\lambda}_m, \widehat{\phi}_m)\}_{m \geq 1}$ be the eigenpairs of $\widehat{C}$, with $\widehat{\lambda}_1 \geq \widehat{\lambda}_2 \geq \cdots \geq 0$ and $\{\widehat{\phi}_m\}$ orthonormal in $H$. Since $\widehat{C}$ has rank at most $n_0 - 1$, we have $\widehat{\lambda}_m = 0$ for all $m \geq n_0$. On $\mathrm{ran}(\widehat{C})$, the pseudoinverse acts by inverting the nonzero eigenvalues, hence

$$\widehat{C}^\dagger = \sum_{m:\, \widehat{\lambda}_m > 0} \widehat{\lambda}_m^{-1} (\widehat{\phi}_m \otimes \widehat{\phi}_m), \tag{23}$$

with $\widehat{C}^\dagger f = 0$ for $f \in \ker(\widehat{C})$. Define the empirical scores for $\Delta_t$,

$$\widehat{\xi}_{tm} := \langle \Delta_t, \widehat{\phi}_m \rangle_H.$$

Substituting Eq. (23) into Eq. (22) yields

$$T_t^2 = \sum_{m:\, \widehat{\lambda}_m > 0} \frac{\widehat{\xi}_{tm}^2}{\widehat{\lambda}_m}.$$

In practice we retain the leading $K$ components (regularization / truncation), obtaining

$$T_{t,K}^2 := \sum_{m=1}^{K} \frac{\widehat{\xi}_{tm}^2}{\widehat{\lambda}_m}. \tag{24}$$

**Sequential false-alarm control under i.i.d. monitoring statistics**

*Proof of Theorem 3.10.* Let $A_t := \{T_{t,K}^2 > h_{T^2} \text{ or } \mathrm{SPE}_t > h_{\mathrm{SPE}}\}$ for $t \geq n_0 + 1$. Under $\mathbb{P}_\infty$, by the assumed i.i.d. structure and the fact that thresholds are fixed in monitoring phase, the indicators $\{\mathbf{1}(A_t)\}_{t \geq n_0 + 1}$ are i.i.d. Bernoulli with success probability $p_\infty = \mathbb{P}_\infty(A_{n_0+1})$. Therefore, $\tau - (n_0 + 1)$ is geometric with mean $1/p_\infty$, giving $\mathbb{E}_\infty[\tau] = n_0 + 1 + 1/p_\infty$. The union-bound inequality follows from $p_\infty = \mathbb{P}_\infty(A_{n_0+1}) \leq \mathbb{P}_\infty(T_{1,K}^2 > h_{T^2}) + \mathbb{P}_\infty(\mathrm{SPE}_1 > h_{\mathrm{SPE}}) \leq \alpha_{T^2} + \alpha_{\mathrm{SPE}}$. $\qquad\square$

*Proof of Corollary 3.11.* We prove the statement for $T_{t,K}^2$; the argument for $\mathrm{SPE}_t$ is identical. Let $Z_1, \ldots, Z_{n_0}$ be the sample of calibration phase and let $Z^\star$ be an independent monitoring phase draw, all i.i.d. from a continuous distribution. Let $Z_{(1)} \leq \cdots \leq Z_{(n_0)}$ denote the order statistics and set $h := Z_{(k)}$. By exchangeability of $(Z_1, \ldots, Z_{n_0}, Z^\star)$ and continuity (no ties a.s.), the rank of $Z^\star$ among these $n_0 + 1$ values is uniform on $\{1, \ldots, n_0 + 1\}$. Therefore,

$$\mathbb{P}(Z^\star > h) = \mathbb{P}(\mathrm{rank}(Z^\star) \geq k + 1) = \frac{n_0 + 1 - k}{n_0 + 1}.$$

With $k = \lceil (1 - \alpha) n_0 \rceil$, we have $n_0 - k \leq \alpha n_0$, hence $(n_0 + 1 - k)/(n_0 + 1) \leq \alpha + 1/(n_0 + 1)$. Applying this bound to both charts and using the union bound gives $p_\infty \leq \alpha_{T^2} + \alpha_{\mathrm{SPE}} + 2/(n_0 + 1)$. The ARL bound follows from Theorem 3.10. $\quad\square$

**Residual-energy statistic (SPE/Q).** Let $\widehat{P}_K$ be the orthogonal projector onto $\mathrm{span}\{\widehat{\phi}_1, \ldots, \widehat{\phi}_K\}$. The residual-energy detector is

$$\mathrm{SPE}_t := \big\| (I - \widehat{P}_K) \Delta_t \big\|_H^2 = \sum_{m > K} \widehat{\xi}_{tm}^2. \tag{25}$$

The pair $\{T_{t,K}^2, \mathrm{SPE}_t\}$ forms a two-statistic online CPD rule: an alarm is raised when either statistic exceeds its pre-change calibrated threshold.

# F. Theoretical Analysis of Gaussian Translation

In this appendix, we analyze the theoretical signal-to-noise ratio (effect size) of the proposed geometric framework compared to the functional log-density baseline. We consider a controlled Gaussian mean-shift scenario to derive explicit expressions for the displacement signal in both the Optimal Transport tangent space and the $L^2$ log-density space.

**Theorem F.1** (Effect Size Comparison). *Consider a reference distribution $\bar{\mu} = \mathcal{N}(m, \Sigma)$ and a perturbed distribution $\mu_{\Delta_n} = \mathcal{N}(m + \Delta_n, \Sigma)$ representing a local mean-shift with $\Delta_n = \delta/\sqrt{n}$. As $n \to \infty$, the squared norms of the shift in the OT tangent space ($H_{\mathrm{OT}}$) and the log-density space ($H_{\log}$) satisfy:*

$$\|u_{\Delta_n}\|_{H_{\mathrm{OT}}}^2 = \frac{1}{n}\|\delta\|^2, \tag{26}$$

$$\|g_{\Delta_n} - g_0\|_{H_{\log}}^2 = \frac{1}{n}\delta^\top \Sigma^{-1}\delta + O\left(\frac{1}{n^2}\right). \tag{27}$$

*Consequently, the ratio of the effect sizes is determined by the covariance structure:*

$$\frac{\|g_{\Delta_n} - g_0\|_{H_{\log}}^2}{\|u_{\Delta_n}\|_{H_{\mathrm{OT}}}^2} = \frac{\delta^\top \Sigma^{-1}\delta}{\|\delta\|^2} + O\left(\frac{1}{n}\right). \tag{28}$$

*Proof.* We first derive the norm in the Optimal Transport tangent space. For a pure Gaussian translation from $\mathcal{N}(m, \Sigma)$ to $\mathcal{N}(m + \Delta, \Sigma)$, the unique optimal transport map is the identity shift $T(x) = x + \Delta$. The corresponding tangent vector field is the constant map $u_\Delta(x) \equiv \Delta$. The norm in the tangent space $H_{\mathrm{OT}} = L^2(\bar{\mu}; \mathbb{R}^d)$ is calculated directly by integrating the squared Euclidean magnitude of the displacement against the reference measure:

$$\|u_\Delta\|_{H_{\mathrm{OT}}}^2 = \int_{\mathbb{R}^d} \|u_\Delta(x)\|^2 \, d\bar{\mu}(x) = \|\Delta\|^2.$$

Substituting the local alternative $\Delta_n = \delta/\sqrt{n}$, we obtain $\|u_{\Delta_n}\|_{H_{\mathrm{OT}}}^2 = \frac{1}{n}\|\delta\|^2$. Notably, this norm depends strictly on the Euclidean magnitude of the shift and is independent of the distribution's covariance $\Sigma$.

Next, we analyze the norm in the log-density space $H_{\log} = L^2(\bar{\mu}; \mathbb{R})$. Let $g_0$ and $g_\Delta$ denote the log-densities of the reference $\bar{\mu}$ and the shifted measure $\mu_\Delta$, respectively. Defining the centered random variable $Y = X - m$, the difference in log-densities simplifies to:

$$g_\Delta(X) - g_0(X) = (\Sigma^{-1}\Delta)^\top Y - \frac{1}{2}\Delta^\top \Sigma^{-1}\Delta.$$

The squared norm in $H_{\log}$ corresponds to the second moment of this difference under the reference measure $\bar{\mu}$. Letting $a = \Sigma^{-1}\Delta$ and $c = \frac{1}{2}\Delta^\top \Sigma^{-1}\Delta$, and noting that $Y \sim \mathcal{N}(0, \Sigma)$, we have $\mathbb{E}[a^\top Y] = 0$ and $\mathrm{Var}(a^\top Y) = a^\top \Sigma a$. Thus:

$$\begin{aligned}
\|g_\Delta - g_0\|_{H_{\log}}^2 &= \mathrm{Var}(a^\top Y) + (\mathbb{E}[a^\top Y - c])^2 \\
&= a^\top \Sigma a + c^2 \\
&= (\Delta^\top \Sigma^{-1})\Sigma(\Sigma^{-1}\Delta) + \left(\frac{1}{2}\Delta^\top \Sigma^{-1}\Delta\right)^2 \\
&= \Delta^\top \Sigma^{-1}\Delta + O(\|\Delta\|^4).
\end{aligned}$$

Substituting $\Delta_n = \delta/\sqrt{n}$, the leading term is $\frac{1}{n}\delta^\top \Sigma^{-1}\delta$, which yields Eq. equation 27. $\square$

*Remark* F.2 (Effect of KDE Smoothing). The result above highlights a critical practical difference. The Log-KDE baseline typically estimates densities using a kernel bandwidth matrix $H$, which inflates the effective covariance to $\Sigma_{\mathrm{eff}} \approx \Sigma + H$. Substituting $\Sigma_{\mathrm{eff}}$ into Eq. equation 27 shows that the signal strength in the log-density space is attenuated by smoothing:

$$\delta^\top (\Sigma + H)^{-1}\delta \quad < \quad \delta^\top \Sigma^{-1}\delta.$$

In high-variance or high-dimensional regimes where large bandwidths are required for stability, this attenuation is significant. In contrast, the OT signal (Eq. equation 26) remains $\|\delta\|^2/n$, invariant to both the intrinsic covariance $\Sigma$ and any smoothing parameters.

### F.1. Additional Empirical Results: Gaussian Translation

Table 2 presents the detailed $\mathrm{ARL}_1$ comparison for Gaussian mean shifts across dimensions $d \in \{1, 2, 5\}$, variances $\sigma$, and shift magnitudes $\delta$. IDD consistently outperforms Log-KDE, particularly in high-variance regimes ($\sigma = 2.0$) where Log-KDE requires larger bandwidths to stabilize.

*Table 2.* Gaussian translation detection delay (ARL$_1$) at matched ARL$_0$. The Ours column is shaded; best (smallest) per row is bolded.

| $d$ | $\sigma$ | $\delta_1$ | Hotelling $T^2$ | Ours | log–KDE ($h$=0.5) | log–KDE ($h$=1) | log–KDE ($h$=1.5) | log–KDE ($h$=auto) |
|---|---|---|---|---|---|---|---|---|
| 1 | 0.5 | 0.1 | **1.8 ± 0.2 (↑28.0%)** | **1.8 ± 0.2 (↑28.0%)** | 4.7 ± 2.2 | 2.5 ± 0.6 | 2.6 ± 0.5 | 18.3 ± 6.8 |
| 1 | 0.5 | 0.5 | **1.0 ± 0.0 (↑95.1%)** | **1.0 ± 0.0 (↑95.1%)** | **1.0 ± 0.0 (↑95.1%)** | **1.0 ± 0.0 (↑95.1%)** | **1.0 ± 0.0 (↑95.1%)** | **1.0 ± 0.0 (↑95.1%)** |
| 1 | 1.0 | 0.1 | **4.2 ± 1.1 (↑20.8%)** | 5.3 ± 1.6 | 10.8 ± 2.5 | 19.0 ± 10.8 | 17.4 ± 9.7 | 21.5 ± 6.8 |
| 1 | 1.0 | 0.5 | **1.0 ± 0.0 (↑9.1%)** | **1.0 ± 0.0 (↑9.1%)** | **1.0 ± 0.0 (↑9.1%)** | 1.1 ± 0.1 | 1.1 ± 0.1 | 2.0 ± 0.7 |
| 1 | 2.0 | 0.1 | 16.7 ± 3.5 | 17.0 ± 3.4 | **16.4 ± 2.6** | 21.5 ± 6.9 | 19.9 ± 4.8 | 17.1 ± 2.4 |
| 1 | 2.0 | 0.5 | 1.6 ± 0.3 | **1.5 ± 0.2 (↑6.3%)** | 7.8 ± 3.4 | 6.6 ± 3.6 | 3.8 ± 1.3 | 12.9 ± 4.0 |
| 2 | 0.5 | 0.1 | 1.5 ± 0.2 | **1.3 ± 0.2 (↑13.3%)** | 2.3 ± 0.7 | 5.6 ± 3.5 | 9.4 ± 7.6 | 12.3 ± 3.0 |
| 2 | 0.5 | 0.5 | **1.0 ± 0.0 (↑96.4%)** | **1.0 ± 0.0 (↑96.4%)** | **1.0 ± 0.0 (↑96.4%)** | **1.0 ± 0.0 (↑96.4%)** | **1.0 ± 0.0 (↑96.4%)** | **1.0 ± 0.0 (↑96.4%)** |
| 2 | 1.0 | 0.1 | 2.7 ± 0.7 | **2.4 ± 0.6 (↑11.1%)** | 32.5 ± 19.3 | 5.7 ± 1.6 | 6.3 ± 1.6 | 16.1 ± 3.9 |
| 2 | 1.0 | 0.5 | **1.0 ± 0.0 (↑9.1%)** | **1.0 ± 0.0 (↑9.1%)** | 1.1 ± 0.1 | **1.0 ± 0.0 (↑9.1%)** | **1.0 ± 0.0 (↑9.1%)** | 1.5 ± 0.3 |
| 2 | 2.0 | 0.1 | 11.3 ± 5.4 | **10.2 ± 5.3 (↑9.7%)** | 16.9 ± 5.3 | 18.9 ± 7.9 | 23.4 ± 10.0 | 15.4 ± 5.2 |
| 2 | 2.0 | 0.5 | 1.2 ± 0.1 | **1.0 ± 0.0 (↑16.7%)** | 6.9 ± 1.5 | 3.3 ± 0.7 | 2.6 ± 0.6 | 7.2 ± 1.5 |
| 5 | 0.5 | 0.1 | **1.0 ± 0.0 (↑16.7%)** | 1.9 ± 0.5 | 6.0 ± 1.5 | 1.2 ± 0.2 | 1.7 ± 0.4 | 12.6 ± 3.5 |
| 5 | 0.5 | 0.5 | **1.0 ± 0.0 (↑96.3%)** | **1.0 ± 0.0 (↑96.3%)** | **1.0 ± 0.0 (↑96.3%)** | **1.0 ± 0.0 (↑96.3%)** | **1.0 ± 0.0 (↑96.3%)** | **1.0 ± 0.0 (↑96.3%)** |
| 5 | 1.0 | 0.1 | **2.8 ± 0.7 (↑42.9%)** | 4.9 ± 1.7 | 14.7 ± 4.3 | 11.9 ± 4.3 | 9.4 ± 2.3 | 11.1 ± 3.2 |
| 5 | 1.0 | 0.5 | **1.0 ± 0.0 (↑93.2%)** | **1.0 ± 0.0 (↑93.2%)** | **1.0 ± 0.0 (↑93.2%)** | **1.0 ± 0.0 (↑93.2%)** | **1.0 ± 0.0 (↑93.2%)** | **1.0 ± 0.0 (↑93.2%)** |
| 5 | 2.0 | 0.1 | **6.9 ± 1.9 (↑37.3%)** | 11.0 ± 3.2 | 26.5 ± 4.2 | 22.1 ± 4.7 | 19.6 ± 4.4 | 16.0 ± 4.0 |
| 5 | 2.0 | 0.5 | **1.0 ± 0.0 (↑9.1%)** | 1.1 ± 0.1 | 21.7 ± 5.8 | 4.4 ± 1.4 | 4.0 ± 0.8 | 8.8 ± 2.7 |

# G. Details on Synthetic Continuous Stream Generation

This appendix provides the full specification of the synthetic distribution-valued streams used in Section 4.1. We detail: (i) the pre-change reference distribution, (ii) the family of convex deformations used to generate stochastic variability within a stationary regime, (iii) the post-change scenarios, and (iv) a theoretical verification that the generated maps satisfy cyclic monotonicity, thereby coinciding with the quadratic-cost optimal transport (OT) maps.

## G.1. Pre-change Reference Distribution

Let $\mathcal{X} \subset \mathbb{R}^d$ be a compact convex domain (in our experiments, $\mathcal{X} = [0, 1]^d$). We first sample a *reference distribution* $\bar{\mu} \in \mathcal{P}_2(\mathcal{X})$ from a multimodal parametric family. From this distribution, we draw a fixed *base sample* $X_{\text{base}}^{(0)} = \{x_i\}_{i=1}^{N} \overset{\text{i.i.d.}}{\sim} \bar{\mu}$. We utilize a $K$-component mixture of product-Beta distributions, parameterized by mixture weights $\pi \in \Delta^{K-1}$ (where $\Delta^{K-1}$ denotes the simplex) and component parameters $\theta_k$. For the experiments, we set $K = 4$.

## G.2. Stochastic Variability via Convex Deformations

To model benign variability within a stationary regime, at each time step $t$, we generate a random deformation map $T_t : \mathcal{X} \to \mathbb{R}^d$. The observed batch data is generated as the pushforward of the base sample: $X_t = T_t(X_{\text{base}})$. The resulting empirical measure is:

$$\mu_t = \frac{1}{N} \sum_{i=1}^{N} \delta_{(X_t)_i}.$$

**Convex Potential and Transport Map.** To ensure the deformation corresponds to an optimal transport map, we construct $T_t$ as the gradient of a strictly convex potential $\Phi_t$. We define this potential as a perturbation of the identity potential $\frac{1}{2}\|x\|^2$. Fix integers $J \geq 1$ and a smoothness parameter $\beta > 0$. At each time $t$, we draw random directions $a_{t,j} \in \mathbb{S}^{d-1}$, weights $w_{t,j} > 0$ (normalized such that $\sum_j w_{t,j} = 1$), and offsets $c_{t,j} \in \mathbb{R}$. Let $\zeta_\beta(z) = \beta^{-1}\log(1 + e^{\beta z})$ be the softplus function and $\sigma_\beta(z) = (1 + e^{-\beta z})^{-1}$ be the sigmoid function. We define a convex perturbation potential $\psi_t$ as:

$$\psi_t(x) := \frac{\varepsilon_t}{2} \sum_{j=1}^{J} w_{t,j} \left(\zeta_\beta(\langle a_{t,j}, x \rangle - c_{t,j})\right)^2.$$

The full transport potential is $\Phi_t(x) := \frac{1}{2}\|x\|^2 + \psi_t(x)$. The resulting transport map is:

$$T_t(x) = \nabla\Phi_t(x) = x + \nabla\psi_t(x). \tag{29}$$

Computing the gradient of $\psi_t$, the explicit form of the map is:

$$T_t(x) \;=\; x \;+\; \varepsilon_t \sum_{j=1}^{J} w_{t,j}\, h_\beta(\langle a_{t,j}, x \rangle - c_{t,j})\, a_{t,j},$$

where $h_\beta(z) := \zeta_\beta(z)\, \sigma_\beta(z)$. The parameter $\varepsilon_t > 0$ controls the magnitude of the deformation. For the pre-change process, we set $\varepsilon_t = 0.3$.

## G.3. Verification: Cyclic Monotonicity and OT Optimality

The construction in equation 29 ensures that $T_t$ is the gradient of a convex function. This property implies cyclic monotonicity, ensuring that if the reference law is absolutely continuous, $T_t$ coincides with the unique quadratic-cost OT map from $\bar\mu$ to its pushforward $\mu_t$.

**Lemma G.1** (Convexity of the Potential). *For any t, the perturbation potential $\psi_t$ is convex on $\mathcal{X}$. Consequently, the total potential $\Phi_t(x) = \frac{1}{2}\|x\|^2 + \psi_t(x)$ is strictly convex.*

*Proof.* The function $z \mapsto \zeta_\beta(z)$ is convex and non-decreasing. The function $y \mapsto y^2$ is convex and non-decreasing for $y \geq 0$. Since $\zeta_\beta(z) > 0$, the composition $z \mapsto (\zeta_\beta(z))^2$ is convex. The term $x \mapsto \langle a_{t,j}, x \rangle - c_{t,j}$ is affine. Since the composition of a convex function with an affine map preserves convexity, and a non-negative weighted sum of convex functions is convex, $\psi_t$ is convex. Adding the strictly convex term $\frac{1}{2}\|x\|^2$ ensures $\Phi_t$ is strictly convex. $\qquad\square$

**Lemma G.2** (Cyclic Monotonicity). *Let $\Phi : \mathcal{X} \to \mathbb{R}$ be convex and differentiable, and let $T = \nabla\Phi$. Then $T$ is cyclically monotone: for any finite sequence $x_1, \ldots, x_m \in \mathcal{X}$ with $x_{m+1} = x_1$,*

$$\sum_{i=1}^{m} \langle x_i, T(x_i) \rangle \;\geq\; \sum_{i=1}^{m} \langle x_i, T(x_{i+1}) \rangle.$$

*Proof.* By the first-order convexity condition, for any $x, y$, we have $\Phi(y) \geq \Phi(x) + \langle \nabla\Phi(x), y - x \rangle$. Rearranging terms yields $\langle \nabla\Phi(x), x \rangle - \langle \nabla\Phi(x), y \rangle \geq \Phi(x) - \Phi(y)$. Applying this inequality to pairs $(x_i, x_{i+1})$ and summing over $i = 1, \ldots, m$, the right-hand side telescopes to 0, yielding the result. $\qquad\square$

**Proposition G.3** (Optimality of the Constructed Map). *Assume $\bar\mu$ is absolutely continuous with respect to the Lebesgue measure on $\mathcal{X}$. Consider the quadratic cost $c(x, y) = \|x - y\|^2$. Let $\mu_t := (T_t)_\# \bar\mu$, where $T_t = \nabla\Phi_t$ with $\Phi_t$ convex. Then, $T_t$ is the $\bar\mu$-almost everywhere unique optimal transport map from $\bar\mu$ to $\mu_t$.*

*Proof.* By Lemma G.1, $T_t$ is the gradient of a convex function. By Brenier's Theorem, the unique quadratic-cost optimal transport map from an absolutely continuous measure $\bar\mu$ to any target measure $\mu_t$ is characterized as the gradient of a convex function. Since $T_t$ satisfies this characterization and pushes $\bar\mu$ to $\mu_t$ by construction, it is the optimal map. $\qquad\square$

## G.4. Post-Change Scenarios

We simulate a single change-point at time $\kappa$. For $t \leq \kappa$ (pre-change), batches are generated using the reference base sample $X_{\text{base}}^{(0)}$ and random deformations $T_t$ as described above. For $t > \kappa$ (post-change), we modify the *base generator* to induce a distributional shift, while maintaining the same deformation mechanism.

**(S1) Barycenter Change.** We generate a perturbed reference distribution $\bar\mu^{(1)}$ by perturbing the component parameters of the original family with strength $\delta_{\text{loc}}$. We draw a new base sample $X_{\text{base}}^{(1)} \overset{\text{i.i.d.}}{\sim} \bar\mu^{(1)}$ once. For all $t > \kappa$, observations are generated as $X_t = T_t(X_{\text{base}}^{(1)})$. This scenario simulates a shift in the central tendency (mean/mode) of the stream.

**(S2) Multimodal Reweighting.** Let the reference $\bar\mu$ be a mixture with weights $\pi \in \Delta^{K-1}$. We perturb these weights to obtain $\pi^{(1)}$, where $\pi_k^{(1)} \propto \pi_k \eta_k$, with $\eta_k$ drawn from a distribution controlled by perturbation strength $\delta_{\text{mm}}$. We keep the component parameters fixed and sample $X_{\text{base}}^{(1)} \sim \bar\mu^{(1)}$ using the new weights. Post-change batches are generated as $X_t = T_t(X_{\text{base}}^{(1)})$. This scenario isolates changes in mass allocation across modes, which are often difficult to detect using low-order summary statistics.

**(S3) Copula Shift.** We modify the cross-dimensional dependence structure while approximately preserving the marginal distributions. Starting from the initial base sample $X_{\text{base}}^{(0)}$, we apply an Iman-Conover rank-preserving reordering to induce a target correlation $\rho$, resulting in a modified base sample $X_{\text{base}}^{(1)}$. Post-change batches are generated as $X_t = T_t(X_{\text{base}}^{(1)})$. This scenario targets dependence shifts that may be invisible to methods that monitor marginals independently.

# H. Details on Synthetic Discrete Streams

This appendix details the generation processes for discrete distribution-valued streams and the specific formulations of the baseline detectors used for comparison.

## H.1. Data Generation Processes

**Stream of Poisson Counts.** We simulate monitoring scenarios involving count data, where at each time step $t$, we observe a batch of $N$ count-valued samples denoted as $X_t = \{x_{t,i}\}_{i=1}^N$ with $x_{t,i} \in \mathbb{Z}_{\geq 0}$. In the pre-change regime, samples are drawn independently from a standard Poisson distribution with rate $\lambda_0$. To simulate the emergence of rare, extreme events in the post-change phase, we model the distribution as a mixture of the background process and a Dirac mass centered at a high value $k^*$. Specifically, for a mixing proportion $\alpha \in (0, 1)$, the post-change samples follow the mixture law $(1 - \alpha)\operatorname{Pois}(\lambda_0) + \alpha\,\delta_{k^*}$. We also considered a heavy-tail mixture scenario, discussed in the additional results, where samples are drawn from a mixture of two Poisson sources with distinct rates, optionally calibrated such that the global mean remains matched to the pre-change baseline.

**Ordered Categorical Drift.** For data defined on a finite ordinal support $\mathcal{X} = \{1, \dots, M\}$ (with $M = 6$), we model gradual drifts that respect the underlying ordinal structure. Let the pre-change probability mass function be denoted by $p_0 \in \Delta^{M-1}$. We define a target "right-shifted" distribution $p_{\text{shift}}$ wherein probability mass moves to adjacent higher categories. Formally, the shifted probability for category $j$ is zero for the first category, equal $p_{0,j-1}$ for intermediate categories, and accumulates the mass of the top two original categories at the upper boundary $j = M$. The stream evolves via a linear interpolation $p_t = (1 - \gamma_t)p_0 + \gamma_t p_{\text{shift}}$, where the drift parameter $\gamma_t$ ramps linearly from 0 to 1 following the change point.

## H.2. Discrete Baseline Detectors

To establish comparative performance benchmarks for discrete data, we implemented two standard sequential change-point detectors adapted for count and categorical streams, respectively.

The first baseline is the **Poisson Cumulative Sum Detector** (adapted from the $c$-Chart), designed for monitoring shifts in total counts. This method relies on the property that the sum of independent Poisson variables is itself Poisson distributed. For a batch at time $t$, we calculate the sufficient statistic $S_t = \sum_{i=1}^N x_{t,i}$. Using the pre-change samples, we estimate the expected mean count $\bar{c} = \mathbb{E}[S_t]$. A detection threshold is established at three standard deviations from the mean, defining an acceptance region $[\max(0, \bar{c} - 3\sqrt{\bar{c}}), \bar{c} + 3\sqrt{\bar{c}}]$. An alarm is raised at time $t$ if the aggregate count $S_t$ falls outside this interval, signaling a significant deviation in the rate parameter.

The second baseline is the **Multinomial Max-Deviation Detector**, used for monitoring categorical data. This approach tracks the maximum deviation of any single class proportion from its expected baseline, effectively monitoring the $L_\infty$ norm of the proportion shift. Let $\hat{p}_{t,j}$ be the empirical proportion of class $j$ in the batch at time $t$, and let $p_{0,j}$ be the expected proportion estimated from reference data. We compute the standardized deviation score $z_{t,j}$ for each class by normalizing the difference $\hat{p}_{t,j} - p_{0,j}$ by the standard error $\sqrt{p_{0,j}(1 - p_{0,j})/N}$. The monitoring statistic is defined as the maximum absolute deviation across all categories, $Z_t = \max_j |z_{t,j}|$. A change is declared if $Z_t$ exceeds a threshold $h$, which is empirically calibrated to satisfy the target in-control average run length (ARL$_0$).

# I. Computational Acceleration Details

The geometric framework proposed in this work relies on the computation of Wasserstein barycenters and optimal transport maps, which can be computationally intensive in high-dimensional or large-sample regimes. To ensure the scalability of our online change-point detector, we adopt the following acceleration strategies.

## I.1. Efficient Barycenter Estimation via Normalizing Flows

Estimating the reference Wasserstein barycenter $\bar{\mu}$ typically requires solving a large-scale optimization problem over the space of probability measures. To accelerate this in high-dimensional settings ($d \geq 10$) and with large sample sizes, we leverage the *Conditional Normalizing Flow* (CNF) framework proposed by Visentin & Cheridito (2026). Instead of discretizing the support, this method parameterizes the barycenter and the associated transport maps using invertible neural networks (normalizing flows).

## I.2. Entropic Regularization for OT

For the computation of pairwise transport costs and maps in the monitoring phase, exact linear programming solvers (with cubic complexity $O(N^3)$) become a bottleneck for streaming applications. We therefore adopt the Sinkhorn algorithm (Cuturi, 2013), which solves the entropically regularized optimal transport problem:

$$\mathcal{L}_\varepsilon(\mu, \nu) = \inf_{\gamma \in \Pi(\mu,\nu)} \int c(x,y) d\gamma(x,y) + \varepsilon H(\gamma),$$

where $H(\gamma)$ is the entropic regularization term. This formulation allows the optimal coupling to be computed via efficient iterative matrix scaling (Sinkhorn-Knopp algorithm), reducing the complexity to approximately $O(N^2)$.

# J. Detection-Delay Analysis

We complement the false-alarm analysis in Section 3.4 with a detection-delay ($\text{ARL}_1$) guarantee. Under sustained post-change batches from $\mathbb{P}_2$, the statistics $(T^2_{t,K}, \text{SPE}_t)$ are conditionally i.i.d., so the run length after $\kappa$ is geometric with parameter $p_1 := \mathbb{P}_{\mathbb{P}_2}(T^2_{t,K} > h_{T^2} \text{ or } \text{SPE}_t > h_{\text{SPE}})$ and $\text{ARL}_1 = 1/p_1$.

Assume that for $t \geq \kappa$ the tangent fields are Gaussian with mean $m \in H$ and the pre-change covariance $\Gamma$. Let $m_k = \langle m, \phi_k \rangle$ denote the projection of the mean shift onto the $k$-th eigenfunction. The retained scores $\xi_{t,k} = \langle v_t, \phi_k \rangle / \sqrt{\lambda_k}$ have mean $m_k / \sqrt{\lambda_k}$, so $T^2_{t,K} \sim \chi^2_K(\delta^2_T)$ with $\delta^2_T = \sum_{k=1}^K m_k^2 / \lambda_k$. This gives $p_T = 1 - F_{\chi^2_K(\delta^2_T)}(h_{T^2})$. The residual $\text{SPE}_t = \sum_{k>K} \xi^2_{t,k}$ is a generalized noncentral chi-square with $p_S = \mathbb{P}(\text{SPE}_t > h_{\text{SPE}})$. Since the retained and residual scores are projections onto orthogonal subspaces of a Gaussian field, $T^2$ and SPE are independent, giving

$$\text{ARL}_1 = \frac{1}{1 - (1 - p_T)(1 - p_S)}.$$

Without Gaussianity, a union bound yields

$$\frac{1}{p_T + p_S} \leq \text{ARL}_1 \leq \frac{1}{\max(p_T, p_S)}.$$

This characterizes how the non-centrality of the post-change shift translates into expected detection delay.

# K. Sensitivity to Pre-Change Calibration Length $n_0$

**Experimental settings.** We use $n_0 = 300$ for all synthetic experiments and the FlowCAP-II case study, and $n_0 = 50$ for the Reddit case study, where a longer pre-change window is unavailable due to data availability constraints.

**Role of $n_0$.** The calibration length $n_0$ governs three estimation tasks: (i) the Fréchet barycenter $\bar{\mu}$; (ii) the eigendecomposition of the empirical covariance operator $\widehat{\mathcal{C}}$ (MFPCA); and (iii) the empirical quantile thresholds. Corollary 3.11 quantifies the finite-sample effect on (iii) directly: the lower bound on $\text{ARL}_0$ contains an additive correction of $2/(n_0 + 1)$, which equals 0.007 for $n_0 = 300$ (negligible) but 0.039 for $n_0 = 50$ (non-trivial at aggressive false-alarm rates). This finite-sample inflation partially explains our choice of a qualitative evaluation for the Reddit case study. Functional eigenpair consistency requires $n_0 \to \infty$ with estimation error $O(n_0^{-1/2})$ for fixed truncation $K$ (Hall & Hosseini-Nasab, 2006), which provides sufficient accuracy at $n_0 = 300$ for the truncation levels used here ($K \leq 10$). Empirical barycenters enjoy finite-sample concentration in metric spaces with curvature bounds, so the barycenter approximation error in our setting is dominated by within-batch sampling error rather than $n_0$ when $n_0$ is moderately large.

*Table 3.* Summary of IDD performance across all experiments, including settings where competitors match or outperform IDD.

| Scenario | IDD advantage | Methods that match/beat IDD |
|---|---|---|
| Barycenter change | Ties with best | Hotelling $T^2$, Shewhart, NEWMA |
| Multimodal reweight | Strong ($d \geq 5$) | Log-KDE competitive in $d=1$ |
| Copula shift | Strong (small $N$) | Log-KDE competitive at large $N$ |
| Gaussian translation | Disadvantage (small $\delta$) | Hotelling $T^2$ better for pure mean shifts |
| Poisson spike (Fig. 2a) | Moderate disadvantage | $C$-CHART better (specialized detector) |
| Ordered categorical | Advantage | — |
| AML FlowCAP (Fig. 3) | Best overall | NEWMA comparable at low $\mathrm{ARL}_0$ |

**Limitation and practitioner guidance.** A small $n_0$ simultaneously inflates all three estimation errors, making the detector less calibrated and less powerful. Based on the above analysis, we recommend the following minimum calibration lengths as practical guidance:

- **Threshold calibration (task iii):** $n_0 \geq 100$ to keep the $\mathrm{ARL}_0$ correction $2/(n_0 + 1) \leq 0.02$, i.e., below 2% additive inflation at any false-alarm rate.

- **MFPCA eigendecomposition (task ii):** $n_0 \geq 5K$ is a conservative rule of thumb to ensure the leading $K$ eigenpairs are estimated with $O(n_0^{-1/2})$ error well below the signal level. For $K \leq 10$, this requires $n_0 \geq 50$.

- **Barycenter estimation (task i):** Since barycenter error is dominated by within-batch sampling noise when $N_t$ is moderate, $n_0$ plays a secondary role here; $n_0 \geq 30$ suffices in practice.

When data availability forces a small $n_0$ (as in Reddit), we recommend interpreting results qualitatively, increasing the nominal $\mathrm{ARL}_0$ target to absorb the $2/(n_0 + 1)$ inflation, or using permutation-based thresholds that are exact for any $n_0$.

## L. Comprehensive Per-Scenario Method Comparison

We provide a per-scenario discussion of all baselines, including settings where IDD is matched or outperformed.

**Barycenter change.** The shift is essentially a location change, so moment-based methods recover their classical efficiency. Across $d \in \{1, 5, 10, 50\}$ and $N \geq 100$, Hotelling $T^2$, Shewhart, Log-KDE, NEWMA and IDD all attain $\mathrm{ARL}_1 \approx 1.0$. Log-KDE degrades at $d=50$, $N=300$ due to bandwidth instability. IDD matches the best but offers no advantage in this regime.

**Multimodal reweight.** Mixture weights change while the global mean is preserved, so Shewhart fails ($\mathrm{ARL}_1 \in [36, 190]$). IDD dominates for $d \geq 5$. In $d=1$, Log-KDE is competitive (and beats IDD at $N=100$) because univariate KDE is highly accurate; NEWMA is the most consistent runner-up.

**Copula shift.** Marginals are preserved while dependencies change. IDD and Log-KDE both attain $\mathrm{ARL}_1 = 1.0$ for $N \geq 100$. IDD has a clearer edge at small $N$. Shewhart fails when marginals are preserved.

**Gaussian translation (Tables 1,2).** For pure mean shifts at small magnitude ($\delta = 0.1$), Hotelling $T^2$ directly monitors the sufficient statistic and frequently outperforms IDD (e.g., $d=5$, $\sigma=0.5$: Hotelling 1.0 vs. IDD $1.9 \pm 0.5$). This is consistent with Theorem F.1: IDD's OT-map estimation introduces variance that an optimally specified mean chart avoids. For larger shifts ($\delta = 0.5$) all methods converge to $\mathrm{ARL}_1 = 1.0$.

**Poisson spike injection.** The $C$-CHART is a specialized count detector monitoring the sufficient statistic $S_t = \sum_i x_{t,i}$; spike injection inflates exactly this quantity, so $C$-CHART achieves shorter delays than IDD by design. IDD still outperforms Log-KDE since OT geometry preserves discrete spike structure better than kernel smoothing.

**Ordered categorical drift.** IDD exploits the ordinal metric and detects coherent mass flow more efficiently than attribute charts, which treat classes as nominal.

*Table 4.* Phase I calibration time for synthetic continuous-stream experiments. Units are ms per sample; entries are mean (SE).

| $N$ | $d$ | $K$ | OT | KDE-Full | KDE-Marginal | $\bar{X}$/Hotelling $T^2$ | NEWMA | Scan-B | F-CPD |
|---|---|---|---|---|---|---|---|---|---|
| 50 | 1 | 4 | 0.184 (0.007) | 0.911 (0.012) | 0.919 (0.017) | 0.010 (0.000) | 0.205 (0.002) | 0.400 (0.003) | 6.864 (0.045) |
| 50 | 5 | 4 | 292.124 (4.869) | 0.903 (0.008) | 4.354 (0.045) | 0.023 (0.000) | 0.230 (0.001) | 0.411 (0.003) | 6.816 (0.020) |
| 50 | 10 | 4 | 332.594 (0.575) | 0.983 (0.009) | 8.496 (0.017) | 0.041 (0.000) | 0.253 (0.001) | 0.439 (0.004) | 6.989 (0.071) |
| 50 | 50 | 4 | 480.422 (0.799) | 1.591 (0.015) | 42.363 (0.059) | 0.234 (0.000) | 1.197 (0.099) | 1.537 (0.113) | 8.112 (0.077) |
| 100 | 1 | 4 | 0.204 (0.006) | 1.341 (0.006) | 1.343 (0.005) | 0.014 (0.000) | 0.383 (0.002) | 0.588 (0.009) | 7.894 (0.024) |
| 100 | 5 | 4 | 303.171 (0.507) | 1.680 (0.008) | 6.813 (0.011) | 0.044 (0.000) | 0.451 (0.002) | 0.642 (0.004) | 8.150 (0.021) |
| 100 | 10 | 4 | 377.411 (0.923) | 1.856 (0.005) | 13.465 (0.032) | 0.095 (0.000) | 0.489 (0.003) | 0.686 (0.004) | 7.765 (0.024) |
| 100 | 50 | 4 | 501.894 (0.892) | 3.374 (0.010) | 67.953 (0.166) | 0.482 (0.000) | 2.790 (0.021) | 2.969 (0.045) | 10.440 (0.088) |
| 300 | 1 | 4 | 0.316 (0.026) | 3.355 (0.010) | 3.549 (0.100) | 0.031 (0.000) | 1.135 (0.008) | 1.333 (0.009) | 9.496 (0.060) |
| 300 | 5 | 4 | 477.198 (4.262) | 4.465 (0.026) | 17.803 (0.138) | 0.149 (0.000) | 3.898 (0.033) | 4.141 (0.061) | 11.889 (0.065) |
| 300 | 10 | 4 | 727.812 (8.781) | 4.966 (0.014) | 37.531 (1.378) | 0.294 (0.000) | 3.038 (0.055) | 3.742 (0.050) | 11.412 (0.114) |
| 300 | 50 | 4 | 866.880 (21.230) | 8.891 (0.037) | 175.223 (1.749) | 1.410 (0.000) | 4.914 (0.041) | 4.935 (0.077) | 13.517 (0.308) |

*Table 5.* Phase II online deployment time for synthetic continuous-stream experiments. Units are ms per sample; entries are mean (SE). Phase II values are averaged over MM reweight, Copula shift, and Barycenter scenarios.

| $N$ | $d$ | $K$ | OT | KDE-Full | KDE-Marginal | $\bar{X}$/Hotelling $T^2$ | NEWMA | Scan-B | F-CPD |
|---|---|---|---|---|---|---|---|---|---|
| 50 | 1 | 4 | 0.212 (0.006) | 0.969 (0.016) | 0.958 (0.013) | 0.009 (0.000) | 0.206 (0.002) | 0.436 (0.002) | 12.246 (0.559) |
| 50 | 5 | 4 | 17.313 (0.332) | 0.947 (0.006) | 4.436 (0.018) | 0.023 (0.000) | 0.228 (0.001) | 0.461 (0.004) | 11.667 (0.022) |
| 50 | 10 | 4 | 26.433 (0.872) | 0.998 (0.004) | 8.680 (0.015) | 0.042 (0.000) | 0.249 (0.001) | 0.484 (0.001) | 12.027 (0.115) |
| 50 | 50 | 4 | 31.265 (0.995) | 1.599 (0.007) | 48.446 (0.041) | 0.227 (0.002) | 1.452 (0.094) | 1.672 (0.086) | 12.884 (0.092) |
| 100 | 1 | 4 | 0.243 (0.021) | 1.358 (0.013) | 1.375 (0.009) | 0.013 (0.000) | 0.379 (0.003) | 0.619 (0.003) | 13.733 (0.022) |
| 100 | 5 | 4 | 29.651 (0.717) | 1.709 (0.010) | 6.964 (0.020) | 0.046 (0.001) | 0.444 (0.002) | 0.687 (0.006) | 13.994 (0.033) |
| 100 | 10 | 4 | 47.805 (1.633) | 1.885 (0.008) | 13.503 (0.063) | 0.096 (0.002) | 0.495 (0.003) | 0.725 (0.003) | 13.407 (0.025) |
| 100 | 50 | 4 | 48.868 (1.763) | 3.409 (0.009) | 72.002 (0.080) | 0.484 (0.009) | 2.839 (0.015) | 2.983 (0.069) | 16.183 (0.115) |
| 300 | 1 | 4 | 0.272 (0.003) | 3.295 (0.024) | 3.440 (0.043) | 0.030 (0.001) | 1.120 (0.007) | 1.387 (0.011) | 15.498 (0.101) |
| 300 | 5 | 4 | 170.580 (6.157) | 4.511 (0.030) | 17.983 (0.169) | 0.154 (0.003) | 3.919 (0.023) | 4.141 (0.018) | 18.152 (0.136) |
| 300 | 10 | 4 | 236.114 (8.591) | 4.993 (0.020) | 35.330 (0.212) | 0.294 (0.003) | 2.994 (0.058) | 3.780 (0.037) | 17.118 (0.114) |
| 300 | 50 | 4 | 272.048 (11.912) | 8.863 (0.052) | 171.644 (0.981) | 1.409 (0.006) | 4.548 (0.071) | 5.100 (0.030) | 20.158 (0.115) |

**FlowCAP-II AML.**  IDD achieves the highest F1 ($\approx 0.75$) with $\mathrm{ARL}_1 \approx 1$ and near-perfect precision ($\approx 0.99$). NEWMA is comparable in F1 at lenient $\mathrm{ARL}_0$ thresholds (30–75) due to its higher recall (precision $\approx 0.83$, recall $\approx 0.52$) but later first detection. At stricter thresholds ($\mathrm{ARL}_0 > 75$) NEWMA's F1 advantage at a single $\mathrm{ARL}_0$ cross-section coexists with a larger $\mathrm{ARL}_1$, reflecting the difference between batch-level labeling accuracy and first-detection speed.

**Reddit sentiment.**  Qualitative evaluation only; IDD produces sharp alarms aligned with the J&J pause and subsequent policy shifts, whereas NEWMA exhibits monotonic drift and F-CPD high variance unrelated to events.

## M. Computational Cost: Phase I and Phase II Timing

Tables 4–5 report per-sample timing (ms) for all methods across our synthetic configurations. Phase I (offline calibration) ranges from $\sim 0.2$ ms ($d$=1, $N$=50) to $\sim 867$ ms ($d$=50, $N$=300); this is a one-time cost. Phase II (online monitoring) is substantially cheaper: under 50 ms for $d \leq 10$ and $\sim 272$ ms at $d$=50, $N$=300. Competitors KDE-Marginal scale similarly ($\sim 172$ ms), F-CPD incurs 12–20 ms but with weaker detection power, while Shewhart/Hotelling $T^2$ is fastest ($< 1.5$ ms) but fails to detect higher-order distributional changes. For the real case studies, FlowCAP-II ($d$=7, $N_t$=500) has Phase II cost $\sim 2.9$ s, practical given minute-to-hour batch arrivals; Reddit ($d$=20, $N_t \in [20, 412]$) has Phase II cost $\sim 0.19$ s, well below daily monitoring frequency.

## N. Implementation Details and Hyperparameters

Table 6 summarizes our solver choices and hyperparameters across all experiments. We use exact OT (POT `ot.emd()`) for $d \leq 10$ and $N_t \leq 500$, and Sinkhorn (`ot.sinkhorn()`) otherwise. Barycenter computation uses a closed-form quantile

*Table 6.* Example implementation details and hyperparameters used across our experiments. CNF stands for Conditional Normalizing Flows.

| | Synthetic ($d$=1) | Synthetic ($d$=5) | Synthetic ($d$=10) | Synthetic ($d$=50) | FlowCAP-II (AML) | Reddit Sentiment |
|---|---|---|---|---|---|---|
| **Data** | | | | | | |
| Dimension $d$ | 1 | 5 | 10 | 50 | 7 | 20 |
| Batch size $N_t$ | {50, 100, 300} | {50, 100, 300} | {50, 100, 300} | {50, 100, 300} | $\sim$100–300 | $\sim$50–200 |
| Pre-change $n_0$ | 300 | 300 | 300 | 300 | 300 | 50 |
| MFPCA trunc. $K$ | CVE $\geq$ 0.95 | CVE $\geq$ 0.95 | CVE $\geq$ 0.95 | CVE $\geq$ 0.95 | CVE $\geq$ 0.95 | CVE $\geq$ 0.95 |
| **OT Solver** | | | | | | |
| Method | Exact LP | Exact LP | Sinkhorn | Sinkhorn | Sinkhorn | Sinkhorn |
| Library call | `ot.emd()` | `ot.emd()` | `ot.sinkhorn()` | `ot.sinkhorn()` | `ot.sinkhorn()` | `ot.sinkhorn()` |
| `reg` ($\varepsilon$) | — | — | 0.05 | 0.05 | 0.05 | 0.05 |
| `numIterMax` | — | — | 5000 | 5000 | 5000 | 5000 |
| `stopThr` | — | — | 1e-4 | 1e-4 | 1e-4 | 1e-4 |
| `use_eps_scaling` | — | — | True | True | True | True |
| **Barycenter** | | | | | | |
| Method | Closed-form | Fixed-support Sinkhorn | CNF | CNF | CNF | CNF |
| `n_bary` | — | 512 | — | — | — | — |
| Sinkhorn inner iters | — | 500 | — | — | — | — |
| Fixed-point outer iters | — | 300 | — | — | — | — |
| `flow_bar_hidden` | — | — | 32 | 32 | 32 | 32 |
| `flow_bar_blocks` | — | — | 8 | 8 | 8 | 8 |
| `flow_ot_hidden` | — | — | 64 | 64 | 64 | 64 |
| `flow_ot_blocks` | — | — | 16 | 16 | 16 | 16 |
| `epochs` | — | — | 500 | 500 | 500 | 500 |
| `batch_size` (train) | — | — | 2048 | 2048 | 2048 | 2048 |
| `lr` | — | — | 1e-3 | 1e-3 | 1e-3 | 1e-3 |
| `grad_clip` | — | — | 2.0 | 2.0 | 2.0 | 2.0 |
| Temp. schedule | — | — | $1.0 \to 10^{-2}$ | $1.0 \to 10^{-2}$ | $1.0 \to 10^{-2}$ | $1.0 \to 10^{-2}$ |
| LR scheduler | — | — | Plateau (0.8, pat=1000) | Plateau (0.8, pat=1000) | Plateau (0.8, pat=1000) | Plateau (0.8, pat=1000) |

average in $d$=1, fixed-support Sinkhorn barycenters with a fixed-point LP scheme for $d \leq 5$, and Conditional Normalizing Flows (Appendix I) for $d > 5$. MFPCA truncation $K$ is chosen by cumulative variance explained (CVE) $\geq 0.95$, which yields $K$ between 3 and 10 across our experiments.

# O. Additional Experimental Results

In this section, we provide the complete set of experimental results for the synthetic continuous streams. We evaluate the performance across a wide range of configurations by varying the batch size $N \in \{50, 100, 300\}$ and the dimension $d \in \{1, 5, 10, 50\}$. The results presented below include the full trade-off curves comparing $\text{ARL}_1$ against the $\text{ARL}_0$ for the Barycenter Change, Multimodal Reweight and Copula Shift scenarios. Additionally, we report the Detection Rate versus False Alarm Rate to assess the sensitivity of the detectors at varying thresholds. Finally, we provide detailed tabular results for all the scenarios, reporting the mean detection delay with standard errors over 10 replications at a fixed calibration target.

**Marginal monitoring vs. full distribution.** To verify that the proposed framework solves a strictly harder problem than monitoring concatenated marginal streams, we implement a KDE-Marginal baseline that estimates per-coordinate marginal densities by KDE and monitors functional $L^2$ distances. Table 7 compares KDE-Marginal against IDD under the copula-shift scenario at $\alpha = 0.05$. For $d \geq 5$, IDD detects immediately ($\text{ARL}_1 = 1.0$) across all batch sizes, whereas KDE-Marginal detects slowly ($\text{ARL}_1$ between 7 and 18), confirming that marginal monitoring is fundamentally less powerful when the change lies in dependence structure.

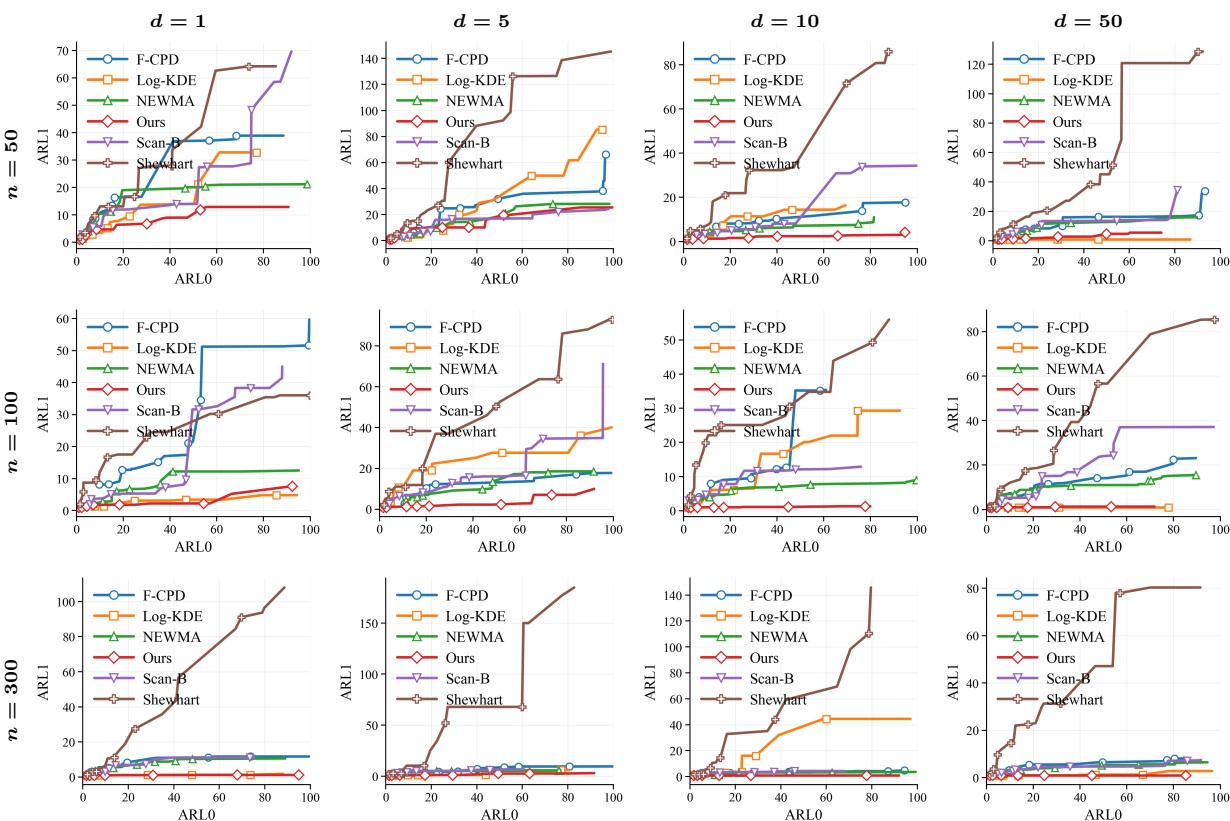

*Figure 5.* Multimodal Reweight ($\text{ARL}_1$ $\text{ARL}_0$):Performance comparison across varying sample sizes $n \in \{50, 100, 300\}$ (rows) and dimensions $d \in \{1, 5, 10, 50\}$ (columns).

*Table 7.* Synthetic results for the **Copula Shift** scenario comparing the proposed OT-based detector against a KDE-Marginal baseline that monitors per-coordinate marginal densities via functional $L^2$ distances. For each $(N, d, K)$ configuration, the row with the smaller mean $ARL_1$ is highlighted in bold.

| Method | $N$ | $d$ | $K$ | $ARL_1$ (mean) | $ARL_1$ (SE) |
|---|---|---|---|---|---|
| KDE-Marginal | 50 | 1 | 4 | 16.600 | 2.729 |
| **OT (Ours)** | **50** | **1** | **4** | **10.800** | **3.782** |
| KDE-Marginal | 50 | 5 | 4 | 13.700 | 3.073 |
| **OT (Ours)** | **50** | **5** | **4** | **1.100** | **0.100** |
| KDE-Marginal | 50 | 10 | 4 | 6.600 | 1.507 |
| **OT (Ours)** | **50** | **10** | **4** | **1.000** | **0.000** |
| KDE-Marginal | 50 | 50 | 4 | 8.400 | 1.454 |
| **OT (Ours)** | **50** | **50** | **4** | **1.000** | **0.000** |
| KDE-Marginal | 100 | 1 | 4 | 13.100 | 2.927 |
| **OT (Ours)** | **100** | **1** | **4** | **10.800** | **4.482** |
| KDE-Marginal | 100 | 5 | 4 | 8.700 | 1.535 |
| **OT (Ours)** | **100** | **5** | **4** | **1.000** | **0.000** |
| KDE-Marginal | 100 | 10 | 4 | 16.000 | 4.222 |
| **OT (Ours)** | **100** | **10** | **4** | **1.000** | **0.000** |
| KDE-Marginal | 100 | 50 | 4 | 17.800 | 3.759 |
| **OT (Ours)** | **100** | **50** | **4** | **1.000** | **0.000** |
| KDE-Marginal | 300 | 1 | 4 | 13.900 | 3.816 |
| **OT (Ours)** | **300** | **1** | **4** | **9.500** | **3.622** |
| KDE-Marginal | 300 | 5 | 4 | 11.300 | 2.305 |
| **OT (Ours)** | **300** | **5** | **4** | **1.000** | **0.000** |
| KDE-Marginal | 300 | 10 | 4 | 7.800 | 1.775 |
| **OT (Ours)** | **300** | **10** | **4** | **1.000** | **0.000** |
| KDE-Marginal | 300 | 50 | 4 | 9.600 | 1.933 |
| **OT (Ours)** | **300** | **50** | **4** | **1.000** | **0.000** |

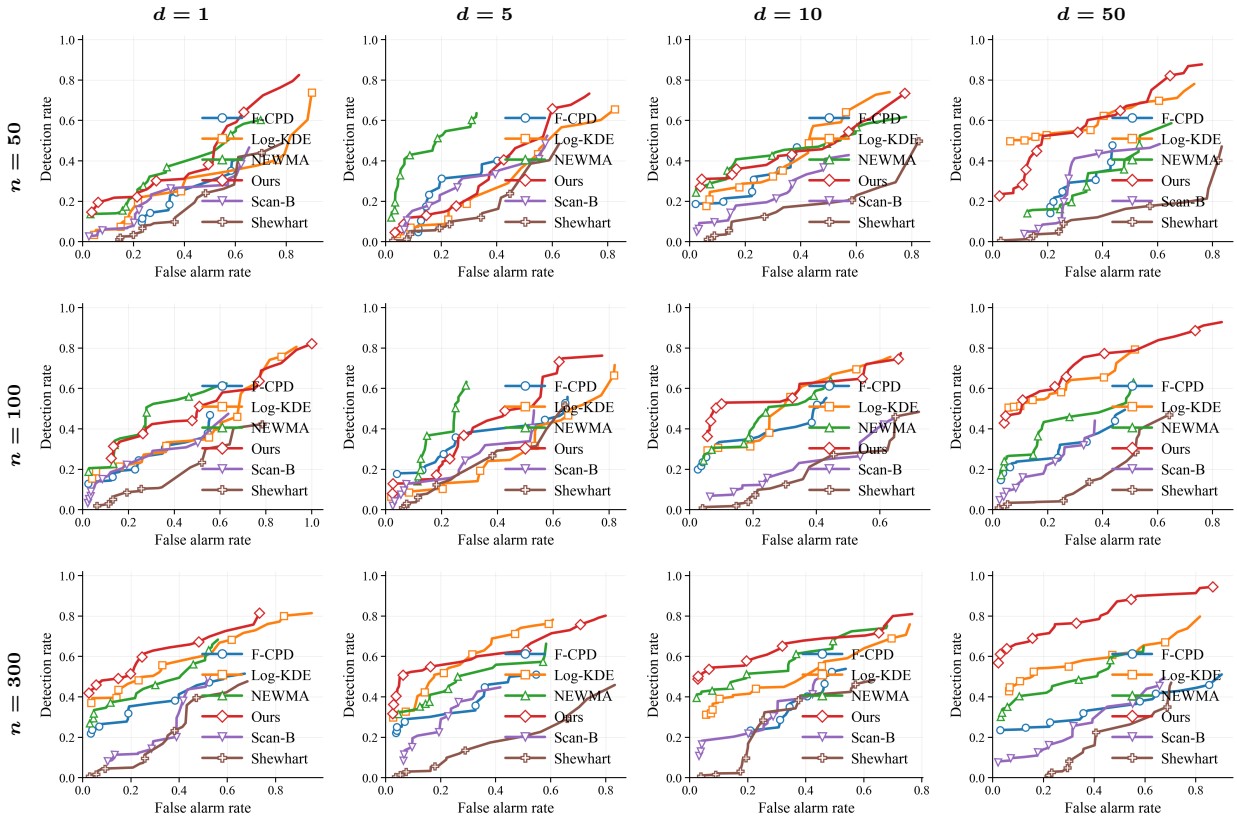

*Figure 6.* Multimodal Reweight (Detection Rate vs FAR): Performance comparison across varying sample sizes $n \in \{50, 100, 300\}$ (rows) and dimensions $d \in \{1, 5, 10, 50\}$ (columns).

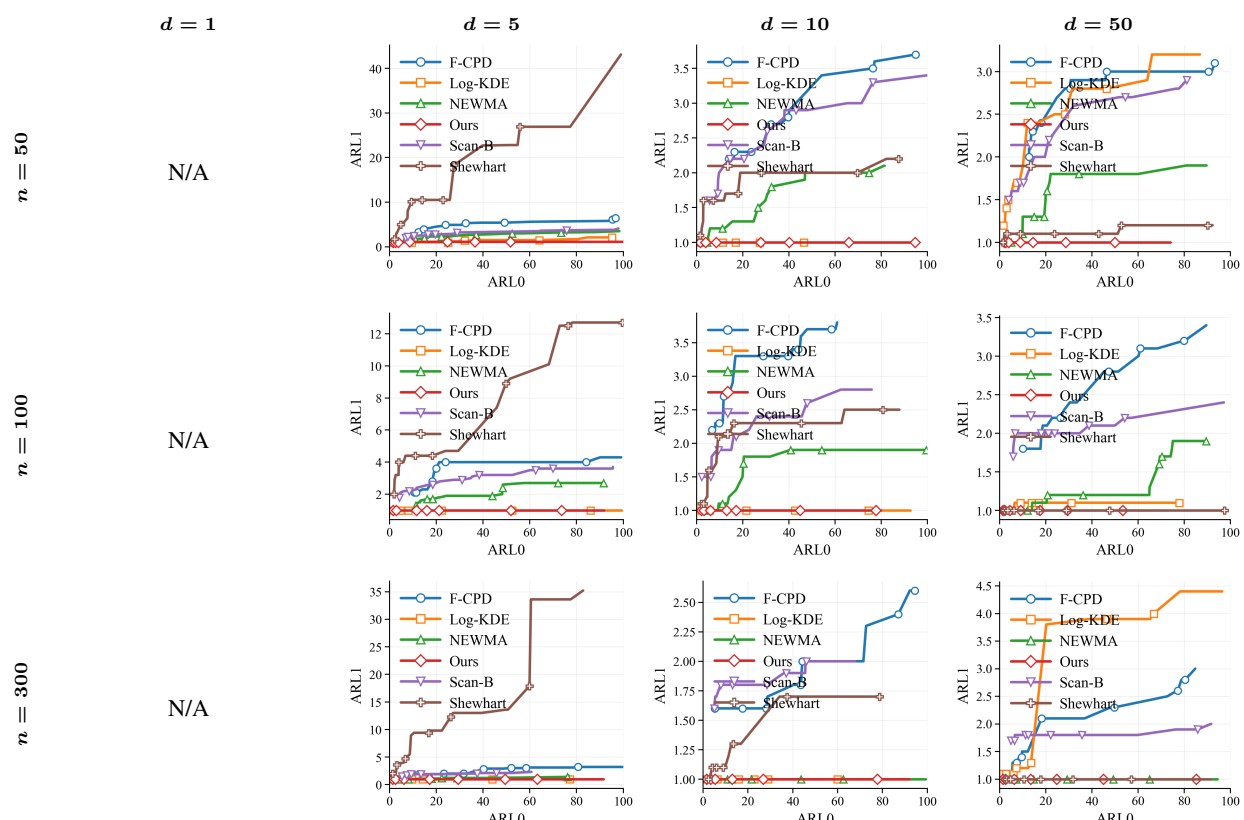

*Figure 7.* Copula Shift (ARL$_1$ ARL$_0$) Performance comparison across varying sample sizes $n \in \{50, 100, 300\}$ (rows) and dimensions $d \in \{5, 10, 50\}$ (columns).

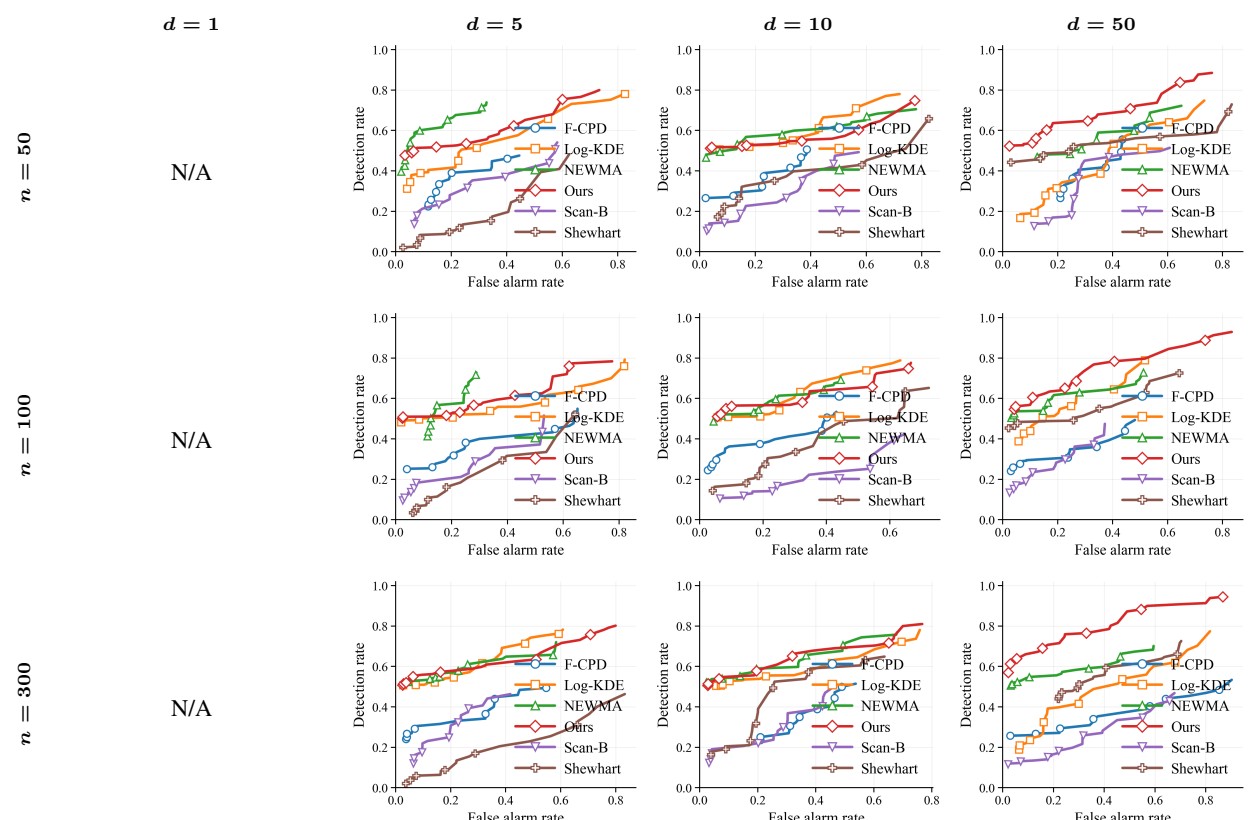

*Figure 8.* Copula Shift (Detection Rate vs FAR): Performance comparison across varying sample sizes $n \in \{50, 100, 300\}$ (rows) and dimensions $d \in \{5, 10, 50\}$ (columns).

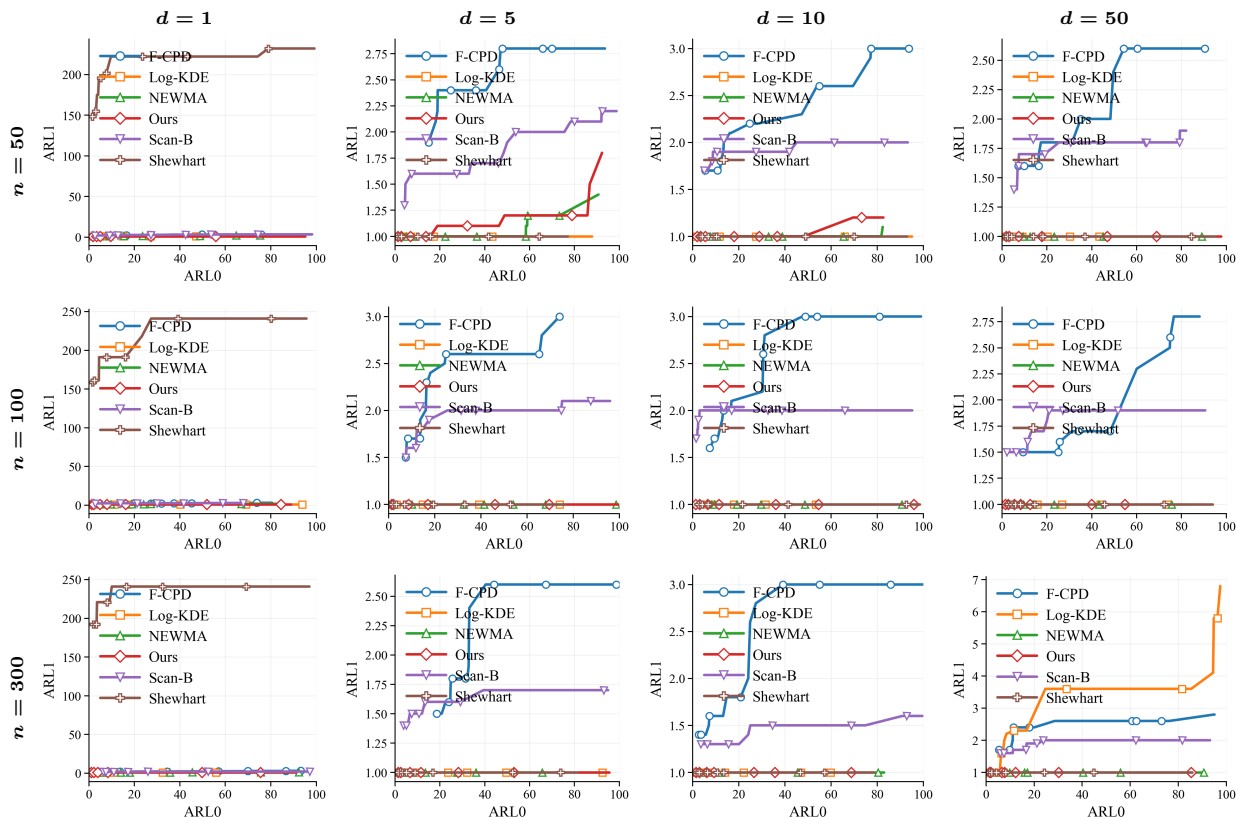

*Figure 9.* Barycenter Change ($\mathrm{ARL}_1$ $\mathrm{ARL}_0$): Performance comparison across varying sample sizes $n \in \{50, 100, 300\}$ (rows) and dimensions $d \in \{1, 5, 10, 50\}$ (columns).

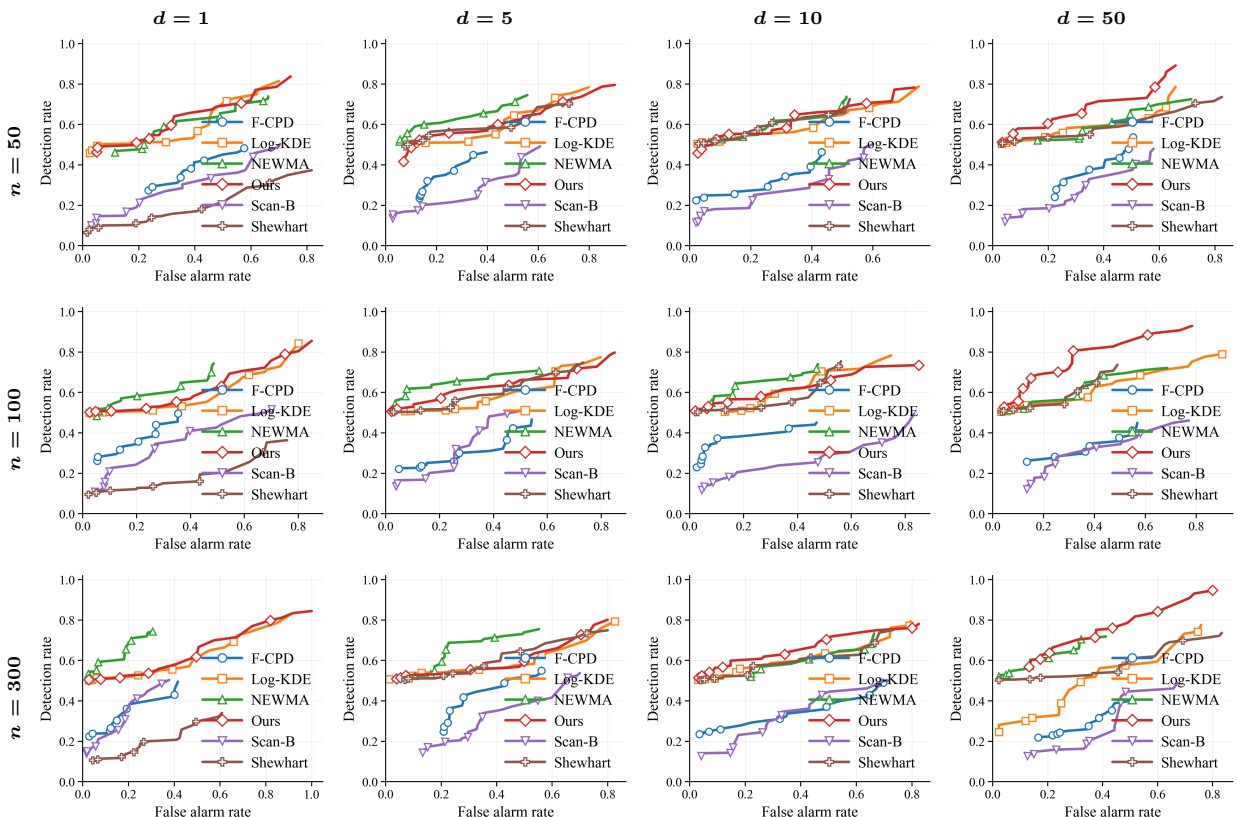

*Figure 10.* **Barycenter Change (Detection Rate vs FAR):** Performance comparison across varying sample sizes $n \in \{50, 100, 300\}$ (rows) and dimensions $d \in \{1, 5, 10, 50\}$ (columns).

## O.1. Summary of Findings

The comprehensive results reinforce the robustness of the proposed framework across diverse shift types and dimensions. In the Barycenter scenario, where the shift is primarily in the mean, IDD achieves near-instantaneous detection ($\mathrm{ARL}_1 \approx 1.0$), matching the performance of the specialized Shewhart chart. However, the advantage of our geometric approach becomes evident in complex distributional shifts such as Multimodal Reweighting. The Shewhart chart fails as the global mean remains unchanged, while IDD consistently maintains the lowest detection delay. Furthermore, regarding scalability, while density-based baselines like Log-KDE are competitive in lower dimensions, their performance degrades in high-dimensional settings ($d = 50$), particularly in the Copula Shift scenario. In contrast, IDD remains unaffected by the curse of dimensionality, outperforming most baselines in these challenging regimes.

## P. Details on Flow Cytometry Case Study

### P.1. Dataset and Preprocessing

The dataset is derived from the FlowCAP-II challenge (Aghaeepour et al., 2013), comprising 2,874 peripheral blood or bone marrow measurements collected from 359 subjects (316 healthy donors and 43 AML patients). Note that multiple measurements are available for each subject. We treat each flow cytometry measurement as a distinct distribution-valued observation. From each measurement, we extract a random subsample of $N = 2,000$ cells to form the empirical measure $\mu_t$. Each cell is represented in $d = 7$ dimensions, consisting of forward scatter (FSC), side scatter (SSC), CD45 expression, and 4 other lineage markers.

*Table 8.* Synthetic results for **Barycenter** with $d = 1$ (10 replications). Best (smallest) $ARL_1$ is bold.

| Method | $N$ | $ARL_0$ (emp) | $ARL_1$ |
|--------|-----|---------------|---------|
| F-CPD | 50 | 79.8 | $3.3 \pm 0.7$ |
| Log-KDE | 50 | 99.4 | $1.1 \pm 0.1$ |
| NEWMA | 50 | 101.6 | $2.4 \pm 0.4$ |
| Ours | 50 | 102.9 | $\mathbf{1.0 \pm 0.0}$ (↑9.1%) |
| Scan-B | 50 | 101.1 | $3.4 \pm 0.4$ |
| Shewhart | 50 | 99.2 | $232.1 \pm 39.5$ |
| F-CPD | 100 | 81.5 | $3.2 \pm 0.3$ |
| Log-KDE | 100 | 104.4 | $\mathbf{1.0 \pm 0.0}$ (↑47.4%) |
| NEWMA | 100 | 111.9 | $1.9 \pm 0.4$ |
| Ours | 100 | 103.4 | $\mathbf{1.0 \pm 0.0}$ (↑47.4%) |
| Scan-B | 100 | 101.2 | $2.9 \pm 0.4$ |
| Shewhart | 100 | 103.7 | $241.0 \pm 40.0$ |
| F-CPD | 300 | 94.2 | $3.0 \pm 0.0$ |
| Log-KDE | 300 | 101.6 | $\mathbf{1.0 \pm 0.0}$ (↑23.1%) |
| NEWMA | 300 | 105.5 | $1.3 \pm 0.2$ |
| Ours | 300 | 107.2 | $\mathbf{1.0 \pm 0.0}$ (↑23.1%) |
| Scan-B | 300 | 97.1 | $2.1 \pm 0.2$ |
| Shewhart | 300 | 97.0 | $241.0 \pm 40.0$ |

*Table 9.* Synthetic results for **Barycenter** with $d = 5$ (10 replications). Best (smallest) $ARL_1$ is bold.

| Method | $N$ | $ARL_0$ (emp) | $ARL_1$ |
|--------|-----|---------------|---------|
| F-CPD | 50 | 93.6 | $2.8 \pm 0.2$ |
| Log-KDE | 50 | 107.2 | $\mathbf{1.0 \pm 0.0}$ (↑28.6%) |
| NEWMA | 50 | 90.8 | $1.4 \pm 0.2$ |
| Ours | 50 | 101.6 | $1.8 \pm 0.5$ |
| Scan-B | 50 | 98.7 | $2.2 \pm 0.1$ |
| Shewhart | 50 | 105.3 | $\mathbf{1.0 \pm 0.0}$ (↑28.6%) |
| F-CPD | 100 | 102.1 | $3.0 \pm 0.0$ |
| Log-KDE | 100 | 97.8 | $\mathbf{1.0 \pm 0.0}$ (↑52.4%) |
| NEWMA | 100 | 98.8 | $\mathbf{1.0 \pm 0.0}$ (↑52.4%) |
| Ours | 100 | 99.7 | $\mathbf{1.0 \pm 0.0}$ (↑52.4%) |
| Scan-B | 100 | 96.0 | $2.1 \pm 0.1$ |
| Shewhart | 100 | 104.4 | $\mathbf{1.0 \pm 0.0}$ (↑52.4%) |
| F-CPD | 300 | 100.4 | $2.6 \pm 0.3$ |
| Log-KDE | 300 | 100.6 | $\mathbf{1.0 \pm 0.0}$ (↑41.2%) |
| NEWMA | 300 | 100.7 | $\mathbf{1.0 \pm 0.0}$ (↑41.2%) |
| Ours | 300 | 95.5 | $\mathbf{1.0 \pm 0.0}$ (↑41.2%) |
| Scan-B | 300 | 95.0 | $1.7 \pm 0.2$ |
| Shewhart | 300 | 101.6 | $\mathbf{1.0 \pm 0.0}$ (↑41.2%) |

*Table 10.* Synthetic results for **Barycenter** with $d = 10$ (10 replications). Best (smallest) $ARL_1$ is bold.

| Method | $N$ | $ARL_0$ (emp) | $ARL_1$ |
|--------|-----|---------------|---------|
| F-CPD | 50 | 94.3 | $3.0 \pm 0.0$ |
| Log-KDE | 50 | 104.3 | $\mathbf{1.0 \pm 0.0}$ (↑16.7%) |
| NEWMA | 50 | 100.8 | $1.2 \pm 0.1$ |
| Ours | 50 | 101.2 | $3.0 \pm 1.9$ |
| Scan-B | 50 | 93.3 | $2.0 \pm 0.0$ |
| Shewhart | 50 | 92.9 | $\mathbf{1.0 \pm 0.0}$ (↑16.7%) |
| F-CPD | 100 | 99.0 | $3.0 \pm 0.0$ |
| Log-KDE | 100 | 108.2 | $\mathbf{1.0 \pm 0.0}$ (↑50.0%) |
| NEWMA | 100 | 101.1 | $\mathbf{1.0 \pm 0.0}$ (↑50.0%) |
| Ours | 100 | 95.9 | $\mathbf{1.0 \pm 0.0}$ (↑50.0%) |
| Scan-B | 100 | 100.5 | $2.0 \pm 0.0$ |
| Shewhart | 100 | 98.5 | $\mathbf{1.0 \pm 0.0}$ (↑50.0%) |
| F-CPD | 300 | 100.1 | $3.0 \pm 0.0$ |
| Log-KDE | 300 | 105.3 | $\mathbf{1.0 \pm 0.0}$ (↑37.5%) |
| NEWMA | 300 | 100.2 | $\mathbf{1.0 \pm 0.0}$ (↑37.5%) |
| Ours | 300 | 100.9 | $\mathbf{1.0 \pm 0.0}$ (↑37.5%) |
| Scan-B | 300 | 99.8 | $1.6 \pm 0.2$ |
| Shewhart | 300 | 101.0 | $\mathbf{1.0 \pm 0.0}$ (↑37.5%) |

*Table 11.* Synthetic results for **Barycenter** with $d = 50$ (10 replications). Best (smallest) $ARL_1$ is bold.

| Method | $N$ | $ARL_0$ (emp) | $ARL_1$ |
|--------|-----|---------------|---------|
| F-CPD | 50 | 101.4 | $2.6 \pm 0.3$ |
| Log-KDE | 50 | 102.6 | $\mathbf{1.0 \pm 0.0}$ (↑50.0%) |
| NEWMA | 50 | 104.0 | $\mathbf{1.0 \pm 0.0}$ (↑50.0%) |
| Ours | 50 | 97.7 | $\mathbf{1.0 \pm 0.0}$ (↑50.0%) |
| Scan-B | 50 | 101.3 | $2.0 \pm 0.0$ |
| Shewhart | 50 | 95.1 | $\mathbf{1.0 \pm 0.0}$ (↑50.0%) |
| F-CPD | 100 | 101.2 | $2.8 \pm 0.2$ |
| Log-KDE | 100 | 94.0 | $\mathbf{1.0 \pm 0.0}$ (↑47.4%) |
| NEWMA | 100 | 95.1 | $\mathbf{1.0 \pm 0.0}$ (↑47.4%) |
| Ours | 100 | 96.8 | $\mathbf{1.0 \pm 0.0}$ (↑47.4%) |
| Scan-B | 100 | 107.2 | $1.9 \pm 0.1$ |
| Shewhart | 100 | 93.9 | $\mathbf{1.0 \pm 0.0}$ (↑47.4%) |
| F-CPD | 300 | 95.1 | $3.3 \pm 0.3$ |
| Log-KDE | 300 | 97.5 | $6.8 \pm 2.9$ |
| NEWMA | 300 | 90.6 | $\mathbf{1.0 \pm 0.0}$ (↑50.0%) |
| Ours | 300 | 110.0 | $\mathbf{1.0 \pm 0.0}$ (↑50.0%) |
| Scan-B | 300 | 104.0 | $2.0 \pm 0.0$ |
| Shewhart | 300 | 107.0 | $\mathbf{1.0 \pm 0.0}$ (↑50.0%) |

*Table 12.* Synthetic results for **Copula Shift** with $d = 5$ (10 replications). Best (smallest) $\mathrm{ARL}_1$ is bold.

| Method | $N$ | $\mathrm{ARL}_0$ (emp) | $\mathrm{ARL}_1$ |
|---|---|---|---|
| F-CPD | 50 | 96.8 | $6.6 \pm 0.7$ |
| Log-KDE | 50 | 103.2 | $2.3 \pm 0.6$ |
| NEWMA | 50 | 89.3 | $3.4 \pm 0.2$ |
| Ours | 50 | 100.1 | $\mathbf{1.1 \pm 0.1}$ (↑52.2%) |
| Scan-B | 50 | 98.3 | $4.3 \pm 0.2$ |
| Shewhart | 50 | 99.0 | $43.1 \pm 17.9$ |
| F-CPD | 100 | 102.0 | $4.4 \pm 0.2$ |
| Log-KDE | 100 | 99.3 | $\mathbf{1.0 \pm 0.0}$ (↑64.3%) |
| NEWMA | 100 | 91.6 | $2.8 \pm 0.2$ |
| Ours | 100 | 91.8 | $\mathbf{1.0 \pm 0.0}$ (↑64.3%) |
| Scan-B | 100 | 95.6 | $3.9 \pm 0.2$ |
| Shewhart | 100 | 99.6 | $12.7 \pm 4.2$ |
| F-CPD | 300 | 99.9 | $3.2 \pm 0.1$ |
| Log-KDE | 300 | 116.7 | $\mathbf{1.0 \pm 0.0}$ (↑37.5%) |
| NEWMA | 300 | 87.6 | $1.6 \pm 0.2$ |
| Ours | 300 | 91.6 | $\mathbf{1.0 \pm 0.0}$ (↑37.5%) |
| Scan-B | 300 | 106.6 | $2.3 \pm 0.2$ |
| Shewhart | 300 | 96.3 | $35.6 \pm 10.2$ |

*Table 13.* Synthetic results for **Copula Shift** with $d = 10$ (10 replications). Best (smallest) $\mathrm{ARL}_1$ is bold.

| Method | $N$ | $\mathrm{ARL}_0$ (emp) | $\mathrm{ARL}_1$ |
|---|---|---|---|
| F-CPD | 50 | 102.6 | $3.7 \pm 0.3$ |
| Log-KDE | 50 | 96.8 | $\mathbf{1.0 \pm 0.0}$ (↑52.4%) |
| NEWMA | 50 | 112.5 | $2.1 \pm 0.1$ |
| Ours | 50 | 100.7 | $\mathbf{1.0 \pm 0.0}$ (↑52.4%) |
| Scan-B | 50 | 99.9 | $3.4 \pm 0.2$ |
| Shewhart | 50 | 105.7 | $2.2 \pm 0.5$ |
| F-CPD | 100 | 108.1 | $3.8 \pm 0.1$ |
| Log-KDE | 100 | 102.1 | $\mathbf{1.0 \pm 0.0}$ (↑47.4%) |
| NEWMA | 100 | 99.6 | $1.9 \pm 0.1$ |
| Ours | 100 | 100.2 | $\mathbf{1.0 \pm 0.0}$ (↑47.4%) |
| Scan-B | 100 | 113.1 | $2.8 \pm 0.1$ |
| Shewhart | 100 | 87.9 | $2.5 \pm 0.4$ |
| F-CPD | 300 | 94.1 | $2.6 \pm 0.3$ |
| Log-KDE | 300 | 102.2 | $\mathbf{1.0 \pm 0.0}$ (↑44.4%) |
| NEWMA | 300 | 100.7 | $\mathbf{1.0 \pm 0.0}$ (↑44.4%) |
| Ours | 300 | 91.7 | $\mathbf{1.0 \pm 0.0}$ (↑44.4%) |
| Scan-B | 300 | 100.1 | $2.0 \pm 0.0$ |
| Shewhart | 300 | 107.7 | $1.8 \pm 0.3$ |

*Table 14.* Synthetic results for **Copula Shift** with $d = 50$ (10 replications), regenerated for camera-ready. Best (smallest) $\mathrm{ARL}_1$ is bold.

| Method | $N$ | $\mathrm{ARL}_0$ (emp) | $\mathrm{ARL}_1$ |
|---|---|---|---|
| F-CPD | 50 | 100.0 | $3.1 \pm 0.4$ |
| Log-KDE | 50 | 100.0 | $3.2 \pm 0.7$ |
| NEWMA | 50 | 100.0 | $1.9 \pm 0.1$ |
| Ours | 50 | 100.0 | $\mathbf{1.0 \pm 0.0}$ (↑16.7%) |
| Scan-B | 50 | 100.0 | $2.9 \pm 0.1$ |
| Shewhart | 50 | 100.0 | $1.2 \pm 0.1$ |
| F-CPD | 100 | 100.0 | $3.4 \pm 0.2$ |
| Log-KDE | 100 | 100.0 | $1.1 \pm 0.1$ |
| NEWMA | 100 | 100.0 | $1.9 \pm 0.1$ |
| Ours | 100 | 100.0 | $\mathbf{1.0 \pm 0.0}$ (↑9.1%) |
| Scan-B | 100 | 100.0 | $2.6 \pm 0.2$ |
| Shewhart | 100 | 100.0 | $\mathbf{1.0 \pm 0.0}$ (↑9.1%) |
| F-CPD | 300 | 100.0 | $3.0 \pm 0.0$ |
| Log-KDE | 300 | 100.0 | $4.4 \pm 1.5$ |
| NEWMA | 300 | 100.0 | $\mathbf{1.0 \pm 0.0}$ (↑50.0%) |
| Ours | 300 | 100.0 | $\mathbf{1.0 \pm 0.0}$ (↑50.0%) |
| Scan-B | 300 | 100.0 | $2.0 \pm 0.0$ |
| Shewhart | 300 | 100.0 | $\mathbf{1.0 \pm 0.0}$ (↑50.0%) |

*Table 15.* Synthetic results for **Multimodal Reweight** with $d = 1$ (10 replications). Best (smallest) $\mathrm{ARL}_1$ is bold.

| Method | $N$ | $\mathrm{ARL}_0$ (emp) | $\mathrm{ARL}_1$ |
|---|---|---|---|
| F-CPD | 50 | 103.0 | $49.6 \pm 21.6$ |
| Log-KDE | 50 | 108.3 | $45.6 \pm 15.6$ |
| NEWMA | 50 | 107.9 | $24.1 \pm 5.4$ |
| Ours | 50 | 107.7 | $\mathbf{15.7 \pm 9.4}$ (↑34.9%) |
| Scan-B | 50 | 93.2 | $69.7 \pm 21.0$ |
| Shewhart | 50 | 101.1 | $93.5 \pm 28.1$ |
| F-CPD | 100 | 99.5 | $59.7 \pm 24.6$ |
| Log-KDE | 100 | 94.3 | $\mathbf{4.8 \pm 1.7}$ (↑36.8%) |
| NEWMA | 100 | 90.1 | $12.2 \pm 4.1$ |
| Ours | 100 | 92.3 | $7.6 \pm 4.0$ |
| Scan-B | 100 | 106.2 | $70.0 \pm 31.2$ |
| Shewhart | 100 | 99.9 | $36.0 \pm 8.7$ |
| F-CPD | 300 | 99.4 | $11.8 \pm 2.4$ |
| Log-KDE | 300 | 88.1 | $1.8 \pm 0.5$ |
| NEWMA | 300 | 105.8 | $10.6 \pm 2.6$ |
| Ours | 300 | 94.9 | $\mathbf{1.2 \pm 0.2}$ (↑33.3%) |
| Scan-B | 300 | 109.5 | $11.1 \pm 4.2$ |
| Shewhart | 300 | 94.5 | $108.0 \pm 20.6$ |

*Table 16.* Synthetic results for **Multimodal Reweight** with $d = 5$ (10 replications). Best (smallest) $ARL_1$ is bold.

| Method | $N$ | $ARL_0$ (emp) | $ARL_1$ |
|---|---|---|---|
| F-CPD | 50 | 96.8 | $66.2 \pm 29.4$ |
| Log-KDE | 50 | 103.2 | $85.4 \pm 34.4$ |
| NEWMA | 50 | 89.3 | $28.2 \pm 6.1$ |
| Ours | 50 | 100.1 | $\mathbf{25.5 \pm 9.5}$ (↑6.9%) |
| Scan-B | 50 | 98.3 | $27.4 \pm 8.6$ |
| Shewhart | 50 | 99.0 | $145.3 \pm 36.6$ |
| F-CPD | 100 | 102.0 | $21.1 \pm 4.3$ |
| Log-KDE | 100 | 99.3 | $40.1 \pm 14.7$ |
| NEWMA | 100 | 91.6 | $18.6 \pm 6.8$ |
| Ours | 100 | 91.8 | $\mathbf{10.0 \pm 4.4}$ (↑46.2%) |
| Scan-B | 100 | 95.6 | $72.3 \pm 32.6$ |
| Shewhart | 100 | 99.6 | $92.9 \pm 26.6$ |
| F-CPD | 300 | 99.9 | $9.7 \pm 1.0$ |
| Log-KDE | 300 | 116.7 | $6.3 \pm 4.5$ |
| NEWMA | 300 | 87.6 | $6.5 \pm 0.8$ |
| Ours | 300 | 91.6 | $\mathbf{3.1 \pm 1.1}$ (↑50.8%) |
| Scan-B | 300 | 106.6 | $6.8 \pm 0.6$ |
| Shewhart | 300 | 96.3 | $190.8 \pm 40.5$ |

*Table 17.* Synthetic results for **Multimodal Reweight** with $d = 10$ (10 replications). Best (smallest) $ARL_1$ is bold.

| Method | $N$ | $ARL_0$ (emp) | $ARL_1$ |
|---|---|---|---|
| F-CPD | 50 | 102.6 | $17.7 \pm 5.2$ |
| Log-KDE | 50 | 96.8 | $16.4 \pm 10.2$ |
| NEWMA | 50 | 112.5 | $11.3 \pm 3.4$ |
| Ours | 50 | 100.7 | $\mathbf{4.3 \pm 1.9}$ (↑61.9%) |
| Scan-B | 50 | 99.9 | $34.3 \pm 17.9$ |
| Shewhart | 50 | 105.7 | $87.3 \pm 25.2$ |
| F-CPD | 100 | 108.1 | $35.4 \pm 22.2$ |
| Log-KDE | 100 | 102.1 | $29.3 \pm 18.2$ |
| NEWMA | 100 | 99.6 | $10.6 \pm 2.7$ |
| Ours | 100 | 100.2 | $\mathbf{1.5 \pm 0.4}$ (↑85.8%) |
| Scan-B | 100 | 113.1 | $13.0 \pm 3.6$ |
| Shewhart | 100 | 87.9 | $56.0 \pm 9.2$ |
| F-CPD | 300 | 94.1 | $4.8 \pm 1.0$ |
| Log-KDE | 300 | 102.2 | $44.5 \pm 30.7$ |
| NEWMA | 300 | 100.7 | $3.7 \pm 0.8$ |
| Ours | 300 | 91.7 | $\mathbf{1.0 \pm 0.0}$ (↑73.0%) |
| Scan-B | 300 | 100.1 | $4.4 \pm 0.7$ |
| Shewhart | 300 | 107.7 | $145.8 \pm 35.1$ |

## P.2. Change Point Detection Experiment Setup

We structure the experiment as a sequential change-point detection task with two phases: (1) Pre-change Calibration Phase: We utilize a sequence of 300 healthy samples to estimate the reference distribution $\bar{\mu}$ and calibrate the detection threshold to satisfy a target $ARL_0$. (2) Monitoring Phase: We monitor a test stream of 300 samples. This stream is constructed by injecting 80% AML-positive samples (randomly sampled from the AML patient pool) interspersed with healthy samples. Since ground truth diagnosis labels are available for every sample, we evaluate performance using both detection delay ($ARL_1$) and classification metrics (F1-score, Precision/Recall) at the batch level.

**Precision/recall breakdown.** Figure 3 in the main text reports both the F1-score (left panel) and the detection delay $ARL_1$ (right panel) versus $ARL_0$. The two metrics evaluate distinct quantities: $ARL_1$ measures first-detection delay (lower is faster), whereas F1 measures batch-level labeling accuracy across the full stream. IDD achieves near-perfect precision ($\approx 0.99$) by raising fewer but highly reliable alarms, yielding the fastest first detection ($ARL_1$ between ∼2 and ∼3) and the highest F1. NEWMA fires more liberally (precision $\approx 0.83$, recall $\approx 0.52$), boosting F1 at lenient $ARL_0$ targets but at the cost of a later first alarm ($ARL_1 \approx 3.6$–3.8). Hence the two methods reflect different trade-offs between first-detection speed and overall labeling coverage. IDD is consistently faster in first detection across the entire $ARL_0$ range.

## Q. Details on Reddit Vaccine Sentiment Study

### Q.1. Preprocessing and Embedding Pipeline

To transform the unstructured text stream into a sequence of distribution-valued observations suitable for our geometric CPD framework, we applied the following preprocessing steps: Firstly, we aggregated user comments into daily batches ($t = 1, \ldots, T$). To ensure statistical stability of the empirical measures, we filtered out days containing fewer than 30 comments. Secondly, we mapped each raw comment to a dense semantic vector using the pre-trained Sentence-BERT model (`all-MiniLM-L6-v2` (Reimers & Gurevych, 2019)), producing 384-dimensional embeddings. This places semantically similar comments (e.g., expressing anxiety or support) close to each other in vector space. Finally, to reduce computational cost while preserving the primary modes of variation, we projected the embeddings to $d = 20$ dimensions using Principal Component Analysis (PCA). Phase I (calibration) consists of the earliest 50-day block (Dec 2, 2020 – Jan 30, 2021); Phase II (monitoring) covers the next 50 days (Jan 31 – May 5, 2021), spanning the J&J EUA and subsequent rollout events.

*Table 18.* Curated COVID-19 vaccine-policy events during the Phase II monitoring window (Jan 31 – May 5, 2021).

| Date | Event | Description |
|---|---|---|
| *Feb 27* | **J&J EUA recommended** | FDA advisory committee recommends Emergency Use Authorization for the Johnson & Johnson vaccine. |
| *Mar 02* | **J&J EUA issued** | FDA issues Emergency Use Authorization for the J&J vaccine. |
| *Mar 11* | All-adults-by-May-1 | President Biden directs states to make all adults eligible for vaccination by May 1. |
| *Apr 13* | **J&J Pause** | FDA/CDC recommend pausing J&J administration due to rare clotting reports. |
| *Apr 23* | **J&J Pause Lifted** | CDC safety panel recommends resuming J&J vaccinations; pause lifted. |
| *Apr 27* | Mask Relaxation | CDC relaxes outdoor mask guidelines for vaccinated people. |
| *May 04* | 70% by July 4 goal | President Biden announces a 70% adult-vaccination goal by July 4. |

### Q.2. Timeline of Interest

Table 18 details the key external events overlapping with our monitoring window. For the camera-ready version we re-curate reference events *before* examining detection results, requiring (i) official FDA/CDC/White House documentation, (ii) being directly vaccine-specific, and (iii) tied to a dateable decision. The qualitative evaluation in the main text assesses whether detection alarms coincide with the onset of public reaction (polarization or anxiety) triggered by these events.

**Calibration robustness and Google Trends proxy.** Compared with the original calibration window (Jan 1 – Feb 26), the revised Phase I (Dec 2 – Jan 30) avoids days immediately preceding vaccine-policy disruptions and utilizes the full dataset. Despite the shifted calibration window, IDD's SPE threshold changes minimally, with similar alarm behavior (5 alarms on 50 Phase II days vs. 4 previously). By contrast, the baselines are sensitive to window choice: F-CPD's threshold shifts from $\approx 0.49$ to $\approx 5.0$, NEWMA's from $\approx 0.019$ to $\approx 0.05$, causing both to produce zero alarms; Hotelling $T^2$ alarms on $13/50$ days (26%), mostly event-unaligned; Scan-B also produces zero alarms. Among the five IDD (SPE) alarms, two align with curated events (Mar 2 J&J EUA issuance; Apr 30 post-J&J-pause discourse reorganization, where the pause Apr 13 and lifting Apr 23 fall within a sparse-data gap Apr 3 – 28, so the first post-gap observation reflects the accumulated shift); May 3 alarms one day before Biden's 70%-by-July-4 goal. The remaining two alarms (Feb 16, Mar 27) do not match curated events; F-CPD and Hotelling $T^2$ also exhibit elevated statistics on these dates, suggesting potential lower-profile distributional activity. IDD's alarm rate (10%) remains far sparser than Hotelling $T^2$'s 26%.

As a proxy ground-truth, we correlate each method's statistic with $|\Delta GT_t|$, the absolute daily change in Google Trends search volume for vaccine-related queries ("vaccine," "covid vaccine," "j&j vaccine," "Pfizer vaccine," "Moderna vaccine"). This forward-differenced proxy captures moments of sudden public attention shift. IDD (SPE) achieves the highest positive Spearman $\rho = 0.17$, while NEWMA ($\rho = -0.20$) and F-CPD ($\rho = -0.19$) show negative correlations (see Figure 11). Individual correlations do not reach significance at $n = 48$; however, the consistent sign pattern (IDD positive, accumulator-type methods negative) provides directional evidence that IDD's alarms better track external attention shifts.

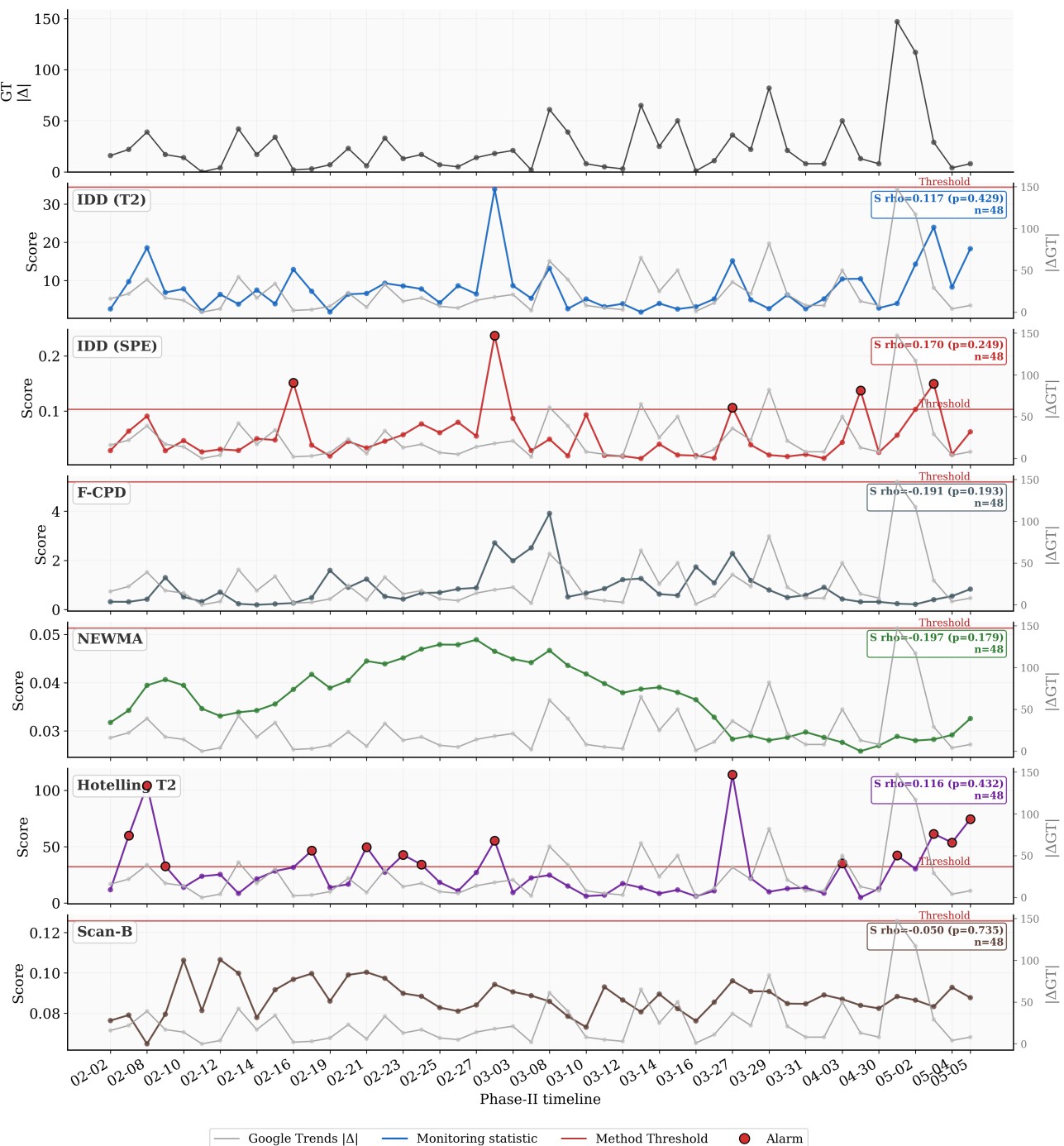

*Figure 11.* Comparison of detection statistics with the Google Trends proxy $|\Delta GT_t|$ over the Phase II monitoring window. IDD (SPE) tracks daily shifts in vaccine-related search volume more closely than accumulator-type baselines (NEWMA, F-CPD), which show negative Spearman correlations with the proxy.

