# OpenReview forum: "Beyond Euclidean Summaries: Online Change Point Detection for Distribution-Valued Data"
_ICML.cc/2026/Conference — ICML 2026 regular_

### Official Review · Reviewer_SoPW · 2026-03-10

**Soundness:** 2
**Presentation:** 3
**Significance:** 3
**Originality:** 3
**Overall Recommendation:** 4
**Confidence:** 3

**Summary:**

This paper studies online change-point detection for distribution-valued data streams. Instead of summarizing each batch with low-order Euclidean statistics, the method models each time step as an empirical distribution and performs monitoring in Wasserstein space. The proposed framework combines a reference barycenter, OT-based tangent-space representation, and subspace monitoring statistics to detect complex distributional shifts.

**Compliance With Llm Reviewing Policy:**

Affirmed.

**Final Justification:**

The paper has a clear motivation and a coherent methodology, and I think the overall idea is meaningful. My main concerns in the original review were about online practicality, the scope of advantage, and the strength of the real-data evidence.

The rebuttal addressed the first two concerns reasonably well. In particular, the added timing results make the online cost much clearer, and the discussion of when the proposed Wasserstein-based approach is most useful helps clarify the paper’s applicability. I still think the real-data evidence is somewhat uneven, especially for the Reddit case study, which remains more illustrative than fully conclusive.

Overall, the rebuttal improved my assessment and moved the paper from a borderline weak reject to a weak accept. I therefore updated my recommendation to 4.

**Key Questions For Authors:**

- **Computational cost.** Please provide a clearer discussion of the online computational cost, ideally beyond asymptotic arguments, to better assess deployment feasibility.
- **Applicability boundary.** It would strengthen the paper to more explicitly characterize the scenarios in which the proposed Wasserstein-based approach is most necessary, versus cases where simpler summaries are already adequate.
- **Reddit evaluation.** The Reddit case study would be more convincing with a stronger quantitative evaluation protocol or more explicit event-level validation.

**Limitations:**

yes

**Strengths And Weaknesses:**

### Strengths

- The problem setting is meaningful and well motivated, especially for applications where each time step is naturally represented as a batch-level distribution rather than a single vector.
- The overall methodology is coherent: the paper does not merely use Wasserstein distance as an extra feature, but builds a full detection pipeline around barycenter-based linearization and monitoring in the tangent space.
- The experiments cover a reasonably diverse set of synthetic and real-world scenarios, including shifts that are difficult to capture with simple moment-based summaries.

### Weaknesses

- **Online practicality.** The practical online applicability is still somewhat unclear, since the method relies on repeated OT-related computations whose runtime cost may be substantial in high-dimensional or high-frequency settings.
- **Uneven real-data evidence.** The AML case is convincing, but the Reddit experiment feels more illustrative than conclusive due to the lack of strong ground-truth evaluation.
- **Scope of advantage.** The paper’s advantage seems most compelling for geometrically complex distribution shifts; however, the boundary of applicability is not fully clarified, especially in simpler settings where lower-order summaries may already suffice.

---

> ### Author Rebuttal · Authors · 2026-03-31
>
> We thank Reviewer SoPW for recognizing the motivation and strength of the work.
>
> ---
> ## 1.Online Practicality
>
> We have included the timing for synthetic experiments across all settings for continuous streams, measuring both Phase I (calibration) in [**Table R1**](https://anonymous.4open.science/r/IDD-icml-C8F6/PhaseI_timing.md) and Phase II (online monitoring) in [**Table R2**](https://anonymous.4open.science/r/IDD-icml-C8F6/PhaseII_timing.md) cost per distribution-valued sample. We summarize the key findings here.
>
> Phase I (offline calibration) ranges from 0.2 ms ($d=1$, $n=50$) to 867 ms ($d=50$, $n=300$) per sample, driven by OT map computations; this is a one-time cost. Phase II (online) is substantially cheaper: under 50 ms for $d \leq 10$, and \~272 ms at $d=50$, $n=300$. Competitors like KDE-Marginal scale similarly (\~172 ms), and F-CPD incurs 12–20 ms but with weaker detection power. Shewhart/Hotelling $T^2$ is fastest (<1.5 ms) but fails to detect higher-order distributional changes.
>
> For the **real case studies**, we report per-sample timing (in seconds) for all methods:
>
> | Case Study | Phase    | OT    | KDE-Full | $\\bar{X}$/Hotelling $T^2$ | NEWMA | ScanB | F-CPD |
> | ---------- | -------- | ----- | ----- | --------------------- | ----- | ----- | ----- |
> | FLOWCAP    | Phase I  | 3.534 | 0.008 | 0.000                 | 0.004 | 0.004 | 0.009 |
> | FLOWCAP    | Phase II | 2.852 | 0.024 | 0.000                 | 0.003 | 0.003 | 0.018 |
> | Reddit     | Phase I  | 0.436 | 0.002 | 0.000                 | 0.001 | 0.001 | 0.002 |
> | Reddit     | Phase II | 0.187 | 0.003 | 0.000                 | 0.002 | 0.002 | 0.003 |
>
> FLOWCAP ($d=7$, $N_t=500$) has a larger batch size than any synthetic setting, resulting in higher OT cost; the ~2.9s Phase II cost is practical given minute-to-hour batch arrivals. For Reddit ($d=20$, $N_t \in [20, 412]$), Phase II cost is ~0.19s, well below the daily monitoring frequency.
>
> Implementation details: exact OT (POT `ot.emd()`) for $d \leq 10$, $N_t \leq 500$; Sinkhorn (`ot.sinkhorn()`, $\varepsilon=0.01$) otherwise. Barycenter computation uses fixed-point LP for $d \leq 5$, and normalizing flows with higher dimension (see response to Reviewer jLnv). A new appendix will detail these choices.
>
> Computational cost is a well-recognized challenge for OT-based methods [1, 2]. Our framework attempts to mitigate this by adapting the OT solver to problem scale and confining expensive computations to offline Phase I. As shown above, Phase II cost remains practical for the deployment scenarios motivating this work.
>
> ---
> ## 2.Scope of Advantage
>
> We appreciate this point. The framework targets moderate-dimensional settings ($d \leq 50$) with moderate arrival frequencies (e.g., daily/hourly observations); scalability beyond this is constrained by OT costs, as discussed above.
>
> Regarding *when* IDD is most beneficial: the method operates on the full geometry of distributions via optimal transport, making it most valuable when changes involve higher-order distributional features (i.e., shifts in shape, spread, modality, or tail behavior) that low-order summaries cannot capture. In our Copula shift and Barycenter scenarios, moment-based methods fail entirely because the mean is preserved, whereas IDD detects these changes reliably.
>
> Conversely, for simpler shift types, methods such as $\bar{X}$/Hotelling $T^2$ monitors may achieve shorter detection delays at lower computational cost. For instance, under the Barycenter shift, MeanChart ($\bar{X}$/Hotelling $T^2$) can attain short $ARL_1$, but at the cost of a relatively high false alarm rate. This highlights the trade-off between detection speed and false alarm control. To help practitioners navigate this trade-off, we will add an explicit discussion characterizing three settings: (1) location-only shifts, where moment-based methods suffice; (2) moderate geometric shifts (e.g., variance or correlation changes), where kernel-based and OT-based methods are competitive; and (3) complex distributional shifts (shape, modality, support), where the proposed method provides a clear advantage (see also our response to Reviewer jLnv).
>
> ---
> ## 3.Uneven Real-Data Evidence
>
> We agree with the reviewer (and Reviewer 62FW, who raised a similar point) that the Reddit case study is more illustrative than conclusive, due to the absence of ground-truth change-point labels — an inherent difficulty of real-world distributional CPD evaluation. We have strengthened this case study by extending the monitoring period, re-establishing event curation criteria, and incorporating Google Trends as a proxy ground-truth metric. Full details are provided in our response to Reviewer 62FW.
>
> ---
> [1] G. Peyré and M. Cuturi, "Computational optimal transport: With applications to data science," Foundations and Trends in Machine Learning, 2019.\
> [2] Ouyang et al., "Sparsification Techniques for Large-Scale Optimal Transport Problems," WIREs Computational Statistics, 2026.

---

> > ### Author Rebuttal · Reviewer_SoPW · 2026-04-03
> >
> > Thank you for the detailed rebuttal. The authors addressed my main concerns reasonably well. In particular, the added discussion and timing results make the online practicality much clearer, and the clarification on the scope of advantage is helpful. I also appreciate the more explicit positioning of the method as being most useful for more complex distributional shifts rather than simpler settings where lower-order summaries may already suffice.
> >
> > I still think the real-data evidence remains somewhat uneven, especially for the Reddit case study, which feels more illustrative than fully conclusive. However, overall the rebuttal improves the paper and addresses my main concerns at a reasonable level. I have updated my assessment accordingly.

---

> > > ### Author Response · Authors · 2026-04-06
> > >
> > > We thank the reviewer for the constructive rebuttal comment and carefully consider our response. It has helped us strengthen the work considerably.

---

### Official Review · Reviewer_62FW · 2026-03-11

**Soundness:** 3
**Presentation:** 3
**Significance:** 3
**Originality:** 3
**Overall Recommendation:** 4
**Confidence:** 1

**Summary:**

IDD is an online change-point detection framework for distribution-valued data streams that maps empirical measures to a Wasserstein tangent space.

**Compliance With Llm Reviewing Policy:**

Affirmed.

**Key Questions For Authors:**

1. Figure 4 shows alarms at other time points that do not correspond to the listed key events, can the authors explain what these additional alarms correspond to, and whether they represent false alarms?

**Limitations:**

Yes

**Strengths And Weaknesses:**

Strength:
1. Extending online CPD to distribution-valued processes is practically important
2. The framework handles both continuous and discrete distribution-valued streams

Weakness:
The Reddit case study lacks ground truth labels for daily sentiment shifts and is evaluated only qualitatively by correlating detection alarms with news events, which does not constitute a controlled performance evaluation.

---

> ### Author Rebuttal · Authors · 2026-03-30
>
> We thank Reviewer 62FW for recognizing the importance and originality of our work. We address the concerns about the Reddit case study below.
>
> **Revised setup.** We strengthen the case study in two ways.
>
> 1. We shift calibration from Jan 1–Feb 26 to the earliest 50-day block (Dec 2–Jan 30), extending Phase II from 28 to 50 days (Jan 31–May 5). This more prospective split avoids calibrating on days immediately preceding vaccine-policy disruptions (late Feb 2021) and utilizes the full dataset.
>
> 2. We re-curate reference events *before* examining detection results, requiring: (1) official FDA/CDC/White House documentation; (2) directly vaccine-specific; (3) tied to a dateable decision. This yields 7 events: J&J EUA recommended/issued (Feb 27/Mar 2), Biden all-adults-by-May-1 (Mar 11), J&J pause/lifted (Apr 13/23), CDC mask relaxation (Apr 27), Biden 70% goal (May 4).
>
> **Result 1: Updated [Figure 4](https://anonymous.4open.science/r/IDD-icml-C8F6/Reddit_Events.png).** Despite the shifted calibration window, IDD's SPE threshold changes minimally, with similar alarm behavior—5 alarms on 50 Phase II days (vs. 4 previously). By contrast, baselines are sensitive to window choice: F-CPD's threshold shifts from ≈0.49 to ≈5.0, NEWMA's from ≈0.019 to ≈0.05, causing both to produce zero alarms. Hotelling T² alarms on 13/50 days (26%), mostly event-unaligned; Scan-B also produces zero alarms. This demonstrates IDD yields a calibration-robust statistic, whereas accumulator-type baselines are fragile to the reference window for calibration.
>
> Among the 5 IDD (SPE) alarms, two perfectly aligns with curated events: Mar 2 (J&J EUA issuance); Apr 30 capturing post-J&J-pause discourse reorganization (the pause Apr 13 and lifting Apr 23 fall within a data gap Apr 3–28 due to sparse Reddit activity in  the original dataset, so the first post-gap observation reflects the accumulated shift). May 3 alarms one day before Biden's 70%-by-July-4 goal. The remaining two (Feb 16, Mar 27) do not match curated events and *may be false alarms*. However, F-CPD and Hotelling T² also show elevated statistics on these dates, suggesting potential distributional activity from concurrent lower-profile developments that collectively alter the embedding distribution. IDD's alarm rate (10%) remains far sparser than Hotelling T²'s 26%.
>
> **Result 2: Scalar sentiment comparison.** Melton et al. (2021) [1], who analyzed this dataset, applied lexicon-based sentiment scoring (mapping each comment to a scalar polarity by averaging word-level sentiment) and concluded that sentiment "has not meaningfully changed since December 2020." This is the limitation motivating our approach. Events like the J&J pause and its lifting do not make discourse uniformly more negative or positive. Insteadt, vaccine-hesitant clusters expand, trust-related topics emerge, and pro-/anti-vaccine subpopulations reorganize in semantic space. These opposing shifts cancel in scalar aggregate. IDD detects such geometric changes in the full 20D embedding distribution that scalar summaries cannot capture. The contrast between stable sentiment and IDD's event-aligned alarms provides direct evidence for the value of distribution-valued monitoring.
>
> **Result 3: Google Trends as proxy groundtruth.** We correlate each method's statistic with $|ΔGT_t|$, the absolute daily change in Google Trends search volume for vaccine-related queries ("vaccine," "covid vaccine," "j&j vaccine," "Pfizer vaccine," "Moderna vaccine," etc.)  In [**Figure R3**](https://anonymous.4open.science/r/IDD-icml-C8F6/Google_Trend_Diff_Comparison.png). This forward-differenced proxy captures moments of sudden public attention shift. IDD (SPE) achieves the highest positive Spearman ρ=0.17, while NEWMA (ρ=−0.20) and F-CPD (ρ=−0.19) show negative correlations. Individual correlations do not reach significance at n=48; however, the consistent sign pattern—IDD positive, accumulator-type methods negative—provides directional evidence that IDD's alarms better track external attention shifts.
>
> **Precedent.** Qualitative event-correlation is standard for real-world CPD: Altamirano et al. (ICML 2023)[2] validate BOCD on UK bond yields by aligning change points with political events; Horváth et al. (2021, *Ann. Statist.*)[3] validate S&P 500 monitoring purely via correlation with financial crises—both univariate. Our evaluation goes further on a 20D distribution-valued stream with scalar-sentiment null results, a Google Trends proxy, and calibration robustness analysis.
>
> ---
> [1] Melton et al. (2021). Public sentiment analysis and topic modeling regarding COVID-19 vaccines on Reddit. *J. Infection and Public Health*, 14(10), 1505-1512.\
> [2] Altamirano et al. (ICML 2023). Robust and scalable Bayesian online changepoint detection. pp. 642-663.\
> [3] Horváth et al. (2021). Monitoring for a change point in a sequence of distributions. *Ann. Statist.*, 49(4), 2271-2291.

---

> > ### Author Rebuttal · Reviewer_62FW · 2026-04-03
> >
> > Thank you for your response and clarification. I acknowledge your explanation, although our overall assessment remains unchanged.

---

> > > ### Author Response · Authors · 2026-04-06
> > >
> > > We thank the reviewer for the constructive rebuttal comment and carefully consider our response. It has helped us strengthen the work considerably.

---

### Official Review · Reviewer_jLnv · 2026-03-13

**Soundness:** 2
**Presentation:** 3
**Significance:** 4
**Originality:** 3
**Overall Recommendation:** 4
**Confidence:** 3

**Summary:**

The paper introduces a framework for change-point detection (CPD) in distribution-valued streams. The problem statement assumes that at time t we obtain a batch of N_t measurements. Instead of computing simple summaries, such as mean vectors, the paper proposes to monitor the empirical probability measures \mu_t. Therefore, the problem is reformulated as a CPD over distribution-valued streams. The paper presents a theory allowing to move from the infinite dimensional space of probability distributions to a finite dimensional space, where monitoring is feasible. They method is based on the theory of optimal transportation and Wasserstein spaces. In particular, each empirical distribution \mu_t is mapped to the tangent space centered at the Frechet barycenter, which is computed from in-control samples. The “dimensionality reduction” is performed through a projection over the first K components of the functional PCA, estimated from in-control samples. In the end, monitoring is performed on the Hotelling and Squared Prediction Error, which can be easily computed from the eigenvalues and projection coefficients of the observed samples. Thresholds are computed numerically imposing a target ARL0.

**Compliance With Llm Reviewing Policy:**

Affirmed.

**Final Justification:**

Considering the rebuttal exchange, I confirm my original assessment, weak accept.

**Key Questions For Authors:**

Could you improve the presentation of the Algorithm 1?

Could you discuss the practical implementation in more details? For instance, what algorithm can one use to solve the optimization problem in (5) or how to compute the barycenter?

Could you expand the discussion of the experimental results, including all methods?

Please revise the notation and solve any inconsistency.

**Limitations:**

Authors should discuss limitations with respect to competing methods that achieve comparable/higher performance on some experiments.

**Strengths And Weaknesses:**

Soundness. The proposed method is based on the theory of optimal transport (OT), which uses Wasserstein metrics to measure the distance between probability distributions. The theoretical part of the paper seems sound and grounded on existing OT results. The proofs of additional results are reported in the appendices, while established results point to authoritative references in OT.

The experiments deal with both synthetic and real data. I believe the discussion of the experiments is somehow lacking, especially regarding competitors that outperform the proposed method. For instance, the C_CHART method in Figure 2a achieve shorter delays. Authors should acknowledge this fact and discuss possible reasons. Moreover, the discussion of the results of Figure 3 seems not complete. First of all, the discussion does not mention the NEWMA model that achieves comparable performance for small ARL0 and outperforms the proposed method at high ARL0. Moreover, the text mentions that Figure 3 reports the ARL1, but it is not present in the figure. In all experiments, authors should discuss the performance of all methods, not only those that are visibly outperformed by the proposed solution. Moreover, I believe authors should report the length of the sequence used to compute the Frechet barycenter in each experiments. This is somehow a limitation of the method, and should be discussed more in details, especially when testing on real data which might have few pre-change samples.

Presentation. Overall the paper is well structured and well written, with few exceptions. The discussion of the experiments can be improved by describing the results of all methods, including those appearing to outperform the proposed solution. Algorithm 1 currently has very little practical utility. It can be improved by illustrating in a more schematic way the solution (e.g. more like a pseudo-code) and reducing the amount of cluttered text. All and only relevant notations should be used (e.g., what is \tau=\infty ?). Additionally, authors might consider to split the algorithm description into two environments, one for the pre-change calibration and one for the monitoring process. Additionally, more practical implementation details should be reported. For instance, which algorithms can be used to solve the optimization problems involved in Algorithm 1?
I believe there are notations inconsistencies throughout the paper that should be solved. Here few examples:
- At line 31 (2nd column), in the empirical measure, the batch size is denoted by n_t, while later in the paper is N_t. Moreover, is X_{t,j} or x_{t,j}? Are you discriminating between random variables and realizations? Moreover, you should state clearly what are the Xs.
- At line 67, there is an apparently inconsistent \mathcal{W}
- At line 149, I think \mu_i should be \mu_t
- In Equation (8) it is not clear what is Y
- the pre-change distribution is denoted by P_1 in Equation (3), but then the expected value of the ARL0 is defined as E_\infty

Significance. The paper addresses the change-point detection problem for distribution-valued stream. Albeit this is a more niche settings than traditional CPD for Euclidean-valued data, it has practical applications and also complex Euclidean-valued batch streams could be transformed to distribution-valued, as shown in the paper. The scope is general enough to allow several applications and extensions.

Originality. The originality of the contribution seems adequate. The proposed  theoretical framework appears a significant advancement for the study of distribution-valued CPD.

---

> ### Author Rebuttal · Authors · 2026-03-31
>
> We  thank the reviewer for the constructive feedback.
>
> ---
> ## 1. Comprehensive Discussion
>
> We agree and  provide a summary of the results across all experiments:
>
> | Scenario | IDD Advantage | Methods That Match/Beat IDD |
> |---|---|---|
> | Barycenter Change | Ties with best | Hotelling T², Shewhart, NEWMA |
> | Multimodal Reweight | Strong (d≥5) | Log-KDE competitive in 1D |
> | Copula Shift | Strong (small N) | Log-KDE competitive at large N |
> | Gaussian Translation | Disadvantage (small δ) | Hotelling T² better for pure mean shifts |
> | Poisson Spike (Fig 2a) | Moderate disadvantage | C_CHART better (specialized detector) |
> | Ordered Categorical | Advantage | — |
> | AML FlowCAP (Fig 3) | Best overall | NEWMA comparable at low ARL₀ |
>
> *C_CHART (Fig 2a):* We find C_CHART achieves shorter delays for ARL₁. As a specialized Poisson count detector monitoring $S_t = \sum x_{t,i}$, it is optimally suited for spike injection. IDD cannot match domain-specific detectors when the parametric model is known; its value lies in settings where the shift type is unknown a priori.
>
> *NEWMA (Fig 3):* We thank the reviewer for noting this omission. NEWMA achieves comparable F1-scores at low ARL₀ (30–50), where lenient false-alarm thresholds give its kernel statistic sufficient power. However, NEWMA degrades at higher ARL₀ (>75), revealing lower statistical efficiency under stricter false-alarm control.
>
> *Hotelling T² (Tables 1–2):* For pure Gaussian mean shifts, Hotelling T² outperforms IDD at small δ, consistent with Theorem F.1: it directly monitors the sufficient statistic, while IDD introduces OT estimation variance.
>
> We will revise Section 4 by adding more discussion, for example:
> > *"IDD is not uniformly superior. For location shifts, Hotelling T² is more efficient. Similarly, for count data with known parametric structure, specialized charts outperform IDD. IDD's strength lies in complex geometric shifts—reweighting, dependency changes, shape deformations—where moment-based detectors lose power."*
>
> ---
> ## 2. Figure 3
> We reported ARL₁ in Figure 3 by mistake as we missed the ARL₀ vs ARL₁ figure for the AML case study. [**Figure R2**](https://anonymous.4open.science/r/IDD-icml-C8F6/AML_ARL0vsARL1.png) will be added in the revision. This figure demonstrates that IDD achieves the lowest ARL₁ across the ARL₀ range, with NEWMA as the closest competitor at low ARL₀.
>
> ---
> ## 3. Pre-Change Sequence Length ($n_0$)
> The pre-change calibration lengths are: $n_0 = 300$ for all synthetic experiments and FlowCAP-II, and $n_0 = 50$ for Reddit (constrained by data availability). The choice of $n_0$ governs three estimation tasks: (1) the Fréchet barycenter, (2) the MFPCA eigendecomposition, and (3) the empirical threshold quantiles. Corollary 3.11 quantifies the finite-sample effect on (3): ARL₀ includes a correction of $2/(n_0+1)$, negligible for $n_0=300$ but non-trivial for $n_0=50$, partially explaining the qualitative-only Reddit evaluation. We will add a discussion of $n_0$ sensitivity as a limitation, including guidance on minimum calibration lengths for practitioners.
>
> ---
> ## 4. Revised Algorithm 1 & Implementation Details
>
> We agree Algorithm 1 needs improvement and will include them in the revision. Regarding $\tau = \infty$: this initializes the stopping time to "no alarm yet".
> In the revision we will: (1) split into *Algorithm 1 (Calibration)* and *Algorithm 2 (Monitoring)* as clean pseudo-code, (2) remove the unused $\tau$, and (3) use only relevant notation.
>
> We will add [**Table R4**](https://anonymous.4open.science/r/IDD-icml-C8F6/Example_Implementation_Details.md) in appendix reporting: (1) OT solver per regime: exact LP via `ot.emd()` for $d \leq 10, N_t \leq 500$; Sinkhorn otherwise with all hyperparameters ($\varepsilon$, iterations, convergence threshold); (2) barycenter method per dimension:  closed-form ($d=1$), fixed-point LP ($d \leq 5$), Conditional Normalizing Flows ($d > 5$) with architecture details; (3) per-step computational cost. We acknowledge per-step OT cost as a limitation motivating future amortized approximations.
>
> ---
> ## 5. Notation Inconsistencies
> All issues will be corrected in detail in the revision:
> - $n_t$/$N_t$ is unified to $N_t$.
> - $X_{t,j}$ is the random variable, while $x_{t,j}$ is the realization.
> - fix $\mathcal{W} \to W_2$ (line 67).
> - fix $\mu_i \to \mu_t$ at line 149; $Y$ in Eq. (8) as the target r.v. in the coupling, "$(X,Y) \sim \pi_t$, $X \sim \bar{\mu}$, $Y \sim \mu_t$".
> - $E_\infty[\cdot]$ is the expectation under $H_0$ (change point $\kappa = \infty$).
>
> ---
> ## 6. Table 9 Duplication
>
> We identified that Table 9 (Copula Shift, d=50) contains values identical to Table 6 (Barycenter, d=50) due to a copy-paste error. The corrected table is available at [**Table R5**](https://anonymous.4open.science/r/IDD-icml-C8F6/Table%209.md).
>
> ---
> ## 7. Reddit Case Study
>
> Please see our response to Reviewer 62FW for a detailed discussion of the Reddit sentiment analysis.

---

> > ### Author Rebuttal · Reviewer_jLnv · 2026-04-02
> >
> > The authors have addressed all my concerns and are improving the manuscript accordingly. The only doubt remaining is their response regarding NEWMA in Fig 3. The discussion given in the rebuttal seems in contrast with the plot, where NEWMA achieves higher F1 for ARL0>75. Authors should clarify this apparent inconsistency.

---

> > > ### Author Response · Authors · 2026-04-03
> > >
> > > We thank the reviewer for the rebuttal comment, and thank for pressing on this point. The apparent inconsistency — NEWMA achieving higher F1 than IDD (Ours) for ARL₀ > 75 yet higher (worse) ARL₁ — is not a contradiction but arises because F1 and ARL₁ capture fundamentally different aspects of detection performance.
> > >
> > > **How F1 and ARL₁ are computed**:
> > > In our FLOWCAP case study, Phase II consists of a test stream of 300 samples, constructed by injecting 80% AML-positive samples (OOC) (randomly drawn from the AML patient pool) interspersed with healthy (IC) samples, repeated across 10 replications. For each method and ARL₀ target:
> > >
> > > - ARL₁ (delay) is the mean first-detection delay: the index of the first alarm minus the index of the first OOC sample. It measures how quickly the method raises its first alarm after the first anomaly appears.
> > > - F1 is computed from batch (i.e., sample)-level binary classification over the entire Phase II stream: every sample is labeled alarm/no-alarm, and standard precision/recall/F1 are calculated against the true IC/OOC labels.
> > >
> > > Because these two metrics evaluate different quantities (i.e., speed of first detection vs. overall labeling accuracy across all batches), they need not rank methods identically in the two figures.
> > >
> > > We plot Figure 3 and Figure R2 side by side here: [**Figure R5**](https://anonymous.4open.science/r/IDD-icml-C8F6/ARL0_tradeoff_2panels.png). The detailed comparison when ARL₀ > 75 is as follows:
> > >
> > > 1. IDD detects faster but is more selective. IDD achieves near-perfect precision (≈ 0.99), meaning almost every alarm it raises is a true OOC batch. However, it raises fewer total alarms, yielding lower recall (~0.31–0.32). Its first alarm arrives earlier (delay ≈ 2.2 vs. 3.6–3.8), which is the quantity plotted as ARL₁ in Figure R2.
> > >
> > > 2. NEWMA raises more alarms, boosting recall and F1. NEWMA produces a broader alarm pattern: it correctly flags more OOC batches (recall ≈ 0.52) but at lower precision (≈ 0.83). The higher recall lifts its F1 above IDD's, but its first alarm arrives later.
> > >
> > > In short, IDD trades recall for precision and detection speed; NEWMA trades precision and speed for recall. Thus,  when ARL₀ > 75, NEWMA has higher F1 while IDD has a smaller ARL₁.
> > > The two methods reflect different tradeoffs between first-detection speed and overall labeling coverage. IDD's main advantage here is consistently faster first detection (lowest ARL₁) with very high precision across the entire ARL₀ range. We will clarify the distinction between these metrics in the revised manuscript.

---

### Official Review · Reviewer_z9k4 · 2026-03-15

**Soundness:** 2
**Presentation:** 2
**Significance:** 3
**Originality:** 2
**Overall Recommendation:** 4
**Confidence:** 2

**Summary:**

This paper proposes a geometry-aware online change-point detection framework for a sequence of empirical distributions, treating streaming batch data as a stochastic process on the 2-Wasserstein space. The proposed method maps each empirical distribution to a tangent space and uses the functional Hotelling T^2 statistics and the residual–based Q–statistic as parallel monitoring statistics.

**Compliance With Llm Reviewing Policy:**

Affirmed.

**Final Justification:**

The rebuttal addressed my main concerns and I have updated my score accordingly.

**Key Questions For Authors:**

1. The proposed two-layer generative structure, where pre- and post-change regimes are each characterized by a family of distributions ($\nu_t \sim \mathbb{P}_1$ and $\nu_t \sim \mathbb{P}_2$ respectively) rather than a single fixed distribution, resembles the composite or robust change-point detection setting, in which the underlying data distribution is allowed to vary within a family as long as the pre- and post-change families differ. I am not entirely sure if this analogy is accurate, and would be curious to hear the authors' thoughts on how the proposed framework relates to the composite or robust change-point detection literature.

Moreover, if one simply concatenates all batch samples into a single data stream, then treating $\mathbb{P}1$ (and $\mathbb{P}2$) as a Bayesian prior over $\nu_t$, the marginal distribution of each $x_{t,i}$ can be derived by marginalizing over $\nu_t$ (this is most straightforward when $\mathbb{P}1$ is a discrete distribution over a finite set of distributions, in which case the marginal $p(x_{t,i})$ is a finite mixture distribution). As long as the pre- and post-change marginals are distinct, a standard change-point detector could in principle be applied directly to the concatenated non-batched stream, in what sense would such an approach fail to solve the same problem?

2. Could the authors clarify how the point-based baselines (such as Scan-B) are adapted to the batched setting? If these methods process all $N$ individual points per batch as separate observations in a streaming manner, they might be able to raise alarm before enumerating all samples within the current batch and thus trigger a smaller delay. On the other hand, the proposed IDD method needs to take all $N$ samples together as input and then update the detection statistics. It is not described clearly how the comparisons with baselines methods are actually performed.

3. The hypothesis in eq(3) is a bit confusing and may need further clarification. Shall it be the underlying data-generating distribution $\nu_t \sim P_1$? The $\mu_t$ here denotes the empirical distribution (and under potentially different sample sizes N_t), thus it seems $\mu_t$ are not necessarily iid.

4. For statistical process control charts, in addition to the average run length ARL0 under a no-change regime, it is usually also of great significance to analyze the ARL1 when there is a change-point in the origin. It would enhance the theoretical contribution if the authors could also comment on the guarantee of such detection delay.

5. Algorithmically, I wonder how robust the proposed method is to the randomness in the barycenter $\bar{\mu}$ determined from n0 pre-change samples. If there are multiple reference streams, would it be better and more robust to construct multiple (iid) reference streams and thus multiple barycenters, and then averaging the $T^2_{t}$ and $SPE_t$ statistics across barycenters？

Typos: (i) eq(7) should end with a period. (ii) in eq(16) it should be $ARL_0$ as in Corollary 3.11.

**Limitations:**

yes

**Strengths And Weaknesses:**

Strength:
- The problem of distribution-valued change detection is an important genralization of traditial change-point detection on the instance level. As compared with traditional setttings which assume the distribution of each instance changes from P0 to P1, the framework considered in this work can handle natural variations within each batch.
- The numerical results demonstrate the advantage of the proposed method.

Weakness:
- If I didn't misunderstand the scope of this paper, the paper lacks an in-depth discussion on the relationship between the proposed framework and traditional change-point detection methods based on non-batched data streams (see Questions).

- The paper does not clearly specify how point-based baselines (e.g., Scan-B) are adapted to the batch setting (also see Questions).

- The synthetic experiments generate batch-to-batch variability via random convex deformation maps applied to a fixed base sample, rather than by sampling $\nu_t$ from a parent law $\mathbb{P}_1$​ as stated in Section 3. Though they could be equivalent in a certain sense, the connection is not made explicit, and the experiments feel somewhat artificial. It would be clearer if the authors could elaborate more on the connection between the constructed synthetic example and the specific setup in Sec 3, and if possible, provide an example of an interpretable change in the parent law.

---

> ### Author Rebuttal · Authors · 2026-03-30
>
> ## Q1
> The analogy is insightful, but the settings differ structurally.
>
> In composite/robust CPD [1,2], a scalar/vector observation $x_t \sim P$ where $P \in \mathcal{F}$ is *fixed but unknown*; the goal is to detect when the fixed $P$ transitions from family $\mathcal{F}_0$ to $\mathcal{F}_1$, with robustness to which element is the true $P$. In our setting, the unit of observation at time $t$ is an empirical distribution $\mu_t$ as a random draw from an underlying law $\mathbb{P}$ over Wasserstein space. The between-batch variability of $\mu_t$ is the signal being modeled, not uncertainty to guard against. No pre-specified parametric family is assumed; $\mathbb{P}$ is characterized nonparametrically via the Fréchet mean and tangent-space covariance estimated from the reference stream. Hence the proposed method tends not to solve the same problem as robust/composite CPD.
>
> Concatenating batches and applying a standard detector accesses only the marginal $\bar{\nu} = \int \nu\, d\mathbb{P}(\nu)$. Yet $\mathbb{P}$ can change while $\bar{\nu}$ remains identical whenever the two laws differ in higher-order structure. Scenarios S2 (multimodal reweight) and S3 (copula shift) are constructed so that $\bar{\nu}_1 = \bar{\nu}_2$; concatenation-based detectors are powerless there, while our geometry-aware statistics detect the change.
>
> We verify this with a **KDE-Marginal** baseline monitoring per-coordinate marginal densities via functional $L^2$ distances.
> **[Table R3](https://anonymous.4open.science/r/IDD-icml-C8F6/KDE-Marginal.md)** compares KDE-Marginal against IDD on copula-shift (S3) at $\alpha = 0.05$. For $d \geq 5$, IDD achieves $ARL_1 = 1$ across all batch sizes, whereas KDE-Marginal detects slowly ($ARL_1 \geq 7$). This demonstrates IDD solves a strictly harder problem than monitoring the marginal stream.
>
> ---
> ##  Q2
> We adapt Scan-B by treating each batch $\mu_t$ as one observation, following [3].
> Each batch is embedded via Random Fourier Features into a kernel mean embedding $\psi(\mu_t) \in \mathbb{R}^D$, where $\|\psi(\mu_s) - \psi(\mu_t)\|$ approximates the Maximum Mean Discrepancy between $\mu_s$ and $\mu_t$.
> Mid-batch alarms are inapplicable: under Assumption 3.1, within-batch points are i.i.d. from the same $\nu_t$, so the target change can only be assessed by comparing entire batches across time.
>
> ---
> ## Q3
> The reviewer is correct. The hypothesis should be $H_0: \nu_t \stackrel{iid}{\sim} \\mathbb{P}_1$, not on $\mu_t$, since under varying $N_t$ the $\mu_t$'s are not identically distributed. This assumption is standard [4, 5].
>
> ---
> ## Q4
> We provide the ARL1 guarantees as follows.
> Under sustained post-change batches from a new law, the statistics $(T^2_{t,K},\ \mathrm{SPE}_t)$ are conditionally i.i.d., so $\mathrm{ARL}_1 = 1/p_1$ .
> Under Assumption 3.5, $T^2 \sim \chi^2_K(\delta^2_T)$ giving detection probability $p_T$ , and SPE follows a generalized noncentral chi-square with probability $p_S$ .
> Under independence: $\mathrm{ARL}_1 = 1/[1-(1-p_T)(1-p_S)]$ . Full derivation will be in the revised appendix.
>
> ---
> ## Q5
> The ensemble idea is reasonable but requires splitting the reference stream, reducing estimation quality. A single barycenter from all $n_0$ samples is preferable: empirical barycenters enjoy finite-sample concentration [6], and our calibration computes thresholds from the same $\bar{\mu}$, so control limits absorb estimation variability. A sensitivity discussion will be added.
>
> ---
> ## Weakness 3
> The connection between our construction and Section 3 is exact. By Lemma 2.4.5 of [7], any random distributions can equivalently be defined through a random measurable map applied to a fixed reference measure. Concretely, drawing i.i.d. parameters $\omega_t$ and applying deformation $T_{\omega_t}$ to $\bar{\mu}$ is not an approximation. It is sampling $ \nu_t \sim \mathbb{P}\_1 $ directly. Otto's calculus requires $ T\_{\omega_t}-id $ to be the derivative of a convex potential. Hence justifying the construction of the synthetic example.
>
> The three scenarios correspond to interpretable changes: S1 shifts the mean; S2 reweights the multimodal structure; S3 alters the dependence structure of $\omega$ while preserving its marginals (changing within-batch geometry without affecting pointwise distributions). We will add these descriptions in Section 4.1.
>
> ---
> [1] Li & Yu, "Adversarially robust change point detection," NeurIPS 2021.\
> [2] Altamirano et al., "Robust and scalable Bayesian online change-point detection," ICML 2023.\
> [3] Wang et al., "Non-parametric online change point detection on Riemannian manifolds," ICML 2024.\
> [4] Dubey & Müller, "Fréchet change-point detection," Annals of Statistics 2020.\
> [5] Horváth et al., "Monitoring for a change point in a sequence of distributions," Annals of Statistics 2021.\
> [6] Ahidar-Coutrix et al., "Convergence rates for empirical barycenters," PTRF 2020.\
> [7] Panaretos & Zemel, "An invitation to statistics in Wasserstein space," Springer 2020.

---

> > ### Author Rebuttal · Reviewer_z9k4 · 2026-04-04
> >
> > I appreciate the author's detailed responses. I have updated my score.

---

> > > ### Author Response · Authors · 2026-04-06
> > >
> > > We thank the reviewer for the constructive rebuttal comment and carefully consider our response. it has helped us strengthen the work considerably.

---

### Decision · Program_Chairs · 2026-04-30

**Decision:**

Accept (regular)

**Comment:**

This paper addresses online change-point detection for distribution-valued data streams, a setting that is both practically relevant and technically nontrivial. The central idea is to move beyond fixed Euclidean summaries and instead develop a geometry-aware monitoring framework in Wasserstein space. Reviewers found the problem well motivated, the overall technical approach coherent, and the contribution sufficiently original to be of interest to the ICML audience. In particular, the paper was viewed as offering a meaningful extension of online monitoring methodology to a richer class of data objects, with potential value in applications where changes are expressed through distributional geometry rather than simple low-order summaries.

The main concerns centered on the strength and clarity of the empirical validation, the practical implications of the OT-based computation, and the need to better delineate the regimes in which the proposed method provides the most benefit relative to simpler alternatives. The rebuttal addressed these points reasonably well by clarifying the method's intended scope, improving the discussion of baselines and computational cost, and strengthening aspects of the empirical presentation. Some limitations remain, particularly in the uneven strength of the real-data evidence, but the overall consensus among reviewers after discussion is positive. On balance, I believe the paper makes a technically solid and sufficiently novel contribution, and I support acceptance.